# Pretrain–Test Task Alignment Governs Generalization in In-Context Learning

**Mary Letey, Jacob A. Zavatone-Veth, Yue M. Lu,** and **Cengiz Pehlevan**
Harvard University
`{maryletey,jzavatoneveth}@fas.harvard.edu,`
`{yuelu,cpehlevan}@seas.harvard.edu`

## Abstract

In-context learning (ICL) is a central capability of Transformer models, but the structures in data that enable its emergence and govern its robustness remain poorly understood. In this work, we study how the structure of pretraining tasks governs generalization in ICL. Using a solvable model for ICL of linear regression by linear attention, we derive an exact expression for ICL generalization error in high dimensions under arbitrary pretraining–testing task covariance mismatch. This leads to a new alignment measure that quantifies how much information about the pretraining task distribution is useful for inference at test time. We show that this measure directly predicts ICL performance not only in the solvable model but also in nonlinear Transformers. Our analysis further reveals a tradeoff between specialization and generalization in ICL: depending on task distribution alignment, increasing pretraining task diversity can either improve or harm test performance. Together, these results identify train-test task alignment as a key determinant of generalization in ICL.

## 1 Introduction

Pre-training on simple next-token prediction enables Transformer models to acquire a remarkably broad array of capabilities, from language translation to code generation and mathematical reasoning (Achiam et al., 2023; Anthropic, 2024; DeepSeek-AI et al., 2025; Vaswani et al., 2017). Among the emergent abilities that enable Transformers to flexibly execute a myriad of tasks, their capacity for in-context learning (ICL) is particularly striking, as it allows for test-time task execution without task-specific pretraining (Von Oswald et al., 2023; Wei et al., 2022). In other words, ICL reflects the ability to emergently meta-learn a learning algorithm during pretraining, which is then applied to learn from data within a context at test time (Akyürek et al., 2023; Raventós et al., 2023; Zhang et al., 2024a).

For ICL to be effective, the tasks encountered at test time must not be totally unrelated to those encountered during pretraining, as there is no free lunch. Though substantial theoretical attention has been devoted to the question of why and how ICL emerges and how well the resulting algorithms perform, this key issue of how pretraining tasks should be selected to enable ICL in the real, test-time world remains underexplored (Lu et al., 2025; Zhang et al., 2024a;b). This motivates the central question of our work:

> **Central Question**
>
> How does mismatch between the structure of tasks seen in pretraining and the structure of tasks faced at test time affect the ability of in-context learners to generalize?

Here, we investigate how pretrain-test task alignment affects generalization in a simple model setting: ICL of linear regression. Our main contributions are as follows:

- We give a precise mathematical analysis of the performance of a simplified linear Attention module learning to do linear regression in-context. We model pretraining and test task distributions

with arbitrary covariance structure, generalizing previous works on this model that assumed identical task distributions (Lu et al., 2025; Zhang et al., 2024a). In this solvable model, ICL generalization is governed by a particular alignment metric between pretraining and test task distributions.

- Though derived for a simplified linear model, we show that this alignment measure predicts the generalization performance of nonlinear Transformers trained to do linear regression in-context.

- Finally we show in the solvable model that it is not always optimal from a generalization standpoint to pretrain on the same distribution of tasks that the model will encounter at test time, as echoed in kernel regression (Canatar et al., 2022).

In all, our work sheds light on how pretrain-test task alignment shapes the performance of in-context algorithms. It reveals how the inductive biases of Transformers can result in optimal task *misalignment*: rather than teaching to the test, a different curriculum of pretraining tasks may better enable a Transformer to learn the algorithm that enables it to generalize well at test time.

## 1.1 RELATED WORK

**Empirical studies of ICL.** Empirical work has shown that LLMs can learn diverse tasks from examples alone, with performance improving predictably with scale (Achiam et al., 2023; Anthropic, 2024; DeepSeek-AI et al., 2025; Vaswani et al., 2017; Von Oswald et al., 2023; Wei et al., 2022). Several studies document how architectural components, such as attention heads or MLP layers, are recruited during training to implement ICL (Kratsios & Furuya, 2025; Reddy, 2024; Tong & Pehlevan, 2025; Zhang et al., 2024b). Various works have also focused on understanding what algorithms transformers can learn to perform, including gradient descent, Bayesian inference, and compression. (Ahn et al., 2023; Cole et al., 2025; Elmoznino et al., 2025; Lee et al., 2025; Liu et al., 2024; Mahankali et al., 2023; McCracken et al., 2025; Shen et al., 2025; Singh et al., 2023; Wurgaft et al., 2025; Zhang et al., 2023). Others have explored the role of task diversity in shaping generalization, showing that diverse pretraining induces transitions from memorization to generalization (Raventós et al., 2023). The specific effect of data structure and its role in ICL emergence has also been studied empirically, notably by Chan et al. (2022) highlighting the importance of *anisotropic* data for the emergence of ICL abilities. Our work provides a theoretical counterpart to these empirical investigations, as we model the generalization effects of structured task distributions.

**Theoretical studies of ICL.** Theoretical work has flourished in simplified Transformer models, particularly with linear or kernelized attention. A number of studies show that these architectures can implement classical learning algorithms, including kernel regression, ridge regression, or gradient descent, purely from in-context tokens (Akyürek et al., 2023; Bai et al., 2023; Li et al., 2023; Von Oswald et al., 2023; Zhang et al., 2024b). These insights have advanced our mechanistic understanding of ICL as algorithm emulation. However, most of these works make restrictive assumptions. Commonly, data is drawn from isotropic Gaussians, train and test distributions are matched, and generalization is studied only in the infinite-sample or population limit. Even recent studies of finite-sample ICL retain these simplifying assumptions *i.e.* without full generality of training and testing task distribution (Fu et al., 2023; Li et al., 2024; Lu et al., 2025; Zhang et al., 2024b). A notable exception is the work of Goddard et al. (2025) that studies tasks sampled from separate portions of a spherical task manifold. Our work advances this "task generalization" frontier by deriving an exact expression for the ICL generalization error in the presence of arbitrary task covariances and finite-sample regimes. This allows us to explore how task-structure mismatch affects generalization, a setting largely absent from prior theoretical models.

**Train-Test task alignment in other settings.** Outside of ICL, the idea of train-test task alignment in linear regression has been studied in the context of out-of-distribution generalization under target vector and covariate shifts. In particular, past works have studied which measures of alignment between train and test feature covariance matrices determine out-of-distribution generalization in high-dimensional ridge regression (Atanasov et al., 2025; Canatar et al., 2021b; Patil et al., 2024; Tripuraneni et al., 2021). These works build on a substantial body of results showing how the generalization performance of ridge regression is determined by the structure of the training feature covariance—principal directions with larger population variance are learned first—and the alignment of the task vector with those high-variance directions (Advani et al., 2020; Atanasov et al., 2024; Canatar et al., 2021a; Dobriban & Wager, 2015; Hastie et al., 2022). Our work brings this

perspective into the ICL setting by analyzing the effects of the pretraining covariance, and pretrain-test misalignment on generalization. We show that even in simple linear regression settings, structure mismatch induces rich, nontrivial behavior, reinforcing the broader principle that distributional alignment between pretraining and test-time data is a key driver of generalization.

## 2 MODEL SETUP

We begin by setting up the solvable model used to derive the results of this work. This setup builds on previous works that apply a reduced-parameter version of linear attention to linear regression ICL (Lu et al., 2025; Wang et al., 2020; Wu et al., 2024; Zhang et al., 2024a).

**ICL of linear regression.** We consider an in-context regression task: the input to the model is a sequence of the form $\{\boldsymbol{x}_1, y_1, \boldsymbol{x}_2, y_2, \ldots, \boldsymbol{x}_\ell, y_\ell, \boldsymbol{x}_{\ell+1}\}$, and the required output is the matching $y_{\ell+1}$ corresponding to $\boldsymbol{x}_{\ell+1}$. This input is called a context, and $\ell$ the context length. We consider the case that the relationship between $\boldsymbol{x}$ and $y$ is approximately linear:

$$y_i = \langle \boldsymbol{x}_i, \boldsymbol{w} \rangle + \epsilon_i \tag{1}$$

for noise $\epsilon_i$ and task vector $\boldsymbol{w}$. Thus, the model needs to form an estimate of $\boldsymbol{w}$ using $\{\boldsymbol{x}_1, y_1, \boldsymbol{x}_2, y_2, \ldots, \boldsymbol{x}_\ell, y_\ell\}$ and then apply it to $\boldsymbol{x}_{\ell+1}$ to estimate $y_{\ell+1}$.

**Pretraining data.** The pretraining data batch will contain $n$ sample sequences of the above form, *i.e.*, for $\mu = 1, \ldots, n$, the $\mu$th sample sequence $\{\boldsymbol{x}_1, y_1, \boldsymbol{x}_2, y_2, \ldots, \boldsymbol{x}_\ell, y_\ell, \boldsymbol{x}_{\ell+1}\}$ related by the approximate linear mapping from (1), $y_i^\mu = \langle \boldsymbol{x}_i^\mu, \boldsymbol{w}^\mu \rangle + \epsilon_i^\mu$, where now $\boldsymbol{w}^\mu$ is the task vector corresponding to the $\mu$-th sample context.

We will sample the pretraining batch as follows: For $i = 1, \ldots, \ell$ and $\mu = 1, \ldots, n$,

$$
\begin{aligned}
\boldsymbol{x}_i^\mu &\sim_{\text{i.i.d.}} \mathcal{N}(0, I_d/d), & \epsilon_i^\mu &\sim_{\text{i.i.d.}} \mathcal{N}(0, \rho), \\
\boldsymbol{w}^\mu &\sim_{\text{unif}} \{\boldsymbol{t}_1, \cdots, \boldsymbol{t}_k\} & \text{where } \boldsymbol{t}_j &\sim_{\text{i.i.d.}} \mathcal{N}(0, C_{\text{train}}) \text{ for } j = 1, \ldots, k.
\end{aligned}
\tag{2}
$$

The parameter $k$ here is called *task diversity*. Note that if $k < n$, the pretraining batch contains some tasks repeated across the contexts. In this way, we control both the amount $k$ of actually unique tasks seen during pretraining, as well as the structure of the task distribution using $C_{\text{train}}$. This distinguishes our setup from previous studies which typically focus on isotropic or matched tasks (Fu et al., 2023; Li et al., 2024; Lu et al., 2025; Raventós et al., 2023).

**Linear attention.** We will study the performance of the linear self-attention block (Wang et al., 2020) on this in-context regression task. The input to the linear self-attention model is an embedding matrix $Z$ made up of our context sequence. Here, following the convention of Wang et al. (2020); Wu et al. (2024); Zhang et al. (2024a), we chose to embed $\{\boldsymbol{x}_1, y_1, \boldsymbol{x}_2, y_2, \ldots, \boldsymbol{x}_\ell, y_\ell, \boldsymbol{x}_{\ell+1}\}$ as

$$
Z = \begin{bmatrix} \boldsymbol{x}_1 & \boldsymbol{x}_2 & \ldots & \boldsymbol{x}_\ell & \boldsymbol{x}_{\ell+1} \\ y_1 & y_2 & \ldots & y_\ell & 0 \end{bmatrix} \in \mathbb{R}^{(d+1) \times (\ell+1)}, \tag{3}
$$

where 0 in the lower-right corner is a placeholder token for the $y_{\ell+1}$ we wish to predict. The model's output (Katharopoulos et al., 2020; Shen et al., 2021; Wang et al., 2020) is given by

$$A = Z + VZ(KZ)^\top (QZ)/\ell \tag{4}$$

for value matrix $V \in \mathbb{R}^{(d+1) \times (d+1)}$ and key, query matrices $K, Q$ such that $K^\top Q \in \mathbb{R}^{(d+1) \times (d+1)}$. Following the positional encoding in (3), the linear attention model's prediction for $y_{\ell+1}$ is

$$\hat{y} = A_{d+1, \ell+1}. \tag{5}$$

Previous works from Zhang et al. (2024a) and Lu et al. (2025) have shown that the output $\hat{y} = A_{d+1, \ell+1}$ of the model can be reduced to give a simpler, analytically-tractable model. Writing the attention and value matrices as

$$
V = \begin{bmatrix} V_{11} & \boldsymbol{v}_{12} \\ \boldsymbol{v}_{21}^\top & v_{22} \end{bmatrix}, \quad M = \begin{bmatrix} M_{11} & \boldsymbol{m}_{12} \\ \boldsymbol{m}_{21}^\top & m_{22} \end{bmatrix} \equiv K^\top Q, \tag{6}
$$

the predictor expands as

$$\hat{y} = \frac{1}{\ell} \boldsymbol{x}_{\ell+1}^\top \left( v_{22} M_{11}^\top \sum_{i=1}^{\ell} y_i \boldsymbol{x}_i + v_{22} \boldsymbol{m}_{21} \sum_{i=1}^{\ell} y_i^2 + M_{11}^\top \sum_{i=1}^{\ell+1} \boldsymbol{x}_i \boldsymbol{x}_i^\top \boldsymbol{v}_{21} + \boldsymbol{m}_{21} \sum_{i=1}^{\ell} y_i \boldsymbol{x}_i^\top \boldsymbol{v}_{21} \right). \quad (7)$$

The last two terms in (7) depend on $\boldsymbol{v}_{21}$. While these terms in principle could contribute to the predictor, prior work argues they may be relatively weak signals for estimating $\boldsymbol{w}$ in high dimensions: $\boldsymbol{x}_i \boldsymbol{x}_i^\top \boldsymbol{v}_{21}$ carries no task information, and $y_i \boldsymbol{x}_i^\top \boldsymbol{v}_{21}$ provides only a one-dimensional projection of $\boldsymbol{x}$ and $\boldsymbol{w}$ (Lu et al., 2025). For analytical tractability, we therefore make the *Ansatz* $\boldsymbol{v}_{21} = \boldsymbol{0}$, which yields a reduced model that we analyze throughout. We justify this choice in Appendix H comparing the parameter-reduced model analyzed below with full-parameter trained linear attention.

With this simplification, we can rewrite the simplified model's output as

$$\hat{y}_{\ell+1} = \mathrm{tr}(\Gamma H_Z^\top) \quad (8)$$

for a parameter matrix

$$\Gamma \equiv v_{22} \left[ M_{11}^\top/d \quad \boldsymbol{m}_{21} \right] \in \mathbb{R}^{d \times (d+1)}. \quad (9)$$

and a data matrix

$$H_Z \equiv \boldsymbol{x}_{\ell+1} \left[ \frac{d}{\ell} \sum_{i \le \ell} y_i \boldsymbol{x}_i^\top \quad \frac{1}{\ell} \sum_{i \le \ell} y_i^2 \right] \in \mathbb{R}^{d \times (d+1)}. \quad (10)$$

**Reduced model optimization.** Given a batch $\{\boldsymbol{x}_1^\mu, y_1^\mu, \ldots, \boldsymbol{x}_{\ell+1}^\mu, y_{\ell+1}^\mu\}_{\mu=1}^n$ of pretraining data (as explained above), we can find finite-sample optimal parameters by minimizing MSE loss on next-output prediction with ridge regularization, giving

$$\Gamma^* = \arg\min_\Gamma \sum_{\mu=1}^n \left( y_{\ell+1}^\mu - \mathrm{tr}(\Gamma (H^\mu)^\top) \right)^2 + \frac{n}{d} \lambda \, \mathrm{tr}(\Gamma \Gamma^\top), \quad (11)$$

where $\lambda > 0$ is a regularization parameter, and $H^\mu$ is defined by (10) for the $\mu$th context. We will focus on the minimum-norm predictor, *i.e.*, on the limit where $\lambda \to 0$ (Hastie et al., 2022).

**Test error.** We finally wish to test the pretrained model on a general task to see if the model can genuinely perform in-context regression. The test distribution $\mathcal{P}_{\text{test}}$ is then

$$\boldsymbol{x}_i^{\text{test}} \sim_{\text{i.i.d.}} \mathcal{N}(0, I_d/d), \qquad \epsilon_i^{\text{test}} \sim_{\text{i.i.d.}} \mathcal{N}(0, \rho), \qquad \boldsymbol{w}^{\text{test}} \sim_{\text{i.i.d.}} \mathcal{N}(0, C_{\text{test}}). \quad (12)$$

We will measure ICL performance by the average MSE error of our optimal estimator $\hat{y}_{\ell+1}^* = \mathrm{tr}(\Gamma^* H^\top)$ over the test distribution,

$$\mathcal{E}_{\text{ICL}}(\Gamma^*) = \mathbb{E}_{\mathcal{P}_{\text{test}}}[(y_{\ell+1} - \mathrm{tr}(\Gamma^* H^\top))^2]. \quad (13)$$

We highlight here the generality of the test task distribution through the new matrix $C_{\text{test}}$, allowing us to study the interaction of training and testing task structure.

**Data parameters in the high-dimensional limit.** We have introduced four data parameters: token dimension $d$, context length $\ell$, pretraining batch size $n$, and task diversity $k$. As is standard in the theory of high-dimensional regression (Advani et al., 2020; Atanasov et al., 2024; Hastie et al., 2022; Lu et al., 2025), we will consider a scaling limit where all four of these parameters are taken to infinity in a way such that the following ratios remain constant:

$$\frac{\ell}{d} \equiv \alpha, \qquad \frac{n}{d^2} \equiv \tau, \qquad \text{and} \qquad \frac{k}{d} \equiv \kappa. \quad (14)$$

Considering this limit makes the model analytically tractable, but preserves interesting phenomena that are present at finite size. Going forward, we will refer to $\alpha$, $\tau$, and $\kappa$ as the context length, batch size, and task diversity parameters, respectively.

**Pretraining task quantities.** Before presenting our formula for this ICL test error, it will be helpful to first define some task-distribution quantities, which depend on the pretraining task covariance $C_{\text{train}}$ and task diversity parameter $\kappa$. These quantities effectively tell us how well we can reconstruct $C_{\text{train}}$ from the $k$-sample task covariance that the model sees during pretraining,

$$R_k = \frac{1}{k} \sum_{j \in [k]} \boldsymbol{t}_j \boldsymbol{t}_j^\top . \tag{15}$$

Because the pretraining tasks $\{\boldsymbol{t}_1, ..., \boldsymbol{t}_k\}$ are random, $R_k$ is a random matrix. However, in high dimensions, the following *deterministic* quantities capture the relevant information about $R_k$: let the deterministic matrix $F_\kappa(z)$ and deterministic scalar $\mathcal{M}_\kappa(z)$ be defined through the implicit equations

$$F_\kappa(z) = \left( \left( 1 - \frac{1}{\kappa} + \frac{z}{\kappa} \mathcal{M}_\kappa(z) \right) C_{\text{train}} + z I_d \right)^{-1} \tag{16}$$

$$\mathcal{M}_\kappa(z) = \frac{1}{d} \operatorname{tr}(F_\kappa(z)) . \tag{17}$$

Then, we have the high-dimensional equivalence $(R_k + z I_d)^{-1} \simeq F_\kappa(z)$.

As we define formally in Definition 1 of Appendix B, this equivalence holds in the sense that the (normalized) traces of $(R_k + z I_d)^{-1}$ and $F_\kappa(z)$ multiplied against test matrices coincide in the high-dimensional limit. Here, $z \in \mathbb{R}_+$ is a noise threshold parameter suppressing smaller eigenvalues of $R_k$.

Intuitively, $F_\kappa(z)$ and $\mathcal{M}_\kappa(z)$ give us information about how much signal in $C_{\text{train}}$ can be recovered after a finite number $k$ of pretraining samples, filtered by noise level $z$. As $\kappa \to \infty$, $R_k$ approaches $C_{\text{train}}$, and we therefore fully recover the original distribution of tasks. We will use $F_\kappa'(z)$ to refer to the derivative of $F_\kappa(z)$ with respect to $z$; this gives a measure of the sensitivity of this covariance recovery matrix to the noise level $z$. These quantities will play a key role in our discussion of task alignment.

Given this setup, we compute a formula for the ICL generalization error $\mathcal{E}_{\text{ICL}}(\Gamma^*)$ in terms of the data parameters $\alpha, \kappa, \tau$, the pretraining task covariance $C_{\text{train}}$, and the testing task covariance $C_{\text{test}}$. We present this formula and discuss its implications in the following sections.

**Notation.** We write $\operatorname{tr}[A] \equiv \operatorname{tr}(A)/d$ for the dimension-scaled trace of matrix $A$. We use a normalized Frobenius inner product between two matrices, *i.e.* $\langle A, B \rangle \equiv \operatorname{tr}[AB^\top] = \operatorname{tr}(AB^\top)/d$. We abbreviate the normalized traces of the task covariances as $c_{\text{train}} = \operatorname{tr}[C_{\text{train}}]$ and $c_{\text{test}} = \operatorname{tr}[C_{\text{test}}]$. We denote high-dimensional equivalence (as in Appendix Definition 1) by $\simeq$.

## 3 ICL TEST ERROR DEPENDS ON TASK MISALIGNMENT

### 3.1 TASK ALIGNMENT DETERMINES GENERALIZATION IN THE SOLVABLE MODEL

We state the main result of our analysis of the simplified linear attention model, which is an analytical formula for the ICL error (13) in high dimensions. Here we give only an informal statement of this result and focus on its implications; the formal statement is given by (C.7) in Appendix C.

> **Result:** High-dimensional formula for the ICL error $\mathcal{E}(\Gamma^*)$
>
> The ICL test error (13) of the simplified linear attention model set up above is given by
>
> $$\mathcal{E}_{\text{ICL}}(\Gamma^*) \simeq e_{\text{scalar}}(\boldsymbol{\lambda}_{\text{train}}, c_{\text{test}}) + e_{\text{misalign}}(C_{\text{train}}, C_{\text{test}}) \equiv e_{\text{ICL}}(C_{\text{train}}, C_{\text{test}}) \tag{18}$$
>
> in the high-dimensional limit, where
>
> $$e_{\text{misalign}}(C_{\text{train}}, C_{\text{test}}) = \langle C_{\text{test}}, \ \mathcal{K} \rangle \tag{19}$$
>
> measures the alignment between $C_{\text{test}}$ and
>
> $$\mathcal{K} \equiv q F_\kappa(\sigma) + (q\tilde{\lambda} - \sigma^2) F_\kappa'(\sigma) . \tag{20}$$

Appearing in this formula are an effective ridge variable $\tilde{\lambda}$ and effective noise variable $\sigma$ given by the solution of the joint equations

$$\tilde{\lambda}\mathcal{M}_{\kappa}(\sigma) = 1 - \tau \quad \text{for } \tau < 1, \qquad \tilde{\lambda} = 0 \quad \text{for } \tau > 1 \tag{21}$$

$$\sigma = (\rho + c_{\text{train}})/\alpha + \tilde{\lambda}. \tag{22}$$

where $\mathcal{M}_{\kappa}(\cdot)$ is determined self-consistently as in (17). Finally $q$ is a pretraining-error term given by (C.8) and $e_{\text{scalar}}(\boldsymbol{\lambda}_{\text{train}}, c_{\text{test}})$ is given by (C.10); note that both $q$ and $e_{\text{scalar}}$ only depend on $C_{\text{test}}$ through its trace $c_{\text{test}}$, and $e_{\text{scalar}}$ only depends on $C_{\text{train}}$ through its spectrum $\boldsymbol{\lambda}_{\text{train}}$.

We first support this result through Figure 1, which shows agreement between our theory curve formula $e_{\text{ICL}}(C_{\text{train}}, C_{\text{test}})$ and numerical simulations of MSE error $\mathcal{E}_{\text{ICL}}(\Gamma^*)$ given by (11) and the test distribution (12). By comparing the left ($e_{\text{ICL}}$) and right ($e_{\text{misalign}}$) panels of Figure 1 we see that the behavior of $e_{\text{ICL}}$ is highly dependent on the $e_{\text{misalign}}$ term: for well-aligned training and testing distributions, low and decreasing $e_{\text{misalign}}$ in $\kappa$ immediately leads to a monotonic decrease in $e_{\text{ICL}}$ in $\kappa$; for worse-aligned training and testing distributions, $e_{\text{ICL}}$ can be nonmonotonic or even monotonically increasing in $\kappa$. This is intriguing, as one would naïvely expect additional task samples to improve in-context performance, but this is not true in general: whether additional task samples are helpful for ICL depends on the alignment of the pretraining and testing distributions.

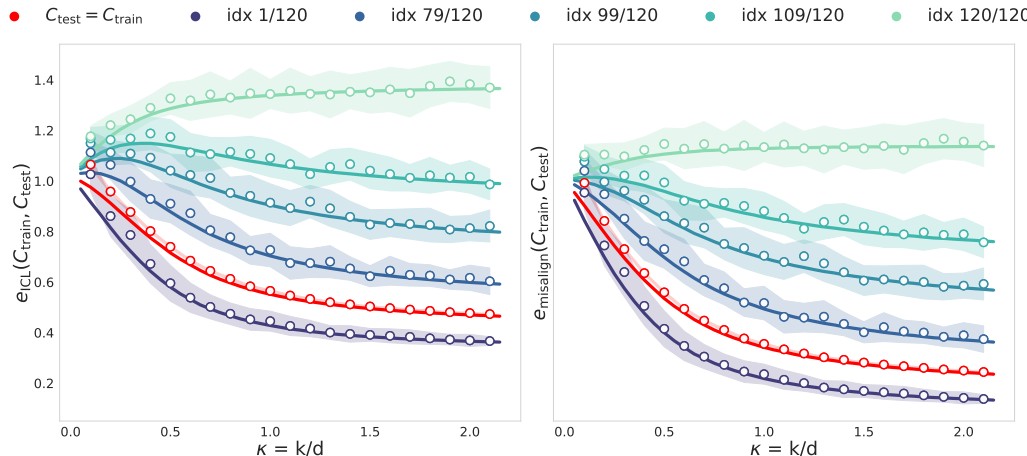

Figure 1: Theoretical $e_{\text{ICL}}$ (left panel) and $e_{\text{misalign}}$ (right panel) curves plotted against numerical simulations of $\mathcal{E}(\Gamma^*)$ computed directly from sampled data. We choose $C_{\text{train}}$ with uniform eigenvalue distribution: $C_{\text{train}} \propto \text{diag}([d, d-1, \cdots, 1])$ such that $\text{tr}(C_{\text{train}}) = d$. We compare $C_{\text{test}} = C_{\text{train}}$ (red curves) with testing on single task directions, *i.e.*, the "idx i/d" labels correspond to rank-1 test covariances $C_{\text{test}}^i = d e_i e_i^\top$ spiked at index $i$. In this way, $C_{\text{test}}^1$ captures the strongest task direction of $C_{\text{train}}$ and $C_{\text{test}}^d$ captures the weakest. *Parameters:* $d = 120$, $\alpha = 2$, $\tau = 4$, $\rho = 0.01$. Shading represents $\pm$std of numerical simulations. We calculate the simulation values of $e_{\text{misalign}}$ in the right panel by subtracting $e_{\text{scalar}}$ from the MSE simulation values $\mathcal{E}(\Gamma^*)$.

To gain intuition for this misalignment error (19), a useful analogy is to consider the simplest matrix "misalignment" measure $\langle C_{\text{test}}C_{\text{train}}^{-1}\rangle$. Firstly, this measure captures misalignment that can occur from misalignment of the *eigenvectors* of $C_{\text{test}}$ and $C_{\text{train}}$; see Corollary 4.1 and its proof for further discussion. Going beyond eigenvectors, because the eigenvalues of $C_{\text{train}}^{-1}$ are ordered opposite to the eigenvalues of $C_{\text{train}}$, this measure effectively captures the relative strength of signal directions between $C_{\text{test}}$ and $C_{\text{train}}$: if $C_{\text{train}}$ and $C_{\text{test}}$ share the same eigenvalues, alignment will be maximized when these eigenvalues appear in the same order, and minimized when these eigenvalues appear in opposing orders. This is precisely why $\langle C_{\text{test}}C_{\text{train}}^{-1}\rangle$ can be interpreted as a misalignment measure, as even this very simple *Ansatz* shows how misalignment arises from mismatches in the relative weighting of signal directions between train and test tasks. This relative strength argument holds for our alignment measure, where instead of $\langle C_{\text{test}}C_{\text{train}}^{-1}\rangle$ we use $\langle C_{\text{test}}\mathcal{K}\rangle$. Here $\mathcal{K}$ obviously depends on $C_{\text{train}}$—they share the same eigenvectors—and importantly, has the same property as $C_{\text{train}}^{-1}$ that its eigenvalues are oppositely ordered to the eigenvalues of $C_{\text{train}}$ (we prove this claim in Appendix D for $\tau > 1$ and study it numerically for $\tau < 1$; we conjecture that it holds for all $\tau$).

However, this eigenvalue-ordering argument is incomplete for the ICL setting. It assumes that $C_{\text{train}}$ can be fully learned, but with finite context length, finite task diversity, and label noise, this cannot be the case. Thus, the model cannot access the full covariance structure, only a partial version, whose resolution depends on both sampling and noise. This is precisely what the resolvent terms $F_\kappa(\cdot)$ and $F'_\kappa(\cdot)$ in $\mathcal{K}$ can capture, as explained in the setup. This is analogous to the alignment measures that emerge in ordinary ridge regression under covariate shift, which capture how much of the training feature covariance can be resolved from finitely many samples (Atanasov et al., 2025; Canatar et al., 2021b; Patil et al., 2024; Tripuraneni et al., 2021). Furthermore, the effective noise $\sigma$ must play a key role in alignment, as the model does not know *a priori* that this is an "in-context" learning problem, and must learn to decouple the tokens from the task for each context in order to extract the task information implicit in that context. The ability of the model to do this depends on both the label noise ($\rho$), the context length ($\alpha$), and a sufficient number of contexts ($\tilde\lambda$). In fact, the form of $\sigma = (\rho + \text{tr}[C_{\text{train}}])/\alpha + \tilde\lambda$ is also familiar from ordinary ridge regression as an effective noise-to-signal term: the optimal ridge regularization parameter balances the variance due to label noise ($\rho$) with the estimation error from having finite data (Atanasov et al., 2024; Canatar et al., 2021a; Hastie et al., 2022; Patil et al., 2024). At infinite sample size, the optimal ridge is simply $\rho$, the label noise. However, at finite sample size, the effective regularization is increased due to the finite-sample estimation error, and becomes $(\rho + \sigma_w^2)/\alpha$, where $\sigma_w^2$ characterizes variability or complexity of the regression task. In the same way, our model has to resolve the statistics of the tokens $x$ over samples ($\ell$, measured by $\alpha$), and so we expect terms familiar from linear regression to appear in our formula.

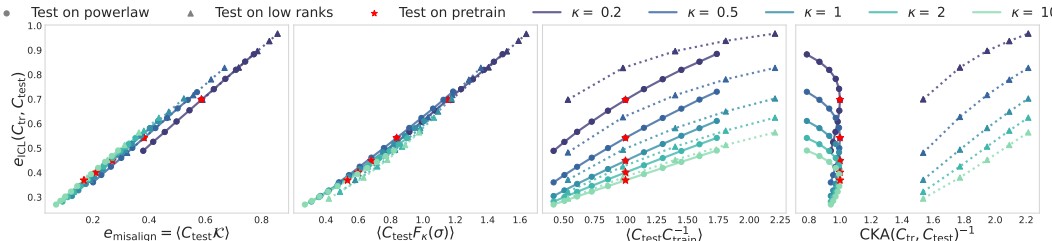

Figure 2: $e_{\text{ICL}}(C_{\text{train}}, C_{\text{test}})$ against alignment measures: $e_{\text{misalign}}(C_{\text{train}}, C_{\text{test}})$, $\text{tr}[C_{\text{test}}F]$, $\text{tr}[C_{\text{test}}C_{\text{train}}^{-1}]$, and $1/\text{CKA}(C_{\text{train}}, C_{\text{test}})$ from left to right. $C_{\text{train}}$ is fixed to be a diagonal matrix with powerlaw spectrum $C_{\text{train}} \propto \text{diag}([1^{-p}, ..., d^{-p}]$ with power $p = 0.9$ and $\text{tr}[C_{\text{train}}] = 1$. $C_{\text{test}}$ is varied over a range of different covariance matrices that are simultaneously diagonalizable with $C_{\text{train}}$, specifically power spectrum with different powers (circles connected by solid line), and low-rank covariance matrices $C_r = \text{diag}[(d/r)\mathbf{1}_r, \mathbf{0}_{d-r}]$ (triangular markers connected by dashed line). Changing the power of the powerlaw tests or the rank of the low-rank tests will make them either more or less aligned with $C_{\text{train}}$. *Parameters:* $d = 120, \alpha = 2, \tau = 4, \rho = 0.01$.

In summary, our alignment measure is motivated both by arguments regarding spectral ordering and relative strength, as well as having all components necessary to capture finite size effects. In Figure 2 we compare our alignment measure to three others: the population matrix measure $\langle C_{\text{train}}C_{\text{test}}^{-1}\rangle$, a simpler version of our alignment measure involving just the resolvent $\langle C_{\text{train}}F_\kappa(\sigma)\rangle$, and finally the Centered Kernel Alignment (CKA) measure (Kornblith et al., 2019). This comparison illustrates how different measures emphasize different aspects: $\langle C_{\text{train}}F_\kappa(\sigma)\rangle$ accounts for finite-sample resolution effects but not as specifically as $\langle C_{\text{train}}\mathcal{K}\rangle$ does, while $\langle C_{\text{train}}C_{\text{test}}^{-1}\rangle$ and CKA both miss finite-sample effects altogether, CKA instead being designed to detect nonlinear representational similarity. We perform this comparison by first noting that, trivially, $e_{\text{misalign}}$ is monotonically related to $e_{\text{ICL}}$: larger $e_{\text{misalign}}$ implies larger $e_{\text{ICL}}$. Figure 2 shows $e_{\text{ICL}}$ for fixed $C_{\text{train}}$ and varying $C_{\text{test}}$ plotted against the above matrix alignment measures between $C_{\text{train}}$ and $C_{\text{test}}$. We see that $\langle C_{\text{train}}\mathcal{K}\rangle$ and $\langle C_{\text{train}}F_\kappa(\sigma)\rangle$ are the strongest drivers of $e_{\text{ICL}}$ (*i.e.* most monotonically related, with $\langle C_{\text{train}}\mathcal{K}\rangle$ obviously being perfectly correlated), while the performance of $\langle C_{\text{test}}C_{\text{train}}^{-1}\rangle$ and CKA as predictors of $e_{\text{ICL}}$ is lacking.

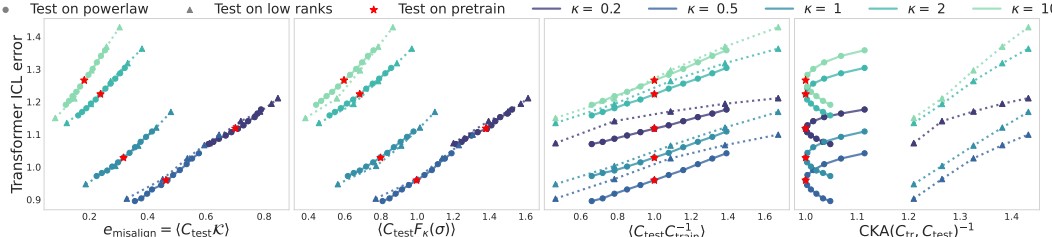

Figure 3: ICL test loss of a nonlinear transformer against different alignment measures. The setup of the covariances is identical to Figure 2, the only difference is that here ICL error is computed as the MSE on the test task as performed by a trained two-layer architecture with softmax attention and MLP connections. Our measure $e_{\mathrm{misalign}}$ achieves the best correlation with ICL error: the Spearman coefficients (measuring monotonicity, over all test covariances and averaged over the different $\kappa$ values) are **0.99 (ours)**, 0.98, 0.96, and 0.39 from left to right. *Parameters:* $d = 20$, $\alpha = 2$, $\tau = 4$, $\rho = 0.01$.

## 3.2 TASK ALIGNMENT PREDICTS GENERALIZATION IN NONLINEAR TRANSFORMERS

We further support the predictive power of our alignment measure defined by $e_{\mathrm{misalign}}$ for ICL error in a two-layer transformer architecture with softmax attention by Figure 3. See Appendix H for details on the architecture setup. We see that even for an architecture far from the linear attention considered by the theory, our theoretically-derived alignment measure is still the best predictor of how changing the test distribution affects ICL error, compared to other alignment measures.

## 4 MISMATCHED TASK DISTRIBUTIONS ARE OFTEN OPTIMAL

We have identified the relevant measure of alignment between the pretraining and testing task distributions for linear transformers learning to perform ICL. It is then natural to ask whether for a fixed test distribution $C_{\mathrm{test}}$, is it always optimal to pretrain on the test distribution, *i.e.*, is $e_{\mathrm{ICL}}(C_{\mathrm{test}}, C_{\mathrm{test}}) \leq e_{\mathrm{ICL}}(C_{\mathrm{train}}, C_{\mathrm{test}})$? In this section, we show that the answer to this question is in general no, it is *not* always optimal to pretrain on the test distribution.

### 4.1 COVARIANCE ALIGNMENT AND EIGENSPACE STRUCTURE

We first show that it suffices to consider the case in which $C_{\mathrm{train}}$ and $C_{\mathrm{test}}$ are co-diagonalizable, as the ICL error is extremal in this case. Note that we can focus on $e_{\mathrm{misalign}}$, as other terms in the ICL error (18) contained in $e_{\mathrm{scalar}}$ (C.10) are independent of the eigenvectors of these two matrices.

> **Corollary 4.1:** Eigenspace alignment extremizes misalignment error
>
> The misalignment error $e_{\mathrm{misalign}}(C_{\mathrm{train}}, C_{\mathrm{test}}) = \langle C_{\mathrm{test}}, \mathcal{K} \rangle$ is extremized when $C_{\mathrm{test}}$ and $C_{\mathrm{train}}$ are co-diagonalizable. Concretely, letting $\lambda_1(C_{\mathrm{test}}) \geq \cdots \geq \lambda_d(C_{\mathrm{test}})$ and $\lambda_1(\mathcal{K}) \geq \cdots \geq \lambda_d(\mathcal{K})$ be the ordered eigenvalues of these two real symmetric matrices, we have
>
> $$\frac{1}{d}\sum_{j=1}^{d}\lambda_j(C_{\mathrm{test}})\lambda_{d-j+1}(\mathcal{K}) \leq e_{\mathrm{misalign}} \leq \frac{1}{d}\sum_{j=1}^{d}\lambda_j(C_{\mathrm{test}})\lambda_j(\mathcal{K}), \qquad (23)$$
>
> with equality in either the upper or lower bound if and only if $C_{\mathrm{train}}$ and $C_{\mathrm{test}}$ are co-diagonalizable.

To show that this result holds, we observe that, as $C_{\mathrm{test}}$ and $\mathcal{K}$ are real symmetric matrices, the desired bound (23) is simply a restatement of Ruhe's trace inequality normalized by $1/d$, where equality holds if and only if $C_{\mathrm{test}}$ and $\mathcal{K}$ are co-diagonalizable. We give a self-contained proof of this inequality in Appendix E (see also Marshall et al. (2010) or Li (2020)'s expository blog post). By its definition (20), $\mathcal{K}$ and $C_{\mathrm{train}}$ have the same eigenvectors, so the claim follows.

Therefore, the misalignment error can be minimized *or* maximized if we assume that $C_{\text{train}}$ and $C_{\text{test}}$ are simultaneously diagonalizable. We therefore restrict our attention to the co-diagonalizable setting, and write the ordered eigenvalues of $C_{\text{train}}$ and $C_{\text{test}}$ as $\lambda_1(C_{\text{train}}) \geq \cdots \geq \lambda_d(C_{\text{train}})$ and $\lambda_1(C_{\text{test}}) \geq \cdots \geq \lambda_d(C_{\text{test}})$, respectively.

## 4.2 Optimal Test Covariance for Fixed Pretraining Distribution

Before presenting results on optimal pretraining structure for fixed task structure, which is a more practical question to answer, we begin first with a simpler question: For fixed training distribution $C_{\text{train}}$, is it always optimal to test on the pretraining distribution, *i.e.*, is $e_{\text{ICL}}(C_{\text{train}}, C_{\text{train}}) \leq e_{\text{ICL}}(C_{\text{train}}, C_{\text{test}})$? The following result answers this question in the negative:

> **Corollary 4.2:** Trace-constrained optimal test covariance
>
> For fixed $C_{\text{train}}$ and fixed $c_{\text{train}} = c_{\text{test}}$, and $\tau > 1$, we have that $e_{\text{ICL}}(C_{\text{train}}, C_{\text{test}})$ is minimized by the single-index spike covariance with eigenvalues
>
> $$[\lambda_1(C_{\text{test}}) \quad \cdots \quad \lambda_i(C_{\text{test}}) \quad \cdots \quad \lambda_d(C_{\text{test}})] = [dc_{\text{train}} \quad \cdots \quad 0 \quad \cdots \quad 0]$$
>
> *i.e.*, all test signal is aligned with the largest eigenvector of $C_{\text{train}}$.

To see that this holds, note that $e_{\text{ICL}}(C_{\text{train}}, C_{\text{test}})$ is a linear function of the eigenvalues of $C_{\text{test}}$ as $e_{\text{misalign}}(C_{\text{train}}, C_{\text{test}})$ is linear in $C_{\text{test}}$ and $e_{\text{scalar}}(\boldsymbol{\lambda}_{\text{train}}, c_{\text{test}})$ does not depend on $C_{\text{test}}$ when $c_{\text{test}} = c_{\text{train}}$. Furthermore, the constraint $c_{\text{test}} = c_{\text{train}}$ is a simplex in the space of $C_{\text{test}}$ eigenvalues. The minimum will thus be attained along the simplex, at the vertex corresponding to the lowest eigenvalue of $\mathcal{K}$. As argued in Appendix D, this corresponds to the largest eigenvalue of $C_{\text{train}}$ when $\tau > 1$. As further discussed in Appendix D, we conjecture this to be true for all $\tau$.

This result illustrates a bias towards exploiting low-dimensional structure seen in pretraining to optimize performance at test time. In these cases, where we are optimizing test structure $C_{\text{test}}$ for a fixed pretraining structure $C_{\text{train}}$, we observe that the best generalization is achieved by concentrating all signal into a single shared direction, collapsing the task manifold into a single dimension. In other words, it is easier to learn to generalize over a degenerate low-rank test structure highly aligned to the pretraining structure than to generalize over the entire pretraining structure learned from limited samples. This is familiar from spectral bias results of ordinary ridge regression (Canatar et al., 2021a;b). Note that, had our training distribution been isotropic ($C_{\text{train}} = I_d$) then all trace-fixed test covariances will perform equally; utilizing anisotropy is crucial.

## 4.3 Non-Optimality of Pretraining on Test Structure

We now investigate the original question of the optimal pretraining task covariance $C_{\text{train}}$, given a particular test task covariance $C_{\text{test}}$. We will provide an example showing it is in fact *not optimal* in general to have $C_{\text{train}} = C_{\text{test}}$. We particularly emphasize that whether or not a particular task structure is better for pretraining depends strongly on the task diversity $\kappa$.

Consider the case of power-law task distributions at both training and test time, *i.e.*,

$$C_{\text{train}} \propto \text{diag} \begin{bmatrix} 1^{-p_{\text{train}}} & \cdots & d^{-p_{\text{train}}} \end{bmatrix}, \qquad C_{\text{test}} \propto \text{diag} \begin{bmatrix} 1^{-p_{\text{test}}} & \cdots & d^{-p_{\text{test}}} \end{bmatrix}, \tag{24}$$

with the normalization constant chosen such that $\text{tr}[C_{\text{train}}] = \text{tr}[C_{\text{test}}]$. We can study the effect of changing $C_{\text{train}}$ relative to $C_{\text{test}}$ on $e_{\text{ICL}}$ by changing the exponent $p_{\text{train}}$ relative to fixed $p_{\text{test}}$.

Figure 4 shows that, for low $\kappa$, pretraining on $C_{\text{train}}$ with higher spectral power relative to the test power can improve ICL error. We see that focusing pretraining on a low-dimensional subspace, effectively creating a strong inductive bias, can improve in-context learning performance when training data is scarce. The model generalizes better by overfitting to fewer directions, rather than learning more directions weakly. However, increasing the pretraining power too much relative to the test power will worsen ICL performance, as the pretraining task set is now too low-dimensional to cover enough variation at test time. We highlight that the potential advantage coming from increasing training spectral power weakens as soon as $\kappa$ is large enough for the model to be able to resolve enough task directions during pretraining.

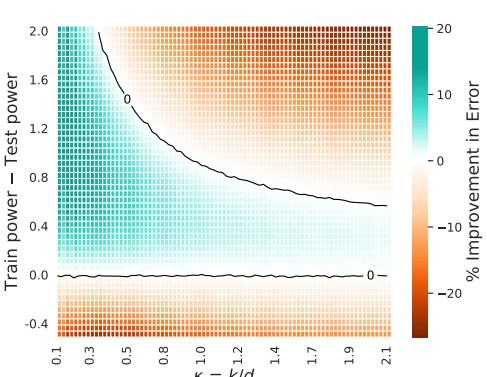

Figure 4: Heatmap of theoretical ICL error given by (18) for simultaneously diagonalizable powerlaw task covariances $C_{\text{train}}$ and $C_{\text{test}}$ with spectral power $p_{\text{train}}$ (variable) and $p_{\text{test}}$ (fixed). The $x$-axis shows task diversity $\kappa$ and the $y$-axis shows the difference $p_{\text{train}} - p_{\text{test}}$ between task spectral powers. The colourbar shows the % improvement in error by training on $C_{\text{train}}$ instead of $C_{\text{test}}$. This shows that increasing spectral power in the pretraining tasks can markedly improve ICL error on the same test distribution. *Parameters:* $d = 100$, $p_{\text{test}} = 0.9$, $\alpha = 1$, $\tau = 4$, $\rho = 0.01$.

## 5 CONCLUSIONS

We have developed a framework of in-context task alignment, showing that our derived measure of *error from task misalignment* is a robust predictor of ICL performance, even in nonlinear architectures. This derivation builds on a model of in-context learning of linear regression, which we extend to *fully general* task covariates. Previous works have shown that the performance of ICL models can suffer under task shift, particularly in cases of low task diversity (Garg et al., 2022; Goddard et al., 2025; Zhang et al., 2024a). We highlight that, while task misalignment is provably a driver of ICL error, our fully-general covariance framework can be used to show that task misalignment can indeed be utilized to *improve* ICL performance in cases of limited data.

Looking forward, our analysis provides a rich model of ICL with many further avenues of investigation. A more detailed analysis of the interaction between pretraining-specific error $e_{\text{scalar}}$ and misalignment error $e_{\text{misalign}}$ could be used to derive a general heuristic for optimal pretraining, with potential implications for practical settings. Beyond task alignment, additional highly relevant phenomena can be investigated: we highlight (1) a generalized learning transition in task diversity (empirically studied by Raventós et al. (2023); see Appendix F) and (2) test-time scaling (see Gozeten et al. (2025) and Appendix G). We leave exploration of these phenomena to future work.

### AUTHOR CONTRIBUTIONS

All authors conceived the project. M.I.L. performed all computations with assistance from J.A.Z.-V., Y.M.L., and C.P. . M.I.L. performed all numerical analysis and experiments, and wrote an initial draft of the paper. J.A.Z.-V., Y.M.L., and C.P. supervised the project and contributed to review and editing of the manuscript.

### ACKNOWLEDGMENTS

M.I.L. is supported by a Graduate Fellowship from the Kempner Institute. J.A.Z.-V. was supported by a Junior Fellowship from the Harvard Society of Fellows. C.P. is supported by an NSF CAREER Award (IIS-2239780), DARPA grants DIAL-FP-038 and AIQ-HR00112520041, the Simons Collaboration on the Physics of Learning and Neural Computation, and the William F. Milton Fund from Harvard University. This work has been made possible in part by a gift from the Chan Zuckerberg Initiative Foundation to establish the Kempner Institute for the Study of Natural and Artificial Intelligence.

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

SUPPLEMENTARY INFORMATION

In the following sections, we provide proofs of the results presented in the main document. We begin in Section A with some preliminary simplifications of the ICL and IDG population risk for general $\Gamma$. Then in Section B we will set up and complete the random matrix theory calculation necessary to justify our main result, equation 18, which will be done in Section C. Section E will cover the proof of the phenomena-based Corollary 4.1. Then, we provide additional analysis of transitions in the behavior of the ICL error as a function of the task diversity $\kappa$ in Section F, and of shifts in the test-time context length in Section G. We conclude with details of our numerical experiments in Section H.

For convenience, we include the following table detailing quantities appearing throughout the main paper as well as this supplementary section.

| Symbol | Meaning | Def'n |
|---|---|---|
| $d$ | Task and token dimension | Sec. 2 |
| $n$ | Number of pretraining contexts | Sec. 2 |
| $\ell, \ell_{\text{test}}$ | Pretraining and testing context length | Sec. 2 |
| $k$ | Pretraining task diversity | Sec. 2 |
| $Z$ or $Z_\mu$ | Embedding choice of a context (context $\mu$) sequence | (3) |
| $\Gamma, H_Z$ | Reduced model parameters and data | (9), (10) |
| $\lambda$ | Ridge parameter | (11) |
| $\Gamma^*$ | Optimised $\Gamma$ given data sampling $H_{Z_1}, ..., H_{Z_n}$ | (11) |
| $\mathcal{E}_{\text{ICL}}(\Gamma)$ | ICL test errors for parameter $\Gamma$ | (13) |
| $C_{\text{train}}$ | Training task covariance (this is finitely sampled by $k$) | (2) |
| $C_{\text{test}}$ | Testing task covariance (no sampling restriction) | (12) |
| $\rho$ | Label noise in both pretraining and testing | (2) |
| $\tau$ | Asymptotic samples parameter between $n$ and $d^2$ | (14) |
| $\alpha$ | Asymptotic context length parameter between $\ell$ and $d$ | (14) |
| $\kappa$ | Asymptotic task diversity parameter between $k$ and $d$ | (14) |
| $R_k$ | Pretraining task sample covariance | (15) |
| $\tilde{\lambda}, \sigma$ | Rescaled ridges and effective noise | (21), (22) |
| $F_\kappa(z)$ | Deterministic equivalent of resolvent of $R_k$ at ridge $z$ | (16) |
| $\mathcal{M}_\kappa(z)$ | Stieltjes transform of $R_k$ at ridge $z$ | (17) |
| $e_{\text{ICL}}(C_{\text{train}}, C_{\text{test}})$ | Deterministic equivalent for $\mathcal{E}_{\text{ICL}}(\Gamma^*)$ | (18) |
| $e_{\text{misalign}}(C_{\text{train}}, C_{\text{test}})$ | Pretrain-Test misalignment term in $e_{\text{ICL}}(C_{\text{train}}, C_{\text{test}})$ | (19) |
| $\mathcal{K}$ | Pretraining-dependent comparision matrix in $e_{\text{misalign}}$ | (20) |

Table 1: Quantities appearing throughout the main document

| Symbol | Meaning | Def'n |
|---|---|---|
| $\Sigma$ | Token covariance matrix (taken to be $I_d$) | Result 1 |
| $\mathcal{E}_{\text{IDG}}(\Gamma)$ | IDG test error for parameter $\Gamma$ | (A.2) |
| $A_{\text{IDG}}, A_{\text{ICL}}$ | Linear response matrix in simplified $\mathcal{E}_{\text{IDG}}, \mathcal{E}_{\text{ICL}}$ | (A.8), (A.9) |
| $B_{\text{IDG}}, B_{\text{ICL}}$ | Quadratic response matrix in simplified $\mathcal{E}_{\text{IDG}}, \mathcal{E}_{\text{ICL}}$ | (A.8), (A.9) |
| $\boldsymbol{b}_k, R_k$ | Pretraining task sample mean and covariance | (A.10) |
| $G$ | Resolvent of feature $H_Z$ covariance used in $\Gamma^*$ | (B.10) |
| $\pi, G(\pi)$ | Parameterisation of resolvent for convenience | (B.11) |
| $G_{\text{ext}}(\pi)$ | Extended parameterised resolvent that contains $\Gamma^*$ | (B.14) |
| $G_{\text{ext}}^{[\mu]}$ | Leave-one-out extended resolvent | (B.19) |
| $\widehat{\mathcal{G}}_e(\pi)$ | Token- and noise- averaged version of $G_{\text{ext}}(\pi)$ | (B.39) |
| $\Gamma_{\text{eq}}^*$ | Deterministic equivalent of $\Gamma^*$ | (B.52) |
| $e_{\text{IDG}}(\boldsymbol{\lambda}_{\text{train}})$ | Deterministic equivalent of $\mathcal{E}_{\text{IDG}}(\Gamma^*)$ | (C.6) |
| $e_{\text{ICL}}(C_{\text{train}}, C_{\text{test}})$ | Full explicit formula for (18) | (C.7) |
| $e_{\text{scalar}}(C_{\text{train}}, c_{\text{test}})$ | $C_{\text{test}}$-isotropic term in $e_{\text{ICL}}(C_{\text{train}}, C_{\text{test}})$ | (C.10) |

Table 2: Quantities appearing only in the supplementary information, particularly Section B

**Notation** Here $\text{vec}(\cdot)$ will mean the vectorization operation under the *row-major* convention: for a $d_1 \times d_2$ matrix $A$, $\text{vec}(A)$ is a vector in $\mathbb{R}^{d_1 d_2}$, formed by stacking the rows of $A$ together. We will use this together with the matrix Kronecker product $\otimes$, where by standard results we have

$$\text{vec}(uv^\top) = u \otimes v \tag{25}$$

$$(u \otimes v)(w \otimes s)^\top = (uw^\top) \otimes (vs^\top). \tag{26}$$

We will also use the notation $[M]_{\backslash 0}$ to mean the principal minor of a matrix $M$ (*i.e.*, first row and column removed). We use the following normalized trace

$$\text{tr}[A] \equiv \frac{1}{d}\text{tr}(A).$$

We use $\equiv$ when defining new quantities. We use $\simeq$ for *deterministic equivalence*, which describes the large-dimensional limit of a random quantity (either a scalar or matrix) that concentrates to a deterministic value as dimension approaches infinity.

## A    GENERAL ERROR FORMULAS

Here we will set up the definition of various test errors more generally than in the main document. As this is an in-context learning setup, there are different types of "generalization" that can be studied: generalization over the tokens and generalization over the tasks. We will define two test errors to capture these separately.

In general, we wish to understand the performance and behavior of this estimator $\Gamma^*$, which is pretrained on data from the pretraining distribution $\mathcal{P}_{\text{train}}$, when tested on new data. Namely, we will analyze the average performance of an estimator $\hat{y} = \text{tr}(\Gamma, H_Z)$ as a function of given fixed $\Gamma$ under the MSE loss, which is the natural loss for a regression test. The most general this expression can be is

$$\mathcal{L}_{\text{MSE}}(\Gamma) = \mathbb{E}_{\mathcal{P}_{\text{test}}}\left[(y_{\ell+1} - \text{tr}(\Gamma, H_Z))^2\right] \tag{A.1}$$

for a general test distribution $\mathcal{P}_{\text{test}}$ detailing how to sample tokens $x$, tasks $w$, and noise $\epsilon$ at test time.

We will consider two different testing regimes: the *in-context learning (ICL)* test, where the model sees new tasks $w$, and the *in-distribution generalization (IDG)* (or *in-weights*) test, where the model sees the *exact same tasks* used in pretraining $\{t_1, \cdots, t_k\}$, where each $t_j \sim_{\text{i.i.d.}} \mathcal{N}(0, C_{\text{train}})$. Explicitly, we define these test distributions and corresponding error functions as follows:

$$\mathcal{E}_{\text{IDG}}(\Gamma) = \mathbb{E}_{\mathcal{P}_{\text{IDG}}}\left[(y_{\ell+1} - \text{tr}(\Gamma, H_Z))^2\right] \tag{A.2}$$

$$\mathcal{P}_{\text{IDG}} := x_i^\mu \sim_{\text{i.i.d.}} \mathcal{N}(0, I_d/d), \quad w^\mu \sim_{\text{unif}} \{t_1, \cdots, t_k\}, \quad \epsilon_i^\mu \sim_{\text{i.i.d.}} \mathcal{N}(0, \rho),$$

where $i \in [\ell]$ and $\mu \in [n]$, and

$$\mathcal{E}_{\text{ICL}}(\Gamma) = \mathbb{E}_{\mathcal{P}_{\text{ICL}}}\left[(y_{\ell+1} - \text{tr}(\Gamma, H_Z))^2\right] \tag{A.3}$$

$$\mathcal{P}_{\text{ICL}} := x_i^\mu \sim_{\text{i.i.d.}} \mathcal{N}(0, I_d/d), \quad w^\mu \sim_{\text{i.i.d.}} \mathcal{N}(0, C_{\text{test}}), \quad \epsilon_i^\mu \sim_{\text{i.i.d.}} \mathcal{N}(0, \rho),$$

where $i \in [\ell_{\text{test}}]$ and $\mu \in [n]$.

Notice here that we've introduced two different context lengths: $\ell$ for the IDG distribution, which is the same context length as the pretraining setup $\mathcal{P}_{\text{train}}$ given by (2), and $\ell_{\text{test}}$ which is the context length at testing time. This allows us to later explore the effect of pretraining and testing on sequences of different context lengths.

We assume that both task covariance matrices $C_{\text{train}}$ and $C_{\text{test}}$ are well-behaved in high dimensions, specifically that $\text{tr}[C_{\text{train}}], \text{tr}[C_{\text{test}}] = \Theta(1)$. This ensures the task signals are not over or under amplified as $d \to \infty$.

**Lemma 1.** *Simplified test losses. Consider IDG and ICL test distribution and corresponding error functions as given by (A.2) and (A.3). For fixed parameters $\Gamma \in \mathbb{R}^{d \times (d+1)}$ and data sampled according to test distribution $\mathcal{P}_{\text{IDG}}$ or $\mathcal{P}_{\text{ICL}}$, the corresponding errors $e_{\text{IDG}}$ and $e_{\text{ICL}}$ can be expressed*

*as*

$$\mathcal{E}_{\mathrm{IDG}}(\Gamma) \simeq \rho + \mathrm{tr}[\Sigma R_k] - 2\,\mathrm{tr}[\Gamma A_{\mathrm{IDG}}^\top] + \mathrm{tr}[\Sigma\Gamma B_{\mathrm{IDG}}\Gamma^\top] \tag{A.4}$$

$$\mathcal{E}_{\mathrm{ICL}}(\Gamma) \simeq \rho + \mathrm{tr}[\Sigma C_{\mathrm{test}}] - 2\,\mathrm{tr}[\Gamma A_{\mathrm{ICL}}^\top] + \mathrm{tr}[\Sigma\Gamma B_{\mathrm{ICL}}\Gamma^\top] \tag{A.5}$$

*where*

$$A_{\mathrm{IDG}} = [\Sigma R_k \Sigma \quad (\mathrm{tr}[\Sigma R_k] + \rho)\,\Sigma \boldsymbol{b}_k]\,, \tag{A.6}$$

$$A_{\mathrm{ICL}} = [\Sigma C_{\mathrm{test}}\Sigma \quad 0]\,, \tag{A.7}$$

$$B_{\mathrm{IDG}} = \begin{bmatrix} \Sigma R_k \Sigma + \frac{d}{\ell}\left(\mathrm{tr}[\Sigma R_k] + \rho\right)\Sigma & \left(\mathrm{tr}[\Sigma R_k] + \rho\right)\Sigma \boldsymbol{b}_k \\ \left(\mathrm{tr}[\Sigma R_k] + \rho\right)(\Sigma \boldsymbol{b}_k)^\top & \left(\mathrm{tr}[\Sigma R_k] + \rho\right)^2 \end{bmatrix} \tag{A.8}$$

$$B_{\mathrm{ICL}} = \begin{bmatrix} \Sigma C_{\mathrm{test}}\Sigma + \frac{d}{\ell_{\mathrm{test}}}\left(\mathrm{tr}[\Sigma C_{\mathrm{test}}] + \rho\right)\Sigma & 0 \\ 0^\top & \left(\mathrm{tr}[\Sigma C_{\mathrm{test}}] + \rho\right)^2 \end{bmatrix} \tag{A.9}$$

*for $\boldsymbol{b}_k$ and $R_k$ defined by the pretraining task sample $\{w_1, \dots, w_k\}$ as*

$$\boldsymbol{b}_k = \frac{1}{k}\sum_{j\in[k]} \boldsymbol{t}_j\,, \qquad R_k = \frac{1}{k}\sum_{j\in[k]} \boldsymbol{t}_j \boldsymbol{t}_j^\top\,. \tag{A.10}$$

*The "$\simeq$" in these formulas comes from ignoring terms of order $1/d$ and weaker, i.e. a high-dimensional treatment.*

**Remark 1.** *Note that Result 1 is a more general result than the setting considered in the main paper, as Result 1 allows for general token covariance $\Sigma$. The rest of this work will take $\Sigma = I_d$ for computational tractability.*

**Lemma 2.** *Fix the task vector $w$ that defines $y_i = \boldsymbol{w}^\top \boldsymbol{x}_i + \epsilon_i$ for context $Z$. Denote the conditional expectation over only $\boldsymbol{x}_i, \epsilon_i$, holding $\boldsymbol{w}$ fixed, by $\mathbb{E}_{\boldsymbol{x},\epsilon}$. Then we have the following*

$$\mathbb{E}_{\boldsymbol{x},\epsilon}[y_{\ell+1}] = 0 \tag{A.11}$$

$$\mathbb{E}_{\boldsymbol{x},\epsilon}[H_Z] = 0 \tag{A.12}$$

$$\mathbb{E}_{\boldsymbol{x},\epsilon}[y_{\ell+1}^2] = \mathrm{tr}[\Sigma \boldsymbol{w}\boldsymbol{w}^\top] + \rho \tag{A.13}$$

$$\mathbb{E}_{\boldsymbol{x},\epsilon}[y_{\ell+1} H_Z] = \frac{1}{d}\left[\Sigma \boldsymbol{w}\boldsymbol{w}^\top \Sigma \quad \left(\mathrm{tr}[\Sigma \boldsymbol{w}\boldsymbol{w}^\top] + \rho\right)\Sigma \boldsymbol{w}\right] \tag{A.14}$$

$$\mathbb{E}_{\boldsymbol{x},\epsilon}[\mathrm{vec}(H_Z)\,\mathrm{vec}(H_Z)^\top] \simeq \frac{1}{d}\Sigma \otimes \begin{bmatrix} \Sigma \boldsymbol{w}\boldsymbol{w}^\top \Sigma + \frac{d}{\ell}\left(\mathrm{tr}[\Sigma \boldsymbol{w}\boldsymbol{w}^\top] + \rho\right)\Sigma & \left(\mathrm{tr}[\Sigma \boldsymbol{w}\boldsymbol{w}^\top] + \rho\right)\Sigma \boldsymbol{w} \\ \left(\mathrm{tr}[\Sigma \boldsymbol{w}\boldsymbol{w}^\top] + \rho\right)(\Sigma \boldsymbol{w})^\top & \left(\mathrm{tr}[\Sigma \boldsymbol{w}\boldsymbol{w}^\top] + \rho\right)^2 \end{bmatrix} \tag{A.15}$$

*where recall*

$$H_Z \equiv \boldsymbol{x}_{\ell+1}\left[\frac{d}{\ell}\sum_{i\le\ell} y_i \boldsymbol{x}_i^\top \quad \frac{1}{\ell}\sum_{i\le\ell} y_i^2\right] \in \mathbb{R}^{d\times(d+1)}\,. \tag{A.16}$$

*Proof.* Equation (A.11) follows immediately from the linearity of $y_{\ell+1}$ in $\epsilon$ and $\boldsymbol{x}_i$, which are both mean-0 random variables. Similarly for (A.12), as $H_Z$ is linear in $\boldsymbol{x}_{\ell+1}$.

For the conditional expectation of $y_{\ell+1}^2$ in (A.13), simply expand

$$\mathbb{E}_{\boldsymbol{x},\epsilon}\left[y_{\ell+1}^2\right] = \mathbb{E}_{\boldsymbol{x},\epsilon}\left[(\boldsymbol{w}^\top \boldsymbol{x}_{\ell+1} + \epsilon_{\ell+1})(\boldsymbol{x}_{\ell+1}^\top \boldsymbol{w} + \epsilon_{\ell+1})\right] \tag{A.17}$$

$$= \mathbb{E}_{\boldsymbol{x},\epsilon}\left[\boldsymbol{w}^\top \boldsymbol{x}_{\ell+1} \boldsymbol{x}_{\ell+1}^\top \boldsymbol{w} + \epsilon_{\ell+1}^2\right] \tag{A.18}$$

$$= \frac{1}{d}\boldsymbol{w}^\top \Sigma \boldsymbol{w} + \rho \tag{A.19}$$

$$= \mathrm{tr}[\Sigma \boldsymbol{w}\boldsymbol{w}^\top] + \rho\,. \tag{A.20}$$

For (A.14), have

$$\mathbb{E}_{\boldsymbol{x},\epsilon}\left[\frac{d}{\ell}\sum_{i\leq\ell}y_{\ell+1}y_i\boldsymbol{x}_{\ell+1}\boldsymbol{x}_i^\top\right] = \frac{d}{\ell}\sum_{i\leq\ell}\mathbb{E}_{\boldsymbol{x},\epsilon}\left[\boldsymbol{x}_{\ell+1}(\boldsymbol{x}_{\ell+1}^\top\boldsymbol{w}+\epsilon_{\ell+1})(\boldsymbol{w}^\top\boldsymbol{x}_i+\epsilon_i)\boldsymbol{x}_i^\top\right] \quad \text{(A.21)}$$

$$= \frac{d}{\ell}\sum_{i\leq\ell}\mathbb{E}_{\boldsymbol{x},\epsilon}\left[\boldsymbol{x}_{\ell+1}\boldsymbol{x}_{\ell+1}^\top\boldsymbol{w}\boldsymbol{w}^\top\boldsymbol{x}_i\boldsymbol{x}_i^\top\right] \quad \text{(A.22)}$$

$$= \frac{d}{\ell}\sum_{i\leq\ell}(\Sigma/d)\,\boldsymbol{w}\boldsymbol{w}^\top\,(\Sigma/d) \quad \text{(A.23)}$$

$$= \frac{1}{d}\Sigma\boldsymbol{w}\boldsymbol{w}^\top\Sigma \quad \text{(A.24)}$$

and

$$\mathbb{E}_{\boldsymbol{x},\epsilon}\left[\frac{1}{\ell}\sum_{i\leq\ell}y_{\ell+1}\boldsymbol{x}_{\ell+1}y_i^2\right] = \frac{1}{\ell}\sum_{i\leq\ell}\mathbb{E}_{\boldsymbol{x},\epsilon}\left[\boldsymbol{x}_{\ell+1}(\boldsymbol{x}_{\ell+1}^\top\boldsymbol{w}+\epsilon_{\ell+1})(\boldsymbol{w}^\top\boldsymbol{x}_i+\epsilon_i)(\boldsymbol{x}_i^\top\boldsymbol{w}+\epsilon_i)\right]$$

$$\text{(A.25)}$$

$$= \frac{1}{\ell}\sum_{i\leq\ell}\mathbb{E}_{\boldsymbol{x},\epsilon}\left[\boldsymbol{x}_{\ell+1}\boldsymbol{x}_{\ell+1}^\top\boldsymbol{w}(\boldsymbol{w}^\top\boldsymbol{x}_i\boldsymbol{x}_i^\top\boldsymbol{w}+\epsilon_i\epsilon_i)\right] \quad \text{(A.26)}$$

$$= \frac{1}{\ell}\sum_{i\leq\ell}(\Sigma/d)\boldsymbol{w}(\boldsymbol{w}^\top(\Sigma/d)\boldsymbol{w}+\rho) \quad \text{(A.27)}$$

$$= \frac{1}{d}(\text{tr}[\Sigma\boldsymbol{w}\boldsymbol{w}^\top]+\rho)\Sigma\boldsymbol{w} \quad \text{(A.28)}$$

as required for (A.14).

Finally, for (A.15,) first it will be helpful to note, by Isserlis's theorem / Wick's theorem, that

$$\mathbb{E}_{\boldsymbol{x},\epsilon}\left[\boldsymbol{x}_1\boldsymbol{x}_1^\top\boldsymbol{w}\boldsymbol{w}^\top\boldsymbol{x}_1\boldsymbol{x}_1^\top\right] = \frac{2}{d^2}\Sigma\boldsymbol{w}\boldsymbol{w}^\top\Sigma + \frac{1}{d}\text{tr}[\Sigma\boldsymbol{w}\boldsymbol{w}^\top]\Sigma. \quad \text{(A.29)}$$

It will also be useful to rewrite

$$\text{vec}(H_Z)\,\text{vec}(H_Z)^\top = (\boldsymbol{x}_{\ell+1}\boldsymbol{x}_{\ell+1}^\top) \otimes \left(\begin{bmatrix}\frac{d}{\ell}\sum_{i\leq\ell}y_i\boldsymbol{x}_i \\ \frac{1}{\ell}\sum_{i\leq\ell}y_i^2\end{bmatrix}\begin{bmatrix}\frac{d}{\ell}\sum_{i\leq\ell}y_i\boldsymbol{x}_i^\top & \frac{1}{\ell}\sum_{i\leq\ell}y_i^2\end{bmatrix}\right)$$

by converting between $\text{vec}()$ and matrix Kronecker. The components of this Kronecker product are independent and can therefore be averaged separately. Clearly

$$\mathbb{E}_{\boldsymbol{x},\epsilon}[\boldsymbol{x}_{\ell+1}\boldsymbol{x}_{\ell+1}^\top] = \frac{\Sigma}{d},$$

so we focus on the second matrix in the product. This matrix will have blocks that sum over two $\ell$ sums, leading to both $\Theta(1)$ and $\Theta(1/\ell)$ terms in the final expression. We will ignore terms of order $1/\ell$ as we eventually will only use this formula in a proportional limit of $\ell, d \to \infty$ such that $\ell/d = \Theta(1)$, and thus $1/\ell$ is negligible in this limit.

We proceed block-by-block. Firstly,

$$\mathbb{E}_{\boldsymbol{x},\epsilon}\left[\frac{d^2}{\ell^2}\sum_{i,j\leq\ell}y_i\boldsymbol{x}_iy_j\boldsymbol{x}_j^\top\right] = \frac{d^2}{\ell^2}\sum_i\mathbb{E}_{\boldsymbol{x},\epsilon}\left[\boldsymbol{x}_i(\boldsymbol{x}_i^\top\boldsymbol{w}+\epsilon_i)(\boldsymbol{w}^\top\boldsymbol{x}_i+\epsilon_i)\boldsymbol{x}_i^\top\right]$$

$$+ \frac{d^2}{\ell^2}\sum_{i\neq j}\mathbb{E}_{\boldsymbol{x},\epsilon}\left[\boldsymbol{x}_i(\boldsymbol{x}_i^\top\boldsymbol{w}+\epsilon_i)(\boldsymbol{w}^\top\boldsymbol{x}_j+\epsilon_j)\boldsymbol{x}_j^\top\right] \tag{A.30}$$

$$= \ell\frac{d^2}{\ell^2}\mathbb{E}_{\boldsymbol{x},\epsilon}[\boldsymbol{x}_1\boldsymbol{x}_1^\top\boldsymbol{w}\boldsymbol{w}^\top\boldsymbol{x}_1\boldsymbol{x}_1^\top + \epsilon_!\epsilon_1\boldsymbol{x}_1\boldsymbol{x}_1^\top]$$

$$+ (\ell^2-\ell)\frac{d^2}{\ell^2}\mathbb{E}_{\boldsymbol{x},\epsilon}[\boldsymbol{x}_1\boldsymbol{x}_1^\top\boldsymbol{w}\boldsymbol{w}^\top\boldsymbol{x}_2\boldsymbol{x}_2^\top] \tag{A.31}$$

$$= \ell\frac{d^2}{\ell^2}\left(\frac{1}{d}\operatorname{tr}[\Sigma\boldsymbol{w}\boldsymbol{w}^\top]\Sigma + \frac{2}{d^2}\Sigma\boldsymbol{w}\boldsymbol{w}^\top\Sigma + \frac{1}{d}\rho\Sigma\right)$$

$$+ (\ell^2-\ell)\frac{d^2}{\ell^2}\left(\frac{1}{d^2}\Sigma\boldsymbol{w}\boldsymbol{w}^\top\Sigma\right) \tag{A.32}$$

$$= \left(1+\frac{1}{\ell}\right)\Sigma\boldsymbol{w}\boldsymbol{w}^\top\Sigma + \frac{d}{\ell}\left(\operatorname{tr}[\Sigma\boldsymbol{w}\boldsymbol{w}^\top]+\rho\right)\Sigma \tag{A.33}$$

$$\simeq \Sigma\boldsymbol{w}\boldsymbol{w}^\top\Sigma + \frac{d}{\ell}\left(\operatorname{tr}[\Sigma\boldsymbol{w}\boldsymbol{w}^\top]+\rho\right)\Sigma. \tag{A.34}$$

Secondly,

$$\mathbb{E}_{\boldsymbol{x},\epsilon}\left[\frac{d}{\ell^2}\sum_{i,j\leq\ell}y_i\boldsymbol{x}_iy_j^2\right] = \ell\frac{d}{\ell^2}\mathbb{E}_{\boldsymbol{x},\epsilon}\left[\boldsymbol{x}_1y_1^3\right] + (\ell^2-\ell)\frac{d}{\ell^2}\mathbb{E}_{\boldsymbol{x},\epsilon}\left[\boldsymbol{x}_1y_1\right]\mathbb{E}_{\boldsymbol{x},\epsilon}\left[y_2^2\right] \tag{A.35}$$

$$= \frac{1}{\ell d}\left(d\operatorname{tr}[\Sigma\boldsymbol{w}\boldsymbol{w}^\top]\Sigma + 2\Sigma\boldsymbol{w}\boldsymbol{w}^\top\Sigma\right)\boldsymbol{w} + 3\frac{1}{\ell}\rho\Sigma\boldsymbol{w}$$

$$+ \left(1-\frac{1}{\ell}\right)\left(\operatorname{tr}[\Sigma\boldsymbol{w}\boldsymbol{w}^\top]+\rho\right)\Sigma\boldsymbol{w} \tag{A.36}$$

$$\simeq \left(\operatorname{tr}[\Sigma\boldsymbol{w}\boldsymbol{w}^\top]+\rho\right)\Sigma\boldsymbol{w}. \tag{A.37}$$

Finally,

$$\mathbb{E}_{\boldsymbol{x},\epsilon}\left[\frac{1}{\ell^2}\sum_{i,j\leq\ell}y_i^2y_j^2\right] = \frac{1}{\ell}\mathbb{E}_{\boldsymbol{x},\epsilon}[y_1^4] + \frac{\ell^2-\ell}{\ell^2}\mathbb{E}_{\boldsymbol{x},\epsilon}[y_1^2]\mathbb{E}_{\boldsymbol{x},\epsilon}[y_2^2] \tag{A.38}$$

$$= \frac{1}{\ell}\left(\mathbb{E}_{\boldsymbol{x},\epsilon}[\boldsymbol{w}^\top xx^\top\boldsymbol{w}\boldsymbol{w}^\top\boldsymbol{x}\boldsymbol{x}^\top\boldsymbol{w}] + 6\rho\mathbb{E}_x[\boldsymbol{w}^\top\boldsymbol{x}\boldsymbol{x}^\top\boldsymbol{w}] + 3\rho^2\right)$$

$$+ \left(1-\frac{1}{\ell}\right)\left(\operatorname{tr}[\Sigma\boldsymbol{w}\boldsymbol{w}^\top]+\rho\right)^2 \tag{A.39}$$

$$\simeq \left(\operatorname{tr}[\Sigma\boldsymbol{w}\boldsymbol{w}^\top]+\rho\right)^2. \tag{A.40}$$

Combining these pieces gives (A.15). □

*Proof.* **[of Lemma 1]** We begin by noting that for both IDG and ICL test errors, we can expand

$$e(\Gamma) = \mathbb{E}_{\boldsymbol{x},\boldsymbol{w},\epsilon}\left[\left(y_{\ell+1}-\operatorname{tr}(\Gamma H_Z^\top)\right)^2\right] = \mathbb{E}_{\boldsymbol{w}}\left[\mathbb{E}_{\boldsymbol{x},\epsilon}\left[\left(y_{\ell+1}-\operatorname{tr}(\Gamma H_Z^\top)\right)^2\right]\right]$$

$$= \mathbb{E}_{\boldsymbol{w}}\left[\mathbb{E}_{\boldsymbol{x},\epsilon}\left[\left(y_{\ell+1}-\operatorname{vec}(\Gamma)^\top\operatorname{vec}(H_Z)\right)^2\right]\right]$$

$$= \mathbb{E}_{\boldsymbol{w}}\left[\mathbb{E}_{\boldsymbol{x},\epsilon}\left[y_{\ell+1}^2\right]\right] - 2\operatorname{vec}(\Gamma)^\top\mathbb{E}_{\boldsymbol{w}}\left[\mathbb{E}_{\boldsymbol{x},\epsilon}\left[y_{\ell+1}\operatorname{vec}(H_Z)\right]\right]$$

$$+ \operatorname{vec}(\Gamma)^\top\mathbb{E}_{\boldsymbol{w}}\left[\mathbb{E}_{\boldsymbol{x},\epsilon}\left[\operatorname{vec}(H_Z)\operatorname{vec}(H_Z)^\top\right]\right]\operatorname{vec}(\Gamma) \tag{A.41}$$

where the difference between IDG and ICL error comes from the different task distribution over which we take $\mathbb{E}_{\boldsymbol{w}}$. Note this $\mathbb{E}_{\boldsymbol{w}}$ is shorthand for "expectation over task distribution," which will be different distributions depending on if we are using $\mathcal{P}_{\text{IDG}}$ or $\mathcal{P}_{\text{ICL}}$.

Now we can use the above Lemma to simplify these terms drastically. For the IDG distribution, we have

$$\mathbb{E}_{\boldsymbol{w}}[\boldsymbol{w}] = \boldsymbol{b}_k\,, \qquad \mathbb{E}_{\boldsymbol{w}}[\boldsymbol{w}\boldsymbol{w}^\top] = R_k\,.$$

Therefore,

$$\mathbb{E}_{\boldsymbol{w}}[\mathbb{E}_{\boldsymbol{x},\epsilon}[y_{\ell+1}^2]] = \mathrm{tr}[\Sigma R_k] + \rho \tag{A.42}$$

$$\mathbb{E}_{\boldsymbol{w}}[\mathbb{E}_{\boldsymbol{x},\epsilon}[y_{\ell+1}H_Z]] = \frac{1}{d}\left[\Sigma R_k\Sigma \quad (\mathrm{tr}[\Sigma R_k] + \rho)\,\Sigma\boldsymbol{b}_k\right] = \frac{1}{d}A_{\mathrm{IDG}} \tag{A.43}$$

$$\mathbb{E}_{\boldsymbol{x},\epsilon}[\mathrm{vec}(H_Z)\,\mathrm{vec}(H_Z)^\top] \simeq \frac{1}{d}\Sigma \otimes \begin{bmatrix} \Sigma R_k\Sigma + \frac{d}{\ell}\left(\mathrm{tr}[\Sigma R_k] + \rho\right)\Sigma & (\mathrm{tr}[\Sigma R_k] + \rho)\,\Sigma\boldsymbol{b}_k \\ (\mathrm{tr}[\Sigma R_k] + \rho)\,(\Sigma\boldsymbol{b}_k)^\top & (\mathrm{tr}[\Sigma R_k] + \rho)^2 \end{bmatrix}$$

$$= \frac{1}{d}\Sigma \otimes B_{\mathrm{IDG}}\,. \tag{A.44}$$

For the ICL distribution, we have

$$\mathbb{E}_{\boldsymbol{w}}[\boldsymbol{w}] = 0\,, \qquad \mathbb{E}_{\boldsymbol{w}}[\boldsymbol{w}\boldsymbol{w}^\top] = C_{\mathrm{test}}\,,$$

and so

$$\mathbb{E}_{\boldsymbol{w}}\left[\mathbb{E}_{\boldsymbol{x},\epsilon}[y_{\ell+1}^2]\right] = \mathbb{E}_{\boldsymbol{w}}\left[\mathrm{tr}[\Sigma\boldsymbol{w}\boldsymbol{w}^\top] + \rho\right] = \mathrm{tr}[\Sigma C_{\mathrm{test}}] + \rho \tag{A.45}$$

$$\mathbb{E}_{\boldsymbol{w}}\left[\mathbb{E}_{\boldsymbol{x},\epsilon}[y_{\ell+1}H_Z]\right] = \frac{1}{d}\left[\Sigma C_{\mathrm{test}}\Sigma \quad 0\right] = \frac{1}{d}A_{\mathrm{ICL}} \tag{A.46}$$

$$\mathbb{E}_{\boldsymbol{w}}\left[\mathbb{E}_{\boldsymbol{x},\epsilon}[\mathrm{vec}(H_Z)\,\mathrm{vec}(H_Z)^\top]\right] \simeq \frac{1}{d}\Sigma \otimes \begin{bmatrix} \Sigma C_{\mathrm{test}}\Sigma + \frac{d}{\ell}\left(\mathrm{tr}[\Sigma C_{\mathrm{test}}] + \rho\right)\Sigma & 0 \\ 0^\top & (\mathrm{tr}[\Sigma C_{\mathrm{test}}] + \rho)^2 \end{bmatrix}$$

$$= \frac{1}{d}\Sigma \otimes B_{\mathrm{ICL}}\,. \tag{A.47}$$

Upon noting that

$$\mathrm{vec}(\Gamma)^\top(\Sigma \otimes B)\,\mathrm{vec}(\Gamma) = \mathrm{tr}\left(\Sigma\Gamma B\Gamma^\top\right)\,,$$

substituting these components into (A.41) gives the required results. ∎

# B  RANDOM MATRIX THEORY CALCULATION

Note that while Result 1 is offered for general token covariance $\Sigma$, all steps and results in this section and the following section assume $\Sigma = I_d$.

The setup and structure of this formalism and corresponding results will follow Section SI.3 and SI.4 of Lu et al. (2025). There will be key differences caused by the different task statistics $\mathcal{N}(0, C_{\mathrm{train}}), \mathcal{N}(0, C_{\mathrm{test}})$ that we use in this work compared to the isotropic tasks used in Lu et al. (2025). Since the token structure is the same between the work of Lu et al. (2025) and our work here, we will not prove various error bounding claims or approximations rigorously here, as such bounds would be identical to what can be found in Lu et al. (2025). However, care will be taken to highlight the distinction and differences required for handling the additional complexity introduced by our consideration of non-isotropic tasks.

We will use $\approx$ to denote two scalar quantities which converge in probability in the high-dimensional limit, or for two matrices to represent deterministic equivalence.

**Definition 1.** *For two $d \times d$ (possibly random) matrices $A_d$ and $B_d$, we write $A_d \simeq B_d$ if $\mathrm{tr}[M_d(A_d - B_d)] \to 0$ in probability as $d \to \infty$ for any sequence of bounded spectral norm test matrices $M_d$ (Atanasov et al., 2024; Lu et al., 2025).*

As noted above, we will not attempt to control rates of convergence.

We begin by considering the closed-form optimized parameters given by (11). This optimization problem can be solved explicitly as

$$\mathrm{vec}(\Gamma^*) = \left(\frac{n}{d}\lambda I + \sum_{\mu=1}^n \mathrm{vec}(H_{Z^\mu})\mathrm{vec}(H_{Z^\mu})^\top\right)^{-1} \sum_{\mu=1}^n y_{\ell+1}^\mu \mathrm{vec}(H_{Z^\mu})\,. \tag{B.1}$$

Ultimately we wish to characterize the ICL and IDG error of these parameters, which are given by Result 1 as

$$\mathcal{E}_{\text{ICL}}(\Gamma^*) = \rho + \text{tr}[C_{\text{test}}] - 2\,\text{tr}[\Gamma^* A_{\text{ICL}}^\top] + \text{tr}[\Gamma^* B_{\text{ICL}}(\Gamma^*)^\top] \tag{B.2}$$

$$\mathcal{E}_{\text{IDG}}(\Gamma^*) = \rho + \text{tr}[R_k] - 2\,\text{tr}[\Gamma^* A_{\text{IDG}}^\top] + \text{tr}[\Gamma^* B_{\text{IDG}}(\Gamma^*)^\top] \tag{B.3}$$

for

$$A_{\text{ICL}} = [C_{\text{test}} \quad 0] \tag{B.4}$$

$$B_{\text{ICL}} = \begin{bmatrix} C_{\text{test}} + \frac{1}{\alpha_{\text{test}}}\left(\text{tr}[C_{\text{test}}] + \rho\right) & 0 \\ 0^\top & \left(\text{tr}[C_{\text{test}}] + \rho\right)^2 \end{bmatrix} \tag{B.5}$$

$$A_{\text{IDG}} = [R_k \quad \left(\text{tr}[R_k] + \rho\right)\boldsymbol{b}_k] \tag{B.6}$$

$$B_{\text{IDG}} = \begin{bmatrix} R_k + \frac{1}{\alpha_{\text{train}}}\left(\text{tr}[R_k] + \rho\right) & \left(\text{tr}[R_k] + \rho\right)\boldsymbol{b}_k \\ \left(\text{tr}[R_k] + \rho\right)\boldsymbol{b}_k^\top & \left(\text{tr}[R_k] + \rho\right)^2 \end{bmatrix}. \tag{B.7}$$

Again, as we are distinguishing between pretraining and testing context lengths, we use

$$\alpha_{\text{train}} = \ell_{\text{train}}/d, \qquad \alpha_{\text{test}} = \ell_{\text{test}}/d.$$

These expressions for $\Gamma^*$ and its corresponding errors depend explicitly on the randomness found in the particular sample of the data $x, w, \epsilon$; we wish to be able to characterize the typical performance, and thus we need to average over this disorder. We will do so by following a random-matrix style computation that computes a deterministic equivalent for the parameter matrix $\Gamma^*$ and its corresponding ICL and IDG errors, valid in high dimensions in the proportional limit

$$\frac{\ell_{\text{train}}}{d} \equiv \alpha_{\text{train}} = \Theta(1), \quad \frac{\ell_{\text{test}}}{d} \equiv \alpha_{\text{test}} = \Theta(1), \quad \frac{k}{d} \equiv \kappa = \Theta(1), \quad \frac{n}{d^2} \equiv \tau = \Theta(1). \tag{B.8}$$

To do this, we will express necessary quantities in terms of *resolvents.*

### RESOLVENT AND EXTENDED RESOLVENT SETUP

$\Gamma^*$ above can be rewritten as

$$\text{vec}(\Gamma^*) = G\left(\sum_{\mu \in [n]} y_\mu \,\text{vec}(H_\mu)\right)/d, \tag{B.9}$$

where $G$ is the resolvent matrix

$$G = \left(\sum_{\mu \in [n]} \text{vec}(H_\mu)\,\text{vec}(H_\mu)^\top/d + \tau\lambda I\right)^{-1}. \tag{B.10}$$

We will find it helpful to explicitly include an additional matrix $B_{\text{placeholder}} \in \mathbb{R}^{(d^2+d)\times(d^2+d)}$ (that is positive-semidefinite)

$$G(\pi) = \left(\sum_{\mu \in [n]} \text{vec}(H_\mu)\,\text{vec}(H_\mu)^\top/d + \pi B_{\text{placeholder}} + \tau\lambda I\right)^{-1}, \tag{B.11}$$

for a non-negative scalar $\pi$. Notice that $G(0) = G$. Eventually $B_{\text{placeholder}}$ will be explicitly related to $B_{\text{ICL}}$ or $B_{\text{IDG}}$ in a way that will allow us to more easily compute the $\Gamma B_{\text{ICL}}\Gamma^\top$ or $\Gamma B_{\text{IDG}}\Gamma^\top$ term in the error formulas.

We can write $G$ in a cleaner and more useful way by concatenating $y_\mu$ and $\text{vec}(H_\mu)$ into an extended vector

$$\boldsymbol{z}_\mu = \begin{bmatrix} y_\mu/d \\ \text{vec}(H_\mu)/\sqrt{d} \end{bmatrix} \in \mathbb{R}^{d^2+d+1}. \tag{B.12}$$

We also extend $B_{\text{placeholder}}$ as

$$B_{\text{ext}} = \begin{bmatrix} 0 & \\ & B_{\text{placeholder}} \end{bmatrix}, \tag{B.13}$$

We then define an extended resolvent (with very similar structure as $G$) to be

$$G_{\text{ext}}(\pi) = \frac{1}{\sum_{\mu \in [n]} \boldsymbol{z}_\mu \boldsymbol{z}_\mu^\top + \pi B_{\text{ext}} + \tau\lambda I}. \tag{B.14}$$

We see that $z_\mu z_\mu^\top$ and $B_{\text{ext}}$ have block structure, and so expanding the block inverse we see

$$G_{\text{ext}}(\pi) = \begin{bmatrix} c(\pi) & -c(\pi)q^\top(\pi) \\ -c(\pi)\boldsymbol{q}(\pi) & G(\pi) + c(\pi)\boldsymbol{q}(\pi)q^\top(\pi) \end{bmatrix}, \qquad (\text{B.15})$$

for

$$\boldsymbol{q}(\pi) \equiv \frac{1}{d^{3/2}} G(\pi)\left(\sum_{\mu\in[n]} y_\mu \operatorname{vec}(H_\mu)\right) \in \mathbb{R}^{d(d+1)} \qquad (\text{B.16})$$

and $c(\pi)$ defined self-consistently by

$$\frac{1}{c(\pi)} = \frac{1}{d^2}\sum_{\mu\in[n]} y_\mu^2 + \tau\lambda - \frac{1}{d^3}\sum_{\mu,\nu\in[n]} y_\mu y_\nu \operatorname{vec}(H_\mu)^\top G(\pi)\operatorname{vec}(H_\nu). \qquad (\text{B.17})$$

At $\pi = 0$ we can now see that $G_{\text{ext}}$ explicitly contains information about the optimal parameters $\Gamma^*$ we're considering, as

$$\boldsymbol{q}(0) = \frac{1}{\sqrt{d}}\operatorname{vec}(\Gamma^*). \qquad (\text{B.18})$$

We will thus proceed by computing a deterministic equivalent for this extended resolvent.

DETERMINISTIC EQUIVALENT FOR THE EXTENDED RESOLVENT

We will closely follow Lu et al. (2025), omitting details that remain unchanged between the formalism in their work and ours here. Intuition-based headings will be given in *red*.

The computation proceeds by defining a "leave-one-out" version of $G_{\text{ext}}$,

$$G_{\text{ext}}^{[\mu]} = \frac{1}{\sum_{\nu\neq\mu} z_\nu z_\nu^\top + \pi B_{\text{ext}} + \tau\lambda I}. \qquad (\text{B.19})$$

By construction,

$$G_{\text{ext}}\left(\sum_{\mu\in[n]} z_\mu z_\mu^\top + \pi B_{\text{ext}} + \tau\lambda I\right) = I. \qquad (\text{B.20})$$

$$\sum_{\mu\in[n]} \frac{1}{1 + z_\mu^\top G_{\text{ext}}^{[\mu]} z_\mu} G_{\text{ext}}^{[\mu]} z_\mu z_\mu^\top + G_{\text{ext}}(\pi B_{\text{ext}} + \tau\lambda I) = I. \qquad (\text{B.21})$$

*Average over disorder in $x$, $\epsilon$, which concentrates.* The term $z_\mu^\top G_{\text{ext}}^{[\mu]} z_\mu$ is well-behaved, specifically as argued by Lu et al. (2025), it concentrates around its conditional expectation when the task $\boldsymbol{w}_\mu$ and $G_{\text{ext}}^{[\mu]}$ is fixed. We thus have

$$z_\mu^\top G_{\text{ext}}^{[\mu]} z_\mu \simeq \chi^\mu(\boldsymbol{w}_\mu) \qquad (\text{B.22})$$

where

$$\chi^\mu(\boldsymbol{w}_\mu) \equiv \frac{1}{d^2}\operatorname{tr}\left([G_{\text{ext}}^\mu]_{\backslash 0} \cdot [I \otimes E(\boldsymbol{w}_\mu)]\right), \qquad (\text{B.23})$$

and

$$E(\boldsymbol{w}) \equiv \begin{bmatrix} \boldsymbol{w}\boldsymbol{w}^\top + \frac{1}{\alpha}\left(\rho + \operatorname{tr}[\boldsymbol{w}\boldsymbol{w}^\top]\right) I_d & \left(\rho + \operatorname{tr}[\boldsymbol{w}\boldsymbol{w}^\top]\right)\boldsymbol{w} \\ \left(\rho + \operatorname{tr}[\boldsymbol{w}\boldsymbol{w}^\top]\right)w^\top & \left(\rho + \operatorname{tr}[\boldsymbol{w}\boldsymbol{w}^\top]\right)^2 \end{bmatrix}. \qquad (\text{B.24})$$

Replacing $z_\mu^\top G_{\text{ext}}^{[\mu]} z_\mu$ in (B.21) with $\chi^\mu(\boldsymbol{w}_\mu)$ gives

$$\sum_{\mu\in[n]} \frac{1}{1 + \chi^\mu(\boldsymbol{w}_\mu)} G_{\text{ext}}^{[\mu]} z_\mu z_\mu^\top + G_{\text{ext}}(\pi B_{\text{ext}} + \tau\lambda I) \simeq I. \qquad (\text{B.25})$$

Next, we will also average the term $z_\mu z_\mu^\top$ on on the left-hand side of (B.25) over the remaining disorder in $\boldsymbol{x}$, $\epsilon$, holding $\boldsymbol{w}$ fixed, and replace $z_\mu z_\mu^\top$ with this conditional expectation. Doing so introduces some small error which is bounded in Lu et al. (2025). From this we have

$$\sum_{\mu\in[n]} \frac{1}{1 + \chi^\mu(\boldsymbol{w}_\mu)} G_{\text{ext}}^{[\mu]} \mathbb{E}_{\boldsymbol{x},\epsilon}[z_\mu z_\mu^\top] + G_{\text{ext}}(\pi B_{\text{ext}} + \tau\lambda I) \simeq I. \qquad (\text{B.26})$$

We already have enough to compute $\mathbb{E}_{\boldsymbol{x},\epsilon}[\boldsymbol{z}_\mu \boldsymbol{z}_\mu^\top]$ using Lemma 2 as

$$\mathbb{E}_{\boldsymbol{x},\epsilon}[\boldsymbol{z}_\mu \boldsymbol{z}_\mu^\top] \simeq \frac{1}{d^2} \Upsilon, \tag{B.27}$$

where because it will appear in subsequent equations we define the matrix

$$\Upsilon \equiv \begin{bmatrix} \rho + \mathrm{tr}[\boldsymbol{w}\boldsymbol{w}^\top] & \frac{1}{\sqrt{d}} \mathrm{vec}\left(\begin{bmatrix} \boldsymbol{w}\boldsymbol{w}^\top & (\rho + \mathrm{tr}[\boldsymbol{w}\boldsymbol{w}^\top])\boldsymbol{w} \end{bmatrix}\right)^\top \\ \frac{1}{\sqrt{d}} \mathrm{vec}\left(\begin{bmatrix} \boldsymbol{w}\boldsymbol{w}^\top & (\rho + \mathrm{tr}[\boldsymbol{w}\boldsymbol{w}^\top])\boldsymbol{w} \end{bmatrix}\right) & I_d \otimes E(\boldsymbol{w}) \end{bmatrix}. \tag{B.28}$$

Now substitute this expression into (B.26) for the conditional expectation $\mathbb{E}_{\boldsymbol{x},\epsilon}\left[\boldsymbol{z}_\mu \boldsymbol{z}_\mu^\top\right]$, giving

$$\frac{\tau}{n} \sum_{\mu \in [n]} \frac{1}{1 + \chi^\mu(\boldsymbol{w}_\mu)} G_{\mathrm{ext}}^{[\mu]} \Upsilon + G_{\mathrm{ext}}(\pi B_{\mathrm{ext}} + \tau \lambda I) \simeq I \tag{B.29}$$

where we recall $\tau = n/d^2$.

*"Leave-one-out" terms behave like their full-sum equivalents.* In high dimensions and for large $n$, there is negligible difference between $\sum_{\nu \neq \mu}$ and $\sum_\mu$. Thus, we replace $G_{\mathrm{ext}}^\mu$ by $G_{\mathrm{ext}}$, and $\chi^\mu(\boldsymbol{w}_\mu)$ by

$$\chi(\boldsymbol{w}_\mu) \equiv \frac{1}{d^2} \mathrm{tr}\left([G_{\mathrm{ext}}]_{\backslash 0} \cdot [I \otimes E(\boldsymbol{w}_\mu)]\right). \tag{B.30}$$

So finally we have the expression for $G_{\mathrm{ext}}$

$$G_{\mathrm{ext}}\left(\frac{\tau}{n} \sum_{\mu \in [n]} \frac{1}{1 + \chi(\boldsymbol{w}_\mu)} \Upsilon + \pi B_{\mathrm{ext}} + \tau \lambda I\right) \simeq I. \tag{B.31}$$

*Exploit finiteness of training task set.* So far we are summing over $n$ task vectors, but really only $n/k$ of these are unique. Thus, we can simplify (B.31) as

$$G_{\mathrm{ext}}\left(\frac{\tau}{k} \sum_{j \in [k]} \frac{1}{1 + \chi(\boldsymbol{t}_j)} \Upsilon + \pi B_{\mathrm{ext}} + \tau \lambda I\right) \simeq I. \tag{B.32}$$

Indeed $\chi(\boldsymbol{t}_j)$ is also self-averaging in $\boldsymbol{t}_j$, and as argued by Lu et al. (2025), concentrates to its mean

$$\widehat{\chi}_{\mathrm{ave}} \equiv \frac{1}{k} \sum_{j \in [k]} \chi(\boldsymbol{t}_j). \tag{B.33}$$

We can thus simplify our expressions by substituting (B.30) into (B.33) and performing the sum over $w_1, ..., w_k$. Here we will use the that

$$\frac{1}{k} \sum_{j \in [k]} \left(\rho + \mathrm{tr}[\boldsymbol{t}_j \boldsymbol{t}_j^\top]\right) = \rho + \mathrm{tr}[R_k] \tag{B.34}$$

$$\frac{1}{k} \sum_{j \in [k]} \left(\rho + \mathrm{tr}[\boldsymbol{t}_j \boldsymbol{t}_j^\top]\right) \boldsymbol{t}_j \simeq \left(\rho + \mathrm{tr}[R_k]\right) \boldsymbol{b}_k \tag{B.35}$$

$$\frac{1}{k} \sum_{j \in [k]} \left(\rho + \mathrm{tr}[\boldsymbol{t}_j \boldsymbol{t}_j^\top]\right)^2 \simeq \left(\rho + \mathrm{tr}[R_k]\right)^2 \tag{B.36}$$

where for (B.35) and (B.36) we have used that $\mathrm{tr}[\boldsymbol{t}_j \boldsymbol{t}_j^\top] \simeq \mathrm{tr}[R_k]$ as $\mathrm{tr}[R_k] - \mathrm{tr}[\boldsymbol{t}_j \boldsymbol{t}_j^\top]$ has mean 0 and variance $\Theta(1/d)$. We thus have that

$$\frac{1}{k} \sum_{j \in [k]} E(\boldsymbol{t}_j) \simeq B_{\mathrm{IDG}} \tag{B.37}$$

and so

$$\widehat{\chi}_{\mathrm{ave}} = \frac{1}{d^2} \mathrm{tr}\left([G_{\mathrm{ext}}]_{\backslash 0} \cdot [I \otimes B_{\mathrm{IDG}}]\right). \tag{B.38}$$

*Final formula for extended resolvent in terms of training task sample.* The extended resolvent $G_{\text{ext}}(\pi)$ is asymptotically equivalent to

$$\widehat{\mathcal{G}}_e(\pi) \equiv \left( \frac{\tau}{1 + \widehat{\chi}_{\text{ave}}} \left[ \begin{array}{cc} \rho_{\text{train}} & \frac{1}{\sqrt{d}} \text{vec}\left( [R_k \quad \rho_{\text{train}} \boldsymbol{b}_k] \right)^\top \\ \frac{1}{\sqrt{d}} \text{vec}\left( [R_k \quad \rho_{\text{train}} \boldsymbol{b}_k] \right) & I_d \otimes B_{\text{IDG}} \end{array} \right] + \pi B_{\text{ext}} + \tau \lambda I \right)^{-1} \tag{B.39}$$

with $\widehat{\chi}_{\text{ave}}$ defined self-consistently by

$$\widehat{\chi}_{\text{ave}} \equiv \frac{1}{k} \sum_{j \in [k]} \chi(\boldsymbol{t}_j) = \frac{1}{d^2} \mathbb{E}\left[ \text{tr}\left( [G_{\text{ext}}]_{\backslash 0} \cdot [I_d \otimes B_{\text{IDG}}] \right) \right] \tag{B.40}$$

with

$$\boldsymbol{b}_k \equiv \frac{1}{k} \sum_{j \in [k]} \boldsymbol{t}_j, \qquad R_k \equiv \frac{1}{k} \sum_{j \in [k]} \boldsymbol{t}_j \boldsymbol{t}_j^\top, \qquad \rho_{\text{train}} \equiv \rho + \text{tr}[R_k], \tag{B.41}$$

$$B_{\text{IDG}} \equiv \begin{bmatrix} R_k + \frac{\rho_{\text{train}}}{\alpha} I_d & \rho_{\text{train}} \boldsymbol{b}_k \\ \rho_{\text{train}} \boldsymbol{b}_k^\top & \rho_{\text{train}}^2 \end{bmatrix}. \tag{B.42}$$

*Simplifying self-consistency variable $\chi$.* $\chi$ is defined in terms of the dominant component $[G_{\text{ext}}]_{\backslash 0}$ of $G_{\text{ext}}$. To simplify this, choose ansatz for $B_{\text{ext}}$ as

$$B_{\text{ext}} = \begin{bmatrix} 0 & 0 \\ 0 & I_d \otimes B_{\text{test}} \end{bmatrix}$$

where $B_{\text{test}}$ will later be $B_{\text{ICL}}$ or $B_{\text{IDG}}$ depending on which error we compute. Then $[G_{\text{ext}}]_{\backslash 0}$ can be expanded as

$$\left( \frac{\tau}{1 + \chi_\pi} I_d \otimes B_{\text{IDG}} + \pi I_d \otimes B_{\text{test}} + \tau \lambda I_d \otimes I_{d+1} \right)^{-1}$$

$$= I_d \otimes \left( \frac{\tau}{1 + \chi_\pi} B_{\text{IDG}} + \pi B_{\text{test}} + \tau \lambda I_{d+1} \right)^{-1} \tag{B.43}$$

$$= I_d \otimes F_E(\pi) \tag{B.44}$$

where

$$F_E(\pi) \equiv \left( \frac{\tau}{1 + \chi_\pi} B_{\text{IDG}} + \pi B_{\text{test}} + \tau \lambda I_{d+1} \right)^{-1}.$$

We can use this to replace $\widehat{\chi}_{\text{ave}}$ with $\chi_\pi$ defined self-consistently by

$$\chi_\pi \simeq \frac{1}{d} \text{tr}\left[ \left( \frac{\tau}{1 + \chi_\pi} B_{\text{IDG}} + \pi B_{\text{test}} + \lambda \tau I_{d+1} \right)^{-1} B_{\text{IDG}} \right] \tag{B.45}$$

*Relating $\widehat{\mathcal{G}}_e(\pi)$ to a deterministic equivalent for $\Gamma^*$.* Lu et al. (2025) shows that two key quantities $\text{tr}[\Gamma^* A^\top]$ and $\text{tr}[\Gamma^* B (\Gamma^*)^\top]$ in $\mathcal{E}(\Gamma^*)$ can also be computed from the extended resolvent $G_{\text{ext}}(\pi)$. As this is purely a matrix algebra claim, and independent of the particular task structure hidden within $\Gamma^*$, it generalizes immediately to our case. We thus include this result here without proof and refer to reader to Lu et al. (2025) for a derivation.

**Lemma 3.** *[From Lu et al. (2025)] For any matrix $A \in \mathbb{R}^{d \times (d+1)}$,*

$$\text{tr}[\Gamma^* A^\top] = \frac{-1}{c(0)\sqrt{d}} \begin{bmatrix} 0 & \text{vec}(A)^T \end{bmatrix} G_{\text{ext}}(0) \boldsymbol{e}_1, \tag{B.46}$$

*where $\boldsymbol{e}_1$ denotes the first natural basis vector in $\mathbb{R}^{d^2 + d + 1}$. For any symmetric and positive semidefinite matrix $B \in \mathbb{R}^{(d+1) \times (d+1)}$, if we set*

$$B_{\text{placeholder}} = I_d \otimes B \tag{B.47}$$

*in (B.13), then*

$$\text{tr}[\Gamma^* B (\Gamma^*)^\top] = \frac{d}{d\pi} \left( \frac{1}{c(\pi)} \right) \Big|_{\pi=0}. \tag{B.48}$$

Using Lemma 3, we can find the deterministic equivalent for $\Gamma^*$ from

$$\text{tr}[\Gamma^* A^\top] \simeq \frac{-1}{c(0)\sqrt{d}} \begin{bmatrix} 0 & \text{vec}(A)^T \end{bmatrix} \mathcal{G}_e(0) e_1 \tag{B.49}$$

$$= \frac{c^*(0)}{c(0)} \cdot \frac{1}{d} \text{tr} \left( \begin{bmatrix} R_k & \rho_{\text{train}} b_k \end{bmatrix} (B_{\text{IDG}} + \lambda(1+\chi_0)I)^{-1} A^\top \right) \tag{B.50}$$

$$\simeq \frac{1}{d} \text{tr} \left( \begin{bmatrix} R_k & \rho_{\text{train}} b_k \end{bmatrix} (B_{\text{IDG}} + \lambda(1+\chi_0)I)^{-1} A^\top \right). \tag{B.51}$$

and thus

$$\Gamma^*_{\text{eq}} \simeq \begin{bmatrix} R_k & \rho_{\text{train}} b_k \end{bmatrix} (B_{\text{IDG}} + \lambda(1+\chi_0)I)^{-1} \tag{B.52}$$

SELF-CONSISTENCY EQUATIONS AND FINITE TASK SAMPLE AVERAGES

So far, everything is left in terms of the quantities $b_k, R_k, B_{\text{IDG}}$ that depend on a typical sample of $k$ task vectors $w_1, \cdots, w_k$. What remains to be analyzed is the behavior of this sample and its sample statistics, *i.e.*, how does the finiteness of $k$ affect $\Gamma^*$. The following steps will deal with this.

*Characterization of $\chi_0$* We see that we almost have a characterization for the deterministic equivalent of $\Gamma^*$ but we need to understand what $\lambda(1+\chi_0)$ term is doing. From (B.45) we know $\chi_0$ is defined self-consistently from

$$\chi_0 \simeq \frac{1}{d} \text{tr} \left[ \left( \frac{\tau}{1+\chi_0} B_{\text{IDG}} + \lambda\tau I_{d+1} \right)^{-1} B_{\text{IDG}} \right] \tag{B.53}$$

$$\frac{\tau\chi_0}{1+\chi_0} \simeq \frac{1}{d} \text{tr} \left[ \left( B_{\text{IDG}} + \lambda(1+\chi_0)I_{d+1} \right)^{-1} B_{\text{IDG}} \right] \tag{B.54}$$

$$= \frac{d+1}{d} - \lambda(1+\chi_0)\frac{1}{d} \text{tr} \left[ \left( B_{\text{IDG}} + \lambda(1+\chi_0)I_{d+1} \right)^{-1} \right] \tag{B.55}$$

Remember $B_{\text{IDG}}$ is a $(d+1) \times (d+1)$ block matrix with upper $d \times d$ block given by $R_k + \rho_{\text{train}}/\alpha$. We wish to express the above resolvent just in terms of $R_k$.

Working heuristically and block-inverting the $B_{\text{IDG}} + \lambda(1+\chi)$ resolvent we have

$$\frac{1}{d} \text{tr} \left[ \left( B_{\text{IDG}} + \lambda(1+\chi_0)I_{d+1} \right)^{-1} \right] \simeq \frac{1}{d} \text{tr}(F_R(\sigma)) \tag{B.56}$$

where

$$\sigma \equiv \frac{\rho_{\text{train}}}{\alpha} + \lambda(1+\chi_0) \tag{B.57}$$

$$F_R(\sigma) \equiv (R_k + \sigma I_d)^{-1} \tag{B.58}$$

*Sample covariance $R_k$ and the Stieltjes transform.* We thus have

$$\frac{\tau\chi_0}{1+\chi_0} \simeq \frac{d+1}{d} - \lambda(1+\chi_0)\frac{1}{d} \text{tr}[F_R(\sigma_0)] \simeq 1 - \lambda(1+\chi_0)\mathcal{M}_\kappa \left( \frac{\rho_{\text{train}}}{\alpha} + \lambda(1+\chi_0) \right)$$

where $\mathcal{M}_\kappa(\sigma)$ is the asymptotic equivalent of the Stieltjes transform of the Wishart resolvent

$$F_R(\sigma) = (R_k + \sigma I_d)^{-1}.$$

Following Atanasov et al. (2024) for correlated feature wisharts, we have

$$\mathcal{M}_\kappa(\sigma) = \frac{1}{d} s \, \text{tr}((C_{\text{train}} + s\sigma\mathbb{I})^{-1}) \tag{B.59}$$

where $s$ is defined by the self consistency equation

$$\frac{1}{s} = 1 - \frac{1}{\kappa}\frac{1}{d} \text{tr}(C_{\text{train}}(C_{\text{train}} + s\sigma\mathbb{I})^{-1}) = 1 - \frac{1}{\kappa} + \frac{\sigma}{\kappa}\mathcal{M}_\kappa(\sigma) \tag{B.60}$$

and also only depends on $\sigma, \kappa$, and $C_{\text{train}}$. In other words, we have a self-consistency equation for $\mathcal{M}_\kappa(\sigma)$ as

$$\mathcal{M}_\kappa(\sigma) = \text{tr}\left[\left(\left(1 - \frac{1}{\kappa} + \frac{\sigma}{\kappa}\mathcal{M}_\kappa(\sigma)\right) C_{\text{train}} + \sigma I_d\right)^{-1}\right]. \qquad (B.61)$$

Similarly, we can express $F_R$ in deterministic form as

$$(R_k + \sigma I_d)^{-1} \simeq F_\kappa(\sigma) \equiv \left(\left(1 - \frac{1}{\kappa} + \frac{\sigma}{\kappa}\mathcal{M}_\kappa(\sigma)\right) C_{\text{train}} + \sigma I_d\right)^{-1}. \qquad (B.62)$$

Note that

$$(R_k + \sigma I_d)^{-2} \simeq -\frac{\mathrm{d}}{\mathrm{d}\sigma}F_\kappa(\sigma) = -F_\kappa(\sigma)\left(\left(\frac{1}{\kappa}\mathcal{M}_\kappa(\sigma) + \frac{\sigma}{\kappa}\frac{\mathrm{d}}{\mathrm{d}\sigma}\mathcal{M}_\kappa(\sigma)\right) C_{\text{train}} + I_d\right) F_\kappa(\sigma). \qquad (B.63)$$

Given a particular problem with a particular $C_{\text{train}}$, each of these can be computed.

*Self-Consistency equation for $\chi_0$* Recall to characterize $\Gamma^*_{\text{eq}}$ we need to understand $\lambda(1 + \chi_0) \equiv \tilde{\lambda}$. We now know this variable is defined by

$$\tilde{\lambda}\mathcal{M}_\kappa\left(\frac{\rho_{\text{train}}}{\alpha} + \tilde{\lambda}\right) - \frac{\lambda\tau}{\tilde{\lambda}} = 1 - \tau \qquad (B.64)$$

In the ridgeless $\lambda \to 0$ limit this equation becomes

$$\tilde{\lambda}\mathcal{M}_\kappa\left(\frac{\rho_{\text{train}}}{\alpha} + \tilde{\lambda}\right) = 1 - \tau. \qquad (B.65)$$

While this equation has nontrivial solutions for $\tau < 1$, its only solution is $\tilde{\lambda} = 0$ when $\tau > 1$.

*Final deterministic equivalence for $\Gamma^*$ in terms of Wishart resolvent.* Recall from above that

$$\Gamma^*_{\text{eq}} = (B_{\text{IDG}} + \hat{\lambda}I_{d+1})^{-1}[R_k \quad \rho_{\text{train}}\boldsymbol{b}_k]$$

We can use the trick that

$$[R_k \quad \rho_{\text{train}}\boldsymbol{b}_k] = S\left(B_{\text{IDG}} + \tilde{\lambda}_C I_{d+1} - \left(\frac{\rho_{\text{train}}}{\alpha} + \tilde{\lambda}_C\right)I_{d+1}\right) \qquad (B.66)$$

where $S$ is an almost-identity matrix that selects top $d \times (d+1)$ of a $(d+1) \times (d+1)$ matrix. We can then simplify

$$\Gamma^*_{\text{eq}} = S\left(I - \left(\frac{\rho_{\text{train}}}{\alpha} + \tilde{\lambda}\right)(B_{\text{IDG}} + \tilde{\lambda}I)^{-1}\right) \qquad (B.67)$$

$$\simeq [I - \sigma F_R(\sigma) \quad 0]. \qquad (B.68)$$

We stress that this is an approximation that comes from the full block inversion of the resolvent of $B_{\text{IDG}}$, but is robustly handled in Lu et al. (2025).

CHARACTERIZATION OF QUADRATIC TERM IN ERROR.

The only remaining error term left to approximate is those with the form $\Gamma B \Gamma^\top$. This is what we introduced $\pi$ and $B_{\text{ext}}$ for.

$$\frac{1}{d}\text{tr}\left(I_d\Gamma B_{\text{test}}\Gamma^\top\right) = \frac{1}{d}\text{vec}(\Gamma)^\top \Pi \text{vec}(\Gamma) = \frac{\mathrm{d}}{\mathrm{d}\pi}\frac{1}{c(\pi)}(\pi = 0) \qquad (B.69)$$

for $B_{\text{placeholder}} = I_d \otimes B_{\text{test}}$. The definition of $c(\pi)$ comes from the scalar term in the block inversion of $G_{\text{ext}}(\pi)$,

$$\mathcal{G}_e(\pi) = \begin{bmatrix} c^*(\pi) & -c^*(\pi)\left(\boldsymbol{q}^*(\pi)\right)^\top \\ -c^*(\pi)\boldsymbol{q}^*(\pi) & I \otimes F_E(\chi_\pi) + c^*(\pi)\boldsymbol{q}^*(\pi)(\boldsymbol{q}^*(\pi))^\top \end{bmatrix}, \qquad (B.70)$$

with

$$\frac{1}{c^*(\pi)} = \frac{\tau\rho_{\text{train}}}{1 + \chi_\pi} + \lambda\tau - \frac{\tau^2}{(1 + \chi_\pi)^2}\frac{1}{d}\text{tr}\left([R_k \quad \rho_{\text{train}}\boldsymbol{b}_k] F_E(\chi_\pi) [R_k \quad \rho_{\text{train}}\boldsymbol{b}_k]^\top\right). \qquad (B.71)$$

with

$$\chi_\pi \simeq \frac{1}{d} \operatorname{tr} \left[ \left( \frac{\tau}{1 + \chi_\pi} B_{\text{IDG}} + \pi B + \lambda \tau I_{d+1} \right)^{-1} B_{\text{IDG}} \right] \tag{B.72}$$

$$F_E(\chi_\pi) = \left( \frac{\tau}{1 + \chi_\pi} B_{\text{IDG}} + \pi B_{\text{test}} + \tau \lambda I_{d+1} \right)^{-1} \tag{B.73}$$

Now following through the calculus, we get

$$\frac{\mathrm{d}}{\mathrm{d}\pi} F_E(\chi_\pi) = -F_E(\chi_\pi) \left( B_{\text{test}} - \frac{\tau}{(1 + \chi_\pi)^2} \left( \frac{\mathrm{d}}{\mathrm{d}\pi} \chi_\pi \right) B_{\text{IDG}} \right) F_E(\chi_\pi) \tag{B.74}$$

$$\frac{\mathrm{d}}{\mathrm{d}\pi} \chi_\pi = \frac{1}{d} \operatorname{tr} \left[ \left( \frac{\mathrm{d}}{\mathrm{d}\pi} F_E(\chi_\pi) \right) B_{\text{IDG}} \right] \tag{B.75}$$

Plugging (B.74) into (B.75) and simplifying for $\chi_0'$ gives

$$\frac{\tau \chi_0'}{(1 + \chi_0)^2} = \frac{\frac{1}{d} \operatorname{tr} (B_{\text{test}} F_0 B_{\text{IDG}} F_0)}{\frac{1}{d} \operatorname{tr} (F_0 B_{\text{IDG}} F_0 B_{\text{IDG}}) - \tau} = \frac{\frac{1}{d} \operatorname{tr}(B_{\text{test}} F_0) - \tilde{\lambda} \frac{1}{d} \operatorname{tr}(B_{\text{test}} F_0^2)}{1 - 2\tilde{\lambda} \frac{1}{d} \operatorname{tr}(F_0) + \tilde{\lambda}^2 \frac{1}{d} \operatorname{tr}(F_0^2) - \tau} \tag{B.76}$$

for

$$F_0 \equiv \left( B_{\text{IDG}} + \tilde{\lambda} I_{d+1} \right)^{-1}, \qquad \tilde{\lambda} \equiv \lambda(1 + \chi_0).$$

We can finally simplify the quadratic term as

$$\left( \frac{\mathrm{d}}{\mathrm{d}\pi} \frac{1}{c(\pi)} \right) (\pi = 0) = \frac{1}{d} \operatorname{tr}(\Gamma_{\text{eq}}^* B_{\text{test}} \Gamma_{\text{eq}}^*)$$
$$- \tau \frac{\chi_0'}{(1 + \chi_0)^2} \left( \rho_{\text{train}} - \frac{1}{d} \operatorname{tr}(\Gamma_{\text{eq}}^* A_{\text{IDG}}^\top) - \tilde{\lambda} \frac{1}{d} \operatorname{tr}(\Gamma_{\text{eq}}^* (\Gamma_{\text{eq}}^*)^\top) \right) \tag{B.77}$$

DICTIONARY OF VARIOUS DETERMINISTIC EQUIVALENTS

**Lemma 4.** *The following is a summary of formulas. Their derivations follow simply from the above discussions and characterizations. Let*

$$F_0 \equiv \left( B_{\text{IDG}} + \tilde{\lambda} I_{d+1} \right)^{-1}, \qquad \tilde{\lambda} \equiv \lambda(1 + \chi_0)$$

*and $\Gamma_{\text{eq}}^*$ as given by (B.68). Then*

$$\operatorname{tr}[F_0(\tilde{\lambda})] \simeq \mathcal{M}_\kappa(\sigma) \tag{B.78}$$

$$\operatorname{tr}[F_0^2(\tilde{\lambda})] \simeq -\mathcal{M}_\kappa'(\sigma) \tag{B.79}$$

$$\operatorname{tr}[\Gamma_{\text{eq}}^* A_{\text{IDG}}^T] \simeq \operatorname{tr}[C_{\text{train}}] - \sigma + \sigma^2 \mathcal{M}_\kappa(\sigma) \tag{B.80}$$

$$\tilde{\lambda} \operatorname{tr}[\Gamma_{\text{eq}}^* (\Gamma_{\text{eq}}^*)^\top] \simeq \tilde{\lambda} \left( 1 - 2\sigma \mathcal{M}_\kappa(\sigma) - \sigma^2 \mathcal{M}_\kappa'(\sigma) \right) \tag{B.81}$$

$$\operatorname{tr}[\Gamma_{\text{eq}}^* A_{\text{ICL}}^\top] \simeq \operatorname{tr}[C_{\text{test}}] - \sigma \operatorname{tr}[F_\kappa(\sigma) C_{\text{test}}] \tag{B.82}$$

$$\operatorname{tr}[B_{\text{IDG}} F_0] = 1 - \tilde{\lambda} \mathcal{M}_\kappa(\sigma) \tag{B.83}$$

$$\operatorname{tr}[B_{\text{IDG}} F_0^2] = \mathcal{M}_\kappa(\sigma) + \tilde{\lambda} \mathcal{M}_\kappa'(\sigma) \tag{B.84}$$

$$\operatorname{tr}[B_{\text{ICL}} F_0] \simeq \operatorname{tr} \left[ \left( C_{\text{test}} + \frac{\rho_{\text{test}}}{\alpha_{\text{test}}} I_d \right) F_\kappa(\sigma) \right] \tag{B.85}$$

$$\operatorname{tr}[B_{\text{ICL}} F_0^2] \simeq - \operatorname{tr} \left[ \left( C_{\text{test}} + \frac{\rho_{\text{test}}}{\alpha_{\text{test}}} I_d \right) F_\kappa'(\sigma) \right] \tag{B.86}$$

$$\operatorname{tr}[\Gamma_{\text{eq}}^* B_{\text{IDG}} (\Gamma_{\text{eq}}^*)^\top] \simeq \operatorname{tr}[\Gamma_{\text{eq}}^* A_{\text{IDG}}^\top] - \tilde{\lambda} \operatorname{tr}[\Gamma_{\text{eq}}^* (\Gamma_{\text{eq}}^*)^\top] \tag{B.87}$$

$$\operatorname{tr}[\Gamma_{\text{eq}}^* B_{\text{ICL}} (\Gamma_{\text{eq}}^*)^\top] \simeq \operatorname{tr} \left[ \left( C_{\text{test}} + \frac{\rho_{\text{test}}}{\alpha_{\text{test}}} I_d \right) \left( I_d - 2\sigma F_\kappa(\sigma) - \sigma^2 F_\kappa'(\sigma) \right) \right] \tag{B.88}$$

## C PROOFS OF ASYMPOTIC IDG AND ICL ERROR CHARACTERIZATION

We finally have all the necessary components to write down the deterministic equivalents of the IDG and ICL errors given by Lemma 1 (or (B.3) and (B.2)). This will comprise our main theoretical result.

---

**Result:** High-dimensional formula for the IDG error

Let

$$F_\kappa(z) = \left( \left( 1 - \frac{1}{\kappa} + \frac{z}{\kappa} \mathcal{M}_\kappa(z) \right) C_{\text{train}} + z I_d \right)^{-1}, \tag{C.1}$$

$$F'_\kappa(z) = -F_\kappa(z) \left( \left( \frac{1}{\kappa} \mathcal{M}_\kappa(z) + \frac{1}{\kappa} \mathcal{M}'_\kappa(z) \right) C_{\text{train}} + I_d \right) F_\kappa(z) = \frac{\mathrm{d}}{\mathrm{d}z} F_\kappa(z), \tag{C.2}$$

$$\mathcal{M}_\kappa(z) = \frac{1}{d} \operatorname{tr}(F_\kappa(z)), \qquad \mathcal{M}'_\kappa(z) = \frac{1}{d} \operatorname{tr}(F'_\kappa(z)) \tag{C.3}$$

and

$$\tilde{\lambda} \mathcal{M}_\kappa(\sigma) + \lambda(\tau/\tilde{\lambda}) = 1 - \tau, \quad \tilde{\lambda} \geq 0 \tag{C.4}$$

$$\sigma = (\rho + c_{\text{train}})/\alpha_{\text{train}} + \tilde{\lambda}. \tag{C.5}$$

Then we have the deterministic equivalent of (A.2) as

$$\mathcal{E}_{\text{IDG}}(\Gamma^*) \simeq \tau \frac{\rho + \sigma - \sigma^2 \mathcal{M}_\kappa(\sigma) - \tilde{\lambda}(1 - 2\sigma \mathcal{M}_\kappa(\sigma) - \sigma^2 \mathcal{M}'_\kappa(\sigma))}{\tau - (1 - 2\tilde{\lambda} \mathcal{M}_\kappa(\sigma) - \tilde{\lambda}^2 \mathcal{M}'_\kappa(\sigma))} \tag{C.6}$$

$$\equiv e_{\text{IDG}}(\boldsymbol{\lambda}_{\text{train}})$$

---

**Result:** High-dimensional formula for the ICL error

Take $\mathcal{M}_\kappa(\sigma), \mathcal{M}'_\kappa(\sigma), F, F', \sigma,$ and $\tilde{\lambda}$ as defined as above, and $q \equiv e_{\text{IDG}}(\boldsymbol{\lambda}_{\text{train}})/\tau$. The deterministic equivalent of (A.3) is then

$$\mathcal{E}_{\text{ICL}}(\Gamma^*) \simeq \rho + \frac{\rho + c_{\text{test}}}{\alpha_{\text{test}}} \left( 1 + (q - 2\sigma)\mathcal{M}_\kappa(\sigma) + (q\tilde{\lambda} - \sigma^2)\mathcal{M}'_\kappa(\sigma) \right)$$

$$+ q \operatorname{tr}[C_{\text{test}} F_\kappa(\sigma)] + (q\tilde{\lambda} - \sigma^2) \operatorname{tr}[C_{\text{test}} F'_\kappa(\sigma)] \tag{C.7}$$

$$\equiv e_{\text{ICL}}(C_{\text{train}}, C_{\text{test}}).$$

---

*Proof.* Using Lemma 4 and (B.77) in (B.3) gives deterministic equivalent for $\mathcal{E}_{\text{IDG}}(\Gamma^*)$ as

$$\mathcal{E}_{\text{IDG}}(\Gamma^*) \simeq \rho + c_{\text{train}} - 2(c_{\text{train}} + \sigma + \sigma^2 \mathcal{M}_\kappa(\sigma)) + c_{\text{train}} + \sigma$$

$$+ \sigma^2 \mathcal{M}_\kappa(\sigma) - \tilde{\lambda}(1 - 2\sigma \mathcal{M}_\kappa(\sigma) - \sigma^2 \mathcal{M}'_\kappa(\sigma))$$

$$- \frac{1 - 2\tilde{\lambda} \mathcal{M}_\kappa(\sigma) - \tilde{\lambda}^2 \mathcal{M}'_\kappa(\sigma)}{-\tau + 1 - 2\tilde{\lambda} \mathcal{M}_\kappa(\sigma) - \tilde{\lambda}^2 \mathcal{M}'_\kappa(\sigma)}$$

$$\times \left( \rho_{\text{train}} - (c_{\text{train}} + \sigma + \sigma^2 \mathcal{M}_\kappa(\sigma)) - \tilde{\lambda}(1 - 2\sigma \mathcal{M}_\kappa(\sigma) - \sigma^2 \mathcal{M}'_\kappa(\sigma)) \right)$$

$$= \left( \rho - \sigma - \sigma^2 \mathcal{M}_\kappa(\sigma) - \tilde{\lambda}(1 - 2\sigma \mathcal{M}_\kappa(\sigma) - \sigma^2 \mathcal{M}'_\kappa(\sigma)) \right)$$

$$\times \left( 1 - \frac{1 - 2\tilde{\lambda} \mathcal{M}_\kappa(\sigma) - \tilde{\lambda}^2 \mathcal{M}'_\kappa(\sigma)}{-\tau + 1 - 2\tilde{\lambda} \mathcal{M}_\kappa(\sigma) - \tilde{\lambda}^2 \mathcal{M}'_\kappa(\sigma)} \right)$$

$$= \tau \frac{\rho - \sigma - \sigma^2 \mathcal{M}_\kappa(\sigma) - \tilde{\lambda}(1 - 2\sigma \mathcal{M}_\kappa(\sigma) - \sigma^2 \mathcal{M}'_\kappa(\sigma))}{\tau - (1 - 2\tilde{\lambda} \mathcal{M}_\kappa(\sigma) - \tilde{\lambda}^2 \mathcal{M}'_\kappa(\sigma))}$$

as required.

Using Lemma 4 and (B.77) in (B.2) gives

$$
\begin{aligned}
\mathcal{E}_{\mathrm{ICL}}(\Gamma^*) \simeq\ & \rho + c_{\mathrm{test}} \\
& - 2\left(c_{\mathrm{test}} - \sigma\,\mathrm{tr}\left[C_{\mathrm{test}} F_\kappa(\sigma)\right]\right) \\
& + \mathrm{tr}\left[\left(C_{\mathrm{test}} + \frac{\rho + c_{\mathrm{test}}}{\alpha_{\mathrm{test}}} I_d\right)\left(I_d - 2\sigma F_\kappa(\sigma) - \sigma^2 F_\kappa'(\sigma)\right)\right] \\
& + \mathrm{tr}\left[\left(C_{\mathrm{test}} + \frac{\rho + c_{\mathrm{test}}}{\alpha_{\mathrm{test}}} I_d\right)\left(F_\kappa(\sigma) + \tilde{\lambda} F_\kappa'(\sigma)\right)\right] \\
& \quad\times \frac{\rho + \sigma - \sigma^2 \mathcal{M}_\kappa(\sigma) - \tilde{\lambda}(1 - 2\sigma\mathcal{M}_\kappa(\sigma) - \sigma^2 \mathcal{M}_\kappa'(\sigma))}{\tau - (1 - 2\tilde{\lambda}\mathcal{M}_\kappa(\sigma) - \tilde{\lambda}^2 \mathcal{M}_\kappa'(\sigma))}\ .
\end{aligned}
$$

Replacing

$$
q \equiv \frac{\rho + \sigma - \sigma^2 \mathcal{M}_\kappa(\sigma) - \tilde{\lambda}(1 - 2\sigma\mathcal{M}_\kappa(\sigma) - \sigma^2 \mathcal{M}_\kappa'(\sigma))}{\tau - (1 - 2\tilde{\lambda}\mathcal{M}_\kappa(\sigma) - \tilde{\lambda}^2 \mathcal{M}_\kappa'(\sigma))} \tag{C.8}
$$

and gathering $C_{\mathrm{test}}$ terms we get

$$
\begin{aligned}
e_{\mathrm{ICL}}(C_{\mathrm{train}}, C_{\mathrm{test}}) \simeq \rho + & \frac{\rho + c_{\mathrm{test}}}{\alpha_{\mathrm{test}}}\left(1 + (q - 2\sigma)\mathcal{M}_\kappa(\sigma) + (q\tilde{\lambda} - \sigma^2)\mathcal{M}_\kappa'(\sigma)\right) \\
& + \langle C_{\mathrm{test}}, q F_\kappa(\sigma) + (q\tilde{\lambda} - \sigma^2) F_\kappa'(\sigma)\rangle
\end{aligned} \tag{C.9}
$$

as required. The remaining terms only depending on the test distribution through $c_{\mathrm{test}}$, and thus not containing any structural information about the testing tasks, we call

$$
e_{\mathrm{scalar}}(\boldsymbol{\lambda}_{\mathrm{train}}, c_{\mathrm{test}}) \equiv \rho + \frac{\rho + c_{\mathrm{test}}}{\alpha_{\mathrm{test}}}\left(1 + (q - 2\sigma)\mathcal{M}_\kappa(\sigma) + (q\tilde{\lambda} - \sigma^2)\mathcal{M}_\kappa'(\sigma)\right) \tag{C.10}
$$

and the remaining $C_{\mathrm{test}}$-dependent terms form the $e_{\mathrm{misalign}}(C_{\mathrm{train}}, C_{\mathrm{test}})$ contribution. $\qquad\square$

Note that the above results define $\tilde{\lambda}$ self-consistently in (C.4) for general ridge $\lambda$. For the main document we have explicitly taken the ridgeless limit, although in general this is not necessary and one can work from (C.4) and use this $\tilde{\lambda}$ in all formulas.

**Remark 2.** *The ridgeless $\lambda \to 0$ limit of (C.4)*

$$
\tilde{\lambda}\mathcal{M}_\kappa\left(\frac{c_{\mathrm{train}} + \rho}{\alpha_{\mathrm{train}}} + \tilde{\lambda}\right) = 1 - \tau \tag{C.11}
$$

*exhibits different behavior for $\tau > 1$ and $\tau < 1$. For $\tau > 1$ in the ridgeless limit, $\tilde{\lambda} = 0$ explicitly.*

We can solve some of the self-consistency equations here explicitly in the isotropic case and recover previous results.

**Remark 3.** *In the ridgeless limit $\lambda \to 0$ with isotropic $C_{\mathrm{train}} = c_{\mathrm{train}} I_d$, have explicitly*

$$
\mathcal{M}_\kappa(\nu) = 2\left(\nu + c_{\mathrm{train}} - \frac{c_{\mathrm{train}}}{\kappa} + \sqrt{\left(\nu + c_{\mathrm{train}} - \frac{c_{\mathrm{train}}}{\kappa}\right)^2 + 4c_{\mathrm{train}}\frac{\nu}{\kappa}}\right)^{-1} \tag{C.12}
$$

$$
\tilde{\lambda} = \begin{cases} \left(\frac{1-\tau}{\tau}\mathcal{M}_{\kappa/\tau}\left(\frac{\rho + c_{\mathrm{train}}}{\alpha_{\mathrm{train}}}\right)\right)^{-1} & \tau < 1 \\ 0 & \tau > 1 \end{cases} \tag{C.13}
$$

*For $c_{\mathrm{train}} = 1, C_{\mathrm{test}} = I_d$, and $\alpha_{\mathrm{train}} = \alpha_{\mathrm{test}}$ this recovers the theoretical results of Lu et al. (2025).*

# D    ORDERING OF EIGENVALUES

Here we analyze the spectrum of the $C_{\text{train}}$-dependent component of the misalignment error term (19),

$$\mathcal{K} \equiv q F_\kappa(\sigma) + (q\tilde{\lambda} - \sigma^2) F'_\kappa(\sigma) \,.$$

Given the self-consistent definitions of $F_\kappa(\sigma)$ and $F'_\kappa(\sigma)$ in (16) and (B.63), we can write the eigenvalues of $\mathcal{K}$ as

$$\lambda_i(\mathcal{K}) = \frac{1}{A_\kappa \lambda_i(C_{\text{train}}) + \sigma} \left( q + (\sigma^2 - q\tilde{\lambda}) \frac{B_\kappa \lambda_i(C_{\text{train}}) + 1}{A_\kappa \lambda_i(C_{\text{train}}) + \sigma} \right) \,,$$

for

$$A_\kappa \equiv 1 - \frac{1}{\kappa} + \frac{\sigma}{\kappa} \mathcal{M}_\kappa(\sigma) \tag{D.1}$$

$$B_\kappa \equiv \frac{1}{\kappa} \mathcal{M}_\kappa(\sigma) + \frac{\sigma}{\kappa} \mathcal{M}'_\kappa(\sigma) \,. \tag{D.2}$$

> **Conjecture:** Eigenvalue Ordering
>
> We wish to show that
> $$\lambda_1(\mathcal{K}) \leq \lambda_2(\mathcal{K}) \leq \cdots \leq \lambda_d(\mathcal{K})$$
> for
> $$\lambda_1(C_{\text{train}}) > \lambda_2(C_{\text{train}}) > \cdots > \lambda_d(C_{\text{train}}) \,,$$
> *i.e.* the ordering of eigenvalues of $\mathcal{K}$ is opposite that of $C_{\text{train}}$.

We will separate this into two cases. For $\tau > 1$ we provide a rigorous proof. For $\tau < 1$, due to the unclear sign of $\nu^2 - q\tilde{\lambda}$ we cannot currently provide a rigorous proof with the same methodology. Instead, we provide numerical plots supporting our claim, with the goal of proving this formally for the $\tau < 1$ case in subsequent iterations of this work.

**Lemma.** *We have that $1 - \kappa \leq \sigma \mathcal{M}_\kappa(\sigma) \leq 1$ and $\sigma^2 \mathcal{M}'_\kappa(\sigma) \leq \kappa - 1$.*

*Proof.* $R_k$ is a positive semi-definite matrix and so each eigenvalue of $\sigma(R_k + \sigma I_d)^{-1}$ is a non-negative increasing function of $\sigma \geq 0$. Therefore $\sigma \operatorname{tr}[(R_k + \sigma I_d)^{-1}]$ is increasing in $\sigma$ and so

$$\lim_{d \to \infty} \sigma \operatorname{tr}[(R_k + \sigma I_d)^{-1}] = \sigma \mathcal{M}_\kappa(\sigma)$$

is an increasing non-negative function of $\sigma$. Thus we have

$$0 \leq \lim_{\sigma \to 0} \sigma \mathcal{M}_\kappa(\sigma) \leq \sigma \mathcal{M}_\kappa(\sigma).$$

Also as $R_k$ is positive definite it is obvious that

$$\sigma \mathcal{M}_\kappa(\sigma) = \lim_{d \to \infty} \sigma \operatorname{tr}[(R_k + \sigma I_d)^{-1}] \leq 1.$$

By an equivalent argument,

$$\sigma^2 \mathcal{M}'_\kappa(\sigma) \leq \lim_{\sigma \to 0} \sigma^2 \mathcal{M}'_\kappa(\sigma) \leq 0 \,.$$

It's helpful to first write the self-consistency equation

$$\mathcal{M}_\kappa(\sigma) = \operatorname{tr}\left[ \left( \left( 1 - \frac{1}{\kappa} + \frac{\sigma}{\kappa} \mathcal{M}_\kappa(\sigma) \right) C_{\text{train}} + \sigma I_d \right)^{-1} \right] \tag{D.3}$$

for $\mathcal{M}_\kappa(\sigma)$ equivalently as a self-consistency equation for the "renormalized ridge" $\tilde{\sigma}$ of this Wishart resolvent, defined by

$$\tilde{\sigma} \operatorname{tr}[(C_{\text{train}} + \tilde{\sigma})^{-1}] = \sigma \mathcal{M}_\kappa(\sigma)$$

or equivalently

$$\tilde{\sigma} = \frac{\sigma}{1 - \frac{1}{\kappa} + \frac{\tilde{\sigma}}{\kappa} \operatorname{tr}\left[(C_{\text{train}} + \tilde{\sigma})^{-1}\right]} \,. \tag{D.4}$$

In the $\sigma \to 0$ limit, $\tilde{\sigma}$ limits to 0 when $\kappa > 1$, and limits to the unique solution of

$$1 - \frac{1}{\kappa} + \frac{\tilde{\sigma}}{\kappa} \operatorname{tr}\left[(C_{\text{train}} + \tilde{\sigma})^{-1}\right] = 0 \tag{D.5}$$

for $\kappa < 1$. For our purposes, this means that

$$\lim_{\sigma \to 0} \sigma \mathcal{M}_\kappa(\sigma) = \begin{cases} 1 - \kappa, & \kappa < 1 \\ 0, & \kappa > 1 \end{cases}. \tag{D.6}$$

Differentiating with respect to $\sigma$ gives

$$\lim_{\sigma \to 0} \sigma^2 \mathcal{M}'_\kappa(\sigma) = \begin{cases} \kappa - 1, & \kappa < 1 \\ 0, & \kappa > 1 \end{cases} \tag{D.7}$$

$$\leq \kappa - 1$$

and so $1 - \kappa \leq \sigma \mathcal{M}_\kappa(\sigma)$ and $\sigma^2 \mathcal{M}'_\kappa(\sigma) \leq \kappa - 1$. $\qquad \square$

With this lemma in hand, we can prove the main claim. For $\tau > 1$, we have $\tilde{\lambda} = 0$ and so $\sigma = (\rho + c_{\text{train}})/\alpha$ and

$$\lambda_i(\mathcal{K}) = \frac{1}{A_\kappa \lambda_i(C_{\text{train}}) + \sigma}\left(q + \sigma^2 \frac{B_\kappa \lambda_i(C_{\text{train}}) + 1}{A_\kappa \lambda_i(C_{\text{train}}) + \sigma}\right) \,.$$

Without loss of generality, let's compare $\lambda_1(\mathcal{K})$ and $\lambda_2(\mathcal{K})$. First, we can see from the Lemma above that $A_\kappa \geq 0$, and thus we have that

$$\frac{1}{A_\kappa \lambda_1(C_{\text{train}}) + \sigma} \leq \frac{1}{A_\kappa \lambda_2(C_{\text{train}}) + \sigma}$$

.

Furthermore, have

$$A_\kappa - \sigma B_\kappa = \frac{1}{\kappa}((\kappa - 1) - \sigma^2 \mathcal{M}'_\kappa(\sigma)).$$

We reason about this quantity as follows. We have that $A_\kappa - \sigma B_\kappa > 0$ as a result of this lemma, and so $(A_\kappa - \sigma B_\kappa)\lambda_2 \leq (A_\kappa - \sigma B_\kappa)\lambda_1$. Rearranging gives

$$\frac{B_\kappa \lambda_1(C_{\text{train}}) + 1}{A_\kappa \lambda_1(C_{\text{train}}) + \sigma} \leq \frac{B_\kappa \lambda_2(C_{\text{train}}) + 1}{A_\kappa \lambda_2(C_{\text{train}}) + \sigma}$$

and so we are done as we've shown $\lambda_1(\mathcal{K}) \leq \lambda_2(\mathcal{K})$ for $\lambda_1(C_{\text{train}}) > \lambda_2(C_{\text{train}})$.

For $\tau < 1$, $\tilde{\lambda} \neq 0$ and so the negative-definite contribution of $q\tilde{\lambda}F'_\kappa(\sigma)$ complicates the above argument. We provide preliminary numerical evidence for this case as follows. We compute $\tilde{\lambda}$ numerically from its self-consistency equation (21) to compute $\mathcal{K}$, of which we plot its eigenvalues against the eigenvalues of $C_{\text{train}}$. As demonstrated schematically in Figure 5, for $\tau$ values less than 1, it is consistently the case that the eigenvalues of $\mathcal{K}$ are negatively correlated with the eigenvalues of $C_{\text{train}}$. We emphasise that this is a heuristic check, and we plan to prove this rigorously in future iterations of this work.

## E    TRACE INEQUALITIES FOR MISALIGNMENT

In this appendix, we give a self-contained expository proof of Ruhe's trace inequality in the form we require for Corollary 4.1. We actually present two lines of analysis: first a direct proof of the inequality, and then an alternative analysis based on Riemannian optimization.

We begin by recalling the general setup: let $A$ and $B$ be $d \times d$ real symmetric matrices. By the spectral theorem, they are orthogonally diagonalizable, with non-increasingly ordered real eigenvalues

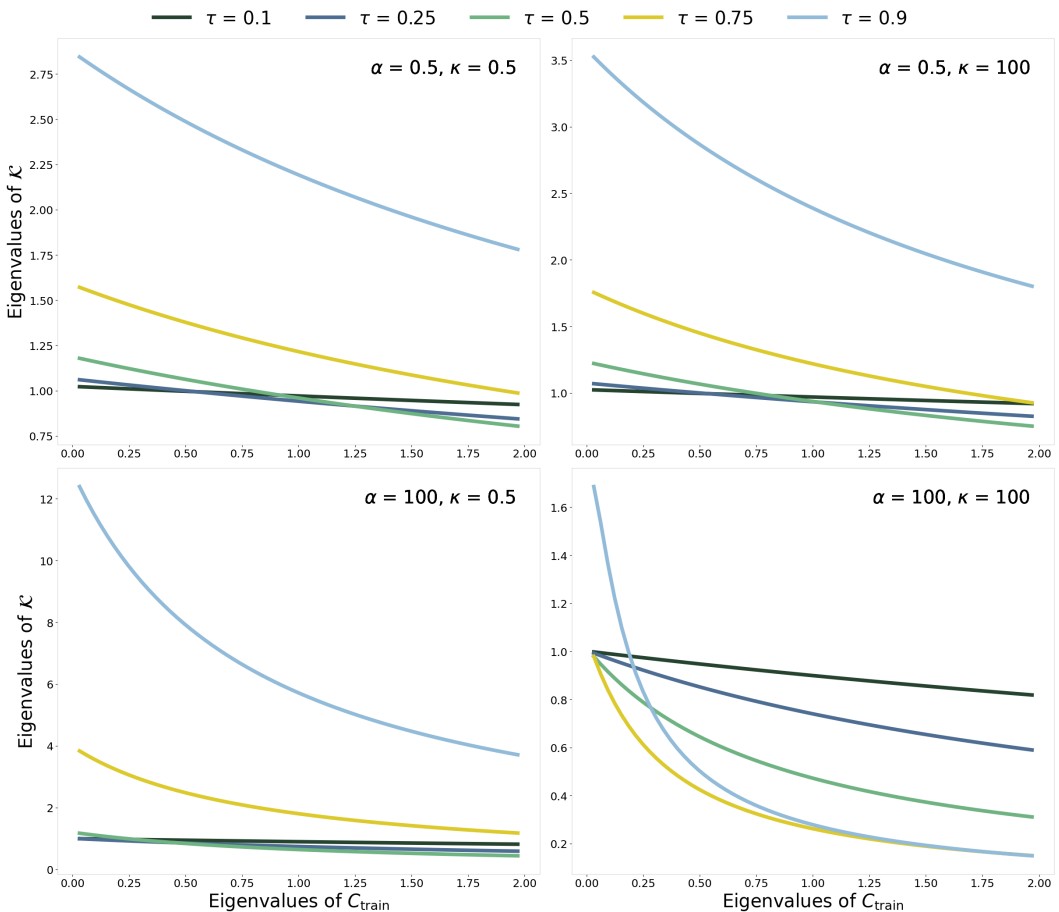

Figure 5: Demonstration of opposing eigenvalues for $\tau < 1$ values, at a range of $\kappa$ and $\alpha$ values. $C_{\text{train}}$ here is the same as in Figure 2, i.e. powerlaw spectrum.

$\lambda_1(A) \geq \lambda_2(A) \geq \cdots \geq \lambda_d(A)$ and $\lambda_1(B) \geq \lambda_2(B) \geq \cdots \geq \lambda_d(B)$, respectively. We want to show that

$$\sum_{j=1}^{d} \lambda_j(A)\lambda_{d-j+1}(A) \leq \text{tr}(AB) \leq \sum_{j=1}^{d} \lambda_j(A)\lambda_j(B), \tag{E.1}$$

with equality attained in either bound if and only if $A$ and $B$ are co-diagonalizable.

Writing the eigendecompositions of $A$ and $B$ as

$$A = O_A \Lambda_A O_A^\top \quad \text{and} \quad B = O_B \Lambda_B O_B^\top, \tag{E.2}$$

respectively, where $O_A O_A^\top = I_d$, $O_B O_B^\top = I_d$, $[\Lambda_A]_{ij} = \lambda_i(A)\delta_{ij}$, and $[\Lambda_B]_{ij} = \lambda_i(B)\delta_{ij}$, we see immediately that it suffices to consider the case in which $A$ is diagonal, with non-decreasing elements, as

$$\text{tr}(AB) = \text{tr}(\Lambda_A O \Lambda_B O^\top) \quad \text{where} \quad O \equiv O_A^\top O_B, \tag{E.3}$$

and the matrix $O\Lambda_B O^\top$ has the same eigenvalues as $B$. We can see that $A$ and $B$ are co-diagonalizable if and only if $O\Lambda_B O^\top$ is diagonal. Moreover, we can see that the claimed bounds can be attained only if $O\Lambda_B O^\top$ is in fact equal to $\Lambda_B$ up to pertmuation of its diagonal elements. What remains is to show that they are in fact bounds.

### E.1 DIRECT PROOF OF RUHE'S INEQUALITY

We now give a direct proof of the claimed inequality. We remark that the version of Ruhe's inequality proved in Marshall et al. (2010) is not sufficient for our purposes, as it assumes that both $A$ and $B$

are positive semi-definite, *i.e.*, that $\lambda_d(A) \geq 0$ and $\lambda_d(B) \geq 0$. We instead follow the proof for general Hermitian matrices outlined in Li (2020)'s blog post.

The strategy of this proof is to proceed by induction on the matrix dimension $d$. The claim clearly holds for $d = 1$, as then $A$ and $B$ are scalars equal to their only eigenvalue, and both bounds collapse. Now consider $d > 1$. Assuming—as noted before—that $A$ is diagonal with elements given by its ordered eigenvalues, we have

$$\text{tr}(AB) = \sum_{j=1}^{d} \lambda_j(A) B_{jj} \tag{E.4}$$

$$= \sum_{j=1}^{d} [\lambda_j(A) - \lambda_d(A) + \lambda_d(A)] B_{jj} \tag{E.5}$$

$$= \sum_{j=1}^{d-1} [\lambda_j(A) - \lambda_d(A)] B_{jj} + \sum_{j=1}^{d} \lambda_d(A) B_{jj}. \tag{E.6}$$

We now observe that

$$\sum_{j=1}^{d-1} [\lambda_j(A) - \lambda_d(A)] B_{jj} = \text{tr}(\Delta \tilde{B}), \tag{E.7}$$

where $\Delta$ is a $(d-1) \times (d-1)$ diagonal matrix with $\Delta_{jj} = \lambda_j(A) - \lambda_d(A)$ and $\tilde{B}$ is the $(d-1) \times (d-1)$ principal submatrix of $B$ given by discarding its last row and column. The ordering of the eigenvalues of $A$ implies the ordering

$$\lambda_1(A) - \lambda_d(A) \geq \cdots \geq \lambda_{d-1}(A) - \lambda_d(A) \geq 0 \tag{E.8}$$

of the eigenvalues of $\Delta$.

Thus, $\text{tr}(\Delta \tilde{B})$ is the trace of a product of $(d-1) \times (d-1)$ real symmetric matrices with $\Delta$ having its eigenvalues in non-decreasing order along the diagonal, so on the induction hypothesis we have the bound

$$\sum_{j=1}^{d-1} [\lambda_j(A) - \lambda_d(A)] \lambda_{d-j}(\tilde{B}) \leq \text{tr}(\Delta \tilde{B}) \leq \sum_{j=1}^{d-1} [\lambda_j(A) - \lambda_d(A)] \lambda_j(\tilde{B}). \tag{E.9}$$

By the Poincaré separation theorem (also known as the Cauchy interlacing theorem), as $\tilde{B}$ is a principal submatrix of the real symmetric matrix $B$, we have the inequality

$$\lambda_{j+1}(B) \leq \lambda_j(\tilde{B}) \leq \lambda_j(B) \tag{E.10}$$

for all $j \in [d-1]$.

Combining these results and using the fact that $\lambda_j(A) - \lambda_d(A) \geq 0$ by our ordering assumption, we thus have the upper bound

$$\text{tr}(AB) \leq \sum_{j=1}^{d-1} [\lambda_j(A) - \lambda_d(A)] \lambda_j(\tilde{B}) + \sum_{j=1}^{d} \lambda_d(A) B_{jj} \tag{E.11}$$

$$\leq \sum_{j=1}^{d-1} [\lambda_j(A) - \lambda_d(A)] \lambda_j(B) + \sum_{j=1}^{d} \lambda_d(A) B_{jj} \tag{E.12}$$

$$= \sum_{j=1}^{d-1} \lambda_j(A) \lambda_j(B) - \lambda_d(A) \sum_{j=1}^{d-1} \lambda_j(B) + \lambda_d(A) \sum_{j=1}^{d} B_{jj} \tag{E.13}$$

$$= \sum_{j=1}^{d-1} \lambda_j(A) \lambda_j(B) - \lambda_d(A) \sum_{j=1}^{d-1} \lambda_j(B) + \lambda_d(A) \sum_{j=1}^{d} \lambda_j(B) \tag{E.14}$$

$$= \sum_{j=1}^{d} \lambda_j(A) \lambda_j(B), \tag{E.15}$$

as $\text{tr}(B) = \sum_{j=1}^d B_{jj} = \sum_{j=1}^d \lambda_j(B)$. Here, the first line uses the induction hypothesis in the form re-stated above, the second line uses the upper bound from the Poincaré separation theorem, and the remaining three lines are just algebraic simplification.

Similarly, we have the lower bound

$$\text{tr}(AB) \geq \sum_{j=1}^{d-1} [\lambda_j(A) - \lambda_d(A)]\lambda_{d-j}(\tilde{B}) + \sum_{j=1}^d \lambda_d(A)B_{jj} \tag{E.16}$$

$$\geq \sum_{j=1}^{d-1} [\lambda_j(A) - \lambda_d(A)]\lambda_{d-j+1}(B) + \sum_{j=1}^d \lambda_d(A)B_{jj} \tag{E.17}$$

$$= \sum_{j=1}^{d-1} \lambda_j(A)\lambda_{d-j+1}(B) - \lambda_d(A)\sum_{j=1}^{d-1} \lambda_{d-j+1}(B) + \lambda_d(A)\sum_{j=1}^d \lambda_j(B) \tag{E.18}$$

$$= \sum_{j=1}^{d-1} \lambda_j(A)\lambda_{d-j+1}(B) - \lambda_d(A)\sum_{j=2}^d \lambda_j(B) + \lambda_d(A)\sum_{j=1}^d \lambda_j(B) \tag{E.19}$$

$$= \sum_{j=1}^d \lambda_j(A)\lambda_{d-j+1}(B). \tag{E.20}$$

By induction, we therefore conclude that

$$\sum_{j=1}^d \lambda_j(A)\lambda_{d-j+1}(B) \leq \text{tr}(AB) \leq \sum_{j=1}^d \lambda_j(A)\lambda_j(B), \tag{E.21}$$

as desired.

### E.2 RIEMANNIAN OPTIMIZATION

We now give an alternative perspective on this result based on the properties of orthogonal matrices. Consider

$$f(O) = \text{tr}(\Lambda_A O \tilde{\Lambda}_B O^\top) \tag{E.22}$$

as a function on the space of orthogonal matrices, where $\tilde{\Lambda}_B$ is a diagonal matrix with non-zero elements given by any permutation of the eigenvalues of $B$. Near the identity, we can write any orthogonal matrix as

$$O = I + tS + \frac{1}{2}t^2 S^2 + \mathcal{O}(t^3), \tag{E.23}$$

where $S$ is a skew-symmetric matrix ($S^\top = -S$) and $t$ is a small parameter. Substituting this expansion into $f(O)$, we have

$$f(O) = \text{tr}(\Lambda_A \tilde{\Lambda}_B) - t\,\text{tr}([\Lambda_A, \tilde{\Lambda}_B]S) + \frac{1}{2}t^2\,\text{tr}(\Lambda_A \tilde{\Lambda}_B S^2 - 2\Lambda_A S \tilde{\Lambda}_B S + \Lambda_A S^2 \tilde{\Lambda}_B) + \mathcal{O}(t^3), \tag{E.24}$$

where $[X, Y] = XY - YX$ is the matrix commutator. Irrespective of the permutation we have chosen to define $\tilde{\Lambda}_B$, $[\Lambda_A, \tilde{\Lambda}_B] = 0$, so $f(O)$ is stationary at the identity.

We therefore consider the quadratic term

$$H = \frac{1}{2}\,\text{tr}(\Lambda_A \tilde{\Lambda}_B S^2 - 2\Lambda_A S \tilde{\Lambda}_B S + \Lambda_A S^2 \tilde{\Lambda}_B), \tag{E.25}$$

as it will determine whether this stationary point is a local minimum, local maximum, or saddle. As $[\Lambda_A, \tilde{\Lambda}_B] = 0$, we can immediately simplify this to

$$H = \text{tr}(\Lambda_A \tilde{\Lambda}_B S^2 - \Lambda_A S \tilde{\Lambda}_B S). \tag{E.26}$$

For brevity, define $a_i = (\Lambda_A)_{ii}$ and $b_i = (\tilde{\Lambda}_B)_{ii}$. Then, expanding in components,

$$H = \sum_{i,j=1}^{d} (a_i b_i - a_i b_j) S_{ij} S_{ji} \tag{E.27}$$

$$= -\sum_{i,j=1}^{d} a_i (b_i - b_j) S_{ij}^2, \tag{E.28}$$

as by skew-symmetry $S_{ji} = -S_{ij}$. Using the fact that the $i = j$ terms clearly vanish, we have

$$H = -\sum_{i<j} a_i (b_i - b_j) S_{ij}^2 - \sum_{i>j} a_i (b_i - b_j) S_{ij}^2. \tag{E.29}$$

By re-labling indices $i \leftrightarrow j$ and using the fact that $S_{ji}^2 = S_{ij}^2$,

$$\sum_{i>j} a_i (b_i - b_j) S_{ij}^2 = -\sum_{i<j} a_j (b_i - b_j) S_{ij}^2 \tag{E.30}$$

so upon re-combining terms

$$H = -\sum_{i<j} (a_i - a_j)(b_i - b_j) S_{ij}^2. \tag{E.31}$$

This now depends on the ordering of the eigenvalues, and thus on the permutation of indices. If the eigenvalues of both $A$ and $B$ are in non-increasing order such that $a_i \geq a_j$ and $b_i \geq b_j$ whenever $i < j$, we have $H < 0$, and the identity is a local maximum of $f(O)$. In contrast, if they are oppositely ordered, then $H > 0$ and the identity is a local minimum. Otherwise, the stationary point is a saddle. These results are consistent with Ruhe's inequality as proved above.

## F    PHASE TRANSITION WITH INCREASING PRETRAINING TASK DIVERSITY

As we have seen throughout, the task diversity parameter is crucial because it quantifies the quality of the pretraining data: higher $\kappa$ implies a richer span of task variations, enabling the in-context learner to infer the structure shared between pretrain and test sets more accurately. Understanding $\kappa$ thus sheds light on the implicit algorithm performed by the model. Previous work from Lu et al. (2025); Raventós et al. (2023) in particular have investigated the performance of ICL as $\kappa$ increases. In particular, Lu et al. (2025) showed, in the isotropic task case, that there is a phase transition at $\kappa = 1$: for $\kappa < 1$, the model has insufficient task diversity to be able to generalize within tasks, and is forced to memorize the training tasks, while for $\kappa > 1$, the model has seen sufficient tasks to generalize in-context efficiently.

The settings considered in both Lu et al. (2025); Raventós et al. (2023) focus on isotropic tasks where the pretraining and testing task distributions are the same. Here, we verify that not only does this phase transition in task-diversity remain in the presence of structured pretraining task distributions, but further is independent of the test distribution (see Figure 6).

---

**Corollary F.1:** Phase transition in task diversity

In the proportional limit of $\alpha, \tau \to \infty$ such that $\tau = \alpha/\gamma$ for $\gamma = \Theta(1)$ fixed, there is a phase transition in $e_{\text{ICL}}$ at $\kappa = \text{rank}(C_{\text{train}})/d$. Specifically, for $r \equiv \text{rank}(C_{\text{train}})/d$ then

$$\lim_{\tau, \alpha \to \infty} e_{\text{ICL}}(C_{\text{train}}, C_{\text{test}}) = \begin{cases} \rho + \left(1 + \frac{\rho}{\rho + c_{\text{train}}}\gamma\right) \text{tr}\left[C_{\text{test}} (x_1 C_{\text{train}} + I_d)^{-1}\right] & \kappa < r \\ \rho + \left(1 + \frac{\rho}{\rho + c_{\text{train}}}\gamma\right) \text{tr}\left[C_{\text{test}} (x_2 C_{\text{train}} + I_d)^{-1}\right] & \kappa > r \end{cases} \tag{F.1}$$

where $x_1$ and $x_2$ are defined self-consistently as solutions of

$$\text{tr}\left[(x_1 C_{\text{train}} + I_d)^{-1}\right] = 1 - \kappa \tag{F.2}$$

$$\text{tr}\left[(x_2 C_{\text{train}} + I_d)^{-1}\right] = 1 - r. \tag{F.3}$$

---

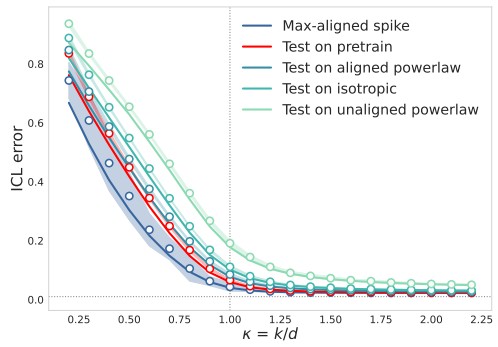
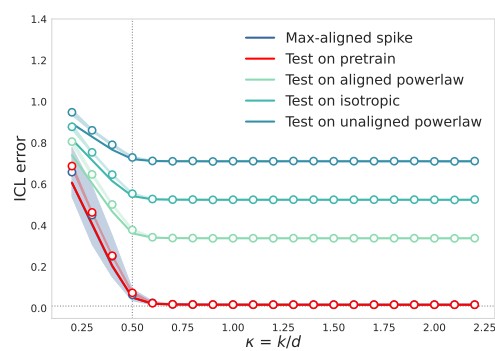

(a) Full-rank phase transition           (b) Half-rank phase transition

Figure 6: Theory curves with corresponding numerical simulations showing phase transition of ICL error in $\kappa$ for a range of test structures. For (a) $C_{\text{train}}$ is the same as in Figure 1 and is full-rank; for (b) $C_{\text{train}} = \text{diag}[2I_{d/2}, 0_{d/2}]$, thus half-rank. *Parameters:* $d = 80, \alpha = 80, \tau = 80, \rho = 0.01$. Tests for each are done on $C_{\text{train}}$, $I_d$, powerlaw (spectral power 0.5, aligned and unaligned with $C_{\text{train}}$), as well as the rank-1 $C_{\text{test}}$ optimal from Result 4.2.

To prove this result, we first suppose that $C_{\text{train}}$ is invertible. We start by writing the self-consistency equation

$$\mathcal{M}_\kappa(\sigma) = \frac{1}{d} \text{tr} \left( \left( \left( 1 - \frac{1}{\kappa} + \frac{\sigma}{\kappa} \mathcal{M}_\kappa(\sigma) \right) C_{\text{train}} + \sigma I_d \right)^{-1} \right). \tag{F.4}$$

for $\mathcal{M}_\kappa(\sigma)$ equivalently as a self-consistency equation for the renormalized ridge $\tilde{\sigma}$, defined by

$$\tilde{\sigma} \, \text{tr}[(C_{\text{train}} + \tilde{\sigma})^{-1}] = \sigma \mathcal{M}_\kappa(\sigma)$$

or equivalently

$$\tilde{\sigma} = \frac{\sigma}{1 - \frac{1}{\kappa} + \frac{\tilde{\sigma}}{\kappa} \text{tr} \left[ (C_{\text{train}} + \tilde{\sigma})^{-1} \right]}. \tag{F.5}$$

In the $\sigma \to 0$ limit, $\tilde{\sigma}$ limits to 0 when $\kappa > 1$, and limits to the unique solution of

$$1 - \frac{1}{\kappa} + \frac{\tilde{\sigma}}{\kappa} \text{tr} \left[ (C_{\text{train}} + \tilde{\sigma})^{-1} \right] = 0 \tag{F.6}$$

for $\kappa < 1$. For our purposes, this means that

$$\lim_{\sigma \to 0} \sigma \mathcal{M}_\kappa(\sigma) = \begin{cases} 1 - \kappa, & \kappa < 1 \\ 0, & \kappa > 1 \end{cases} \tag{F.7}$$

as well as

$$\lim_{\sigma \to 0} \sigma F_\kappa(\sigma) = \lim_{\sigma \to 0} \left( \frac{1}{\sigma} \left( 1 - \frac{1}{\kappa} + \frac{\sigma \mathcal{M}_\kappa(\sigma)}{\kappa} \right) C_{\text{train}} + I_d \right)^{-1} \tag{F.8}$$

$$= \begin{cases} (x C_{\text{train}} + I_d)^{-1}, & \kappa < 1 \\ 0, & \kappa > 1 \end{cases} \tag{F.9}$$

and finally

$$\lim_{\sigma \to 0} \sigma^2 F_\kappa'(\sigma) = \begin{cases} -(x C_{\text{train}} + I_d)^{-1}, & \kappa < 1 \\ 0, & \kappa > 1 \end{cases} \tag{F.10}$$

for $x$ defined by $\text{tr} \left[ (x_1 C_{\text{train}} + I_d)^{-1} \right] = 1 - \kappa$.

Using these characterizations of the $\sigma \to 0$ limit, and noticing for our problem that if $\tau \to \infty$ (so $\tilde{\lambda} = 0$), then we must take the $\sigma \to 0$ limit of $e_{\text{ICL}}$, being careful to use

$$\tau = \frac{\rho + c_{\text{train}}}{\gamma} \frac{1}{\sigma} \tag{F.11}$$

to enforce the proportional $\alpha/\tau$ limit. Doing so gives the required answer.

The above proof requires explicitly that $C_{\text{train}}$ is invertible. If it is not, we have to be more careful with handling the $\sigma \to 0$ limit of (D.4). One potential branch of this solution is

$$1 - \frac{1}{\kappa} + \frac{1}{\kappa}\tilde{\sigma}\operatorname{tr}[(C_{\text{train}} + \tilde{\sigma})^{-1}] \to 0 \tag{F.12}$$

or equivalently

$$\frac{1}{d}\sum_{i=1}^{r} \frac{\tilde{\sigma}}{\lambda_i + \tilde{\sigma}} + \frac{1}{d}(d - r)\frac{\tilde{\sigma}}{0 + \tilde{\sigma}} \to 1 - \kappa \quad \implies \quad \frac{1}{d}\sum_{i=1}^{r} \frac{\tilde{\sigma}}{\lambda_i + \tilde{\sigma}} \to \frac{r}{d} - \kappa \tag{F.13}$$

where $r$ is the rank of $C_{\text{train}}$. This is what causes the split in behavior of the solution at $\kappa = r/d$ (which was previously 1). For $\kappa < r/d$, this is solvable at nonzero $\tilde{\sigma}$ and we end up with the familiar solution branch of

$$\sigma\mathcal{M}_\kappa(\sigma) = \tilde{\sigma}\operatorname{tr}[(C_{\text{train}} + \tilde{\sigma})^{-1}] \to 1 - \kappa. \tag{F.14}$$

For $\kappa > r/d$, there is no longer a sensible solution at nonzero $\tilde{\sigma}$ (since $\tilde{\sigma}$ cannot be negative) and so we're forced to take $\tilde{\sigma} \to 0$. This gives

$$\sigma\mathcal{M}_\kappa(\sigma) = \tilde{\sigma}\operatorname{tr}[(C_{\text{train}} + \tilde{\sigma})^{-1}] = \frac{1}{d}\sum_{i=1}^{r} \frac{\tilde{\sigma}}{\lambda_i + \tilde{\sigma}} + \frac{1}{d}(d - r)\frac{\tilde{\sigma}}{0 + \tilde{\sigma}} \to 1 - \frac{r}{d}. \tag{F.15}$$

This concludes the proof of the claimed result, and recovers the previous result for invertible $C_{\text{train}}$.

## G  CONTEXT LENGTH SHIFTS

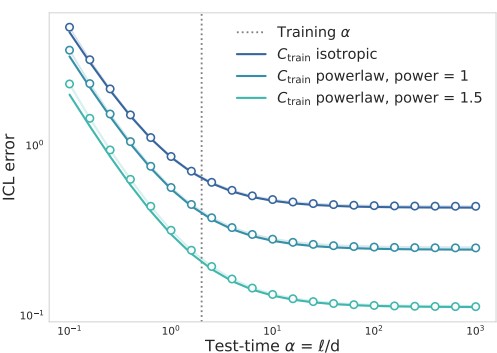

Figure 7: Theory curves with corresponding numerical simulations showing monotonicity of error in test-time context length for a variety of task structures. The dashed line corresponds to training $\alpha$. *Parameters*: $d = 150$, $\alpha = 2$, $\tau = 4$, $\kappa = 1$, $\rho = 0.01$. Testing is done on the same distribution as pretraining.

We have thus far assumed that the pretraining- and test-time context lengths are the same. Our general result in (C.7) allows for an $\alpha_{\text{test}}$ that differs from the training context length $\alpha$, but shows that this shift only affects a single scaling factor in $e_{\text{scalar}}$, where a factor of $1/\alpha$ is replaced by $1/\alpha_{\text{test}}$. Despite $e_{\text{ICL}}$ not being monotonic in training context length, as demonstrated by Lu et al. (2025), it is monotonically decreasing in test context length: testing on larger-context prompts can only decrease error, and testing on shorter-context prompts can only increase error. This is demonstrated in Figure 7 for a variety of different task structures. Recall that $\alpha_{\text{test}}$ only appears in $e_{\text{ICL}}$ through $e_{\text{scalar}}$, as part of an effective noise term: having longer test-time contexts will give the model a better estimate of the token distribution, leading to a better predictor for the task corresponding to that context.

# H    EXPERIMENTAL DETAILS

Code for all simulations can be found on Github.

## H.1    LINEAR SIMULATIONS

Linear simulations are done by sampling pretraining and testing distributions as described by (2) and (12), and computing numerical $\Gamma^*$ by (B.1). We simulate the ridgeless $\lambda \to 0$ limit of this theory by taking $\lambda = 0.00001$.

To justify the *Ansatz* of taking the value component $v_{21} = 0$, we include Figure 8 demonstrating that trained linear attention behaves similarly in ICL error to the $\mathrm{tr}(\Gamma^* H^\top)$ model, with the discrepancy decreasing as $d$ increases. For the full-parameter linear attention experiments here, we initialise $V, K, Q$ with LeCun initialisation (i.e. so $v_{21} \neq 0$), and train for 10000 epochs with BS $d^2$ using AdamW with LR 0.0001.

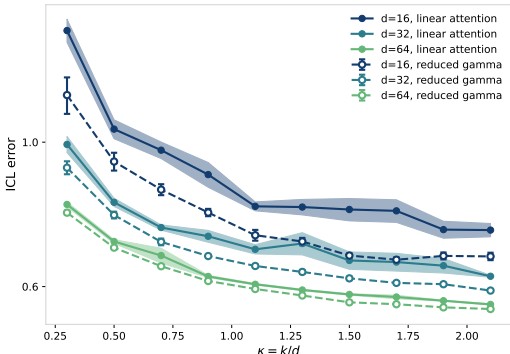

Figure 8: Comparing trained linear attention with full non-zero $V, K, Q$ parameters to reduced $\Gamma^*$ model at various $d$. Here $\rho = 0.01$, $\tau = 5$, $\alpha = 1$. Errorbars and shading show $\pm$std over 20 runs of each experiment.

We also include Figure 9 showing that explicitly initalising $v_{21} = 0$ (i.e. the chosen *Ansatz*) improves the match between trained linear attention and our reduced $\Gamma^*$ model, as expected, but this improvement is not extreme.

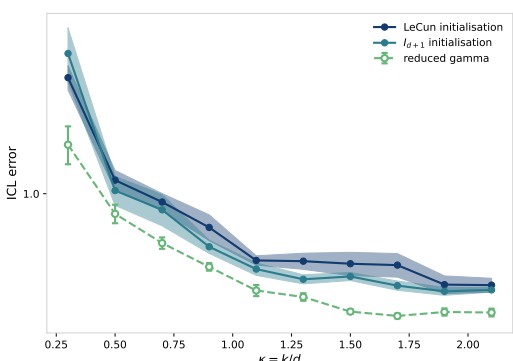

Figure 9: Comparing reduced gamma model at "low dimension" $d = 16$ to trained linear attention with both LeCun initialisation ($v_{21} \neq 0$) and $V = I_{d+1}$ initialisation ($v_{21} = 0$). Indeed the latter achieves a closer match to the reduced model, as expected, but the difference is not stark. Same parameters and training proceedure as Figure 8.

## H.2 NONLINEAR ARCHITECTURE

The architecture used for Figure 3 is a transformer formed of two transformer-blocks, constructed as follows. The input to the architecture is the $Z$ matrix (3). We apply a causal mask to ensure that each token attends to itself and prior tokens in the sequence. Each transformer-block is made of first applying single-head softmax attention to this masked input, with residual connection, and then normalised; this is then passed to a single hidden layer MLP with GELU activation, followed by a final residual connection and layer norm application. The final logit is computed by a dense layer projecting the output of the two transformer-blocks into a scalar.

The model is pretrained with data sampled by (2). We form $n$ $Z$-matrices from these samples, which are again the inputs to the architecture, and the model is pretrained to predict the corresponding $y_{\ell+1}$ value by minimising MSE loss between the final logit and $y_{\ell+1}$. Training is done using AdamW for 1000 epochs with batch size 16 and learning rate 0.0001.

Testing is done by sampling a batch of $n$ contexts from (12), and tracking MSE between the true $y_{\ell+1}$ and the model output. This test sampling is repeated 500 times to average out noise.

