# OpenReview forum: "Pretrain–Test Task Alignment Governs Generalization in In-Context Learning"
_ICLR.cc/2026/Conference — ICLR 2026 Poster_

### Official Review · Reviewer_cB3E · 2025-10-22

**Soundness:** 3
**Presentation:** 4
**Contribution:** 3
**Rating:** 6
**Confidence:** 3

**Summary:**

The authors present a theory of in-context learning of linear regression in a simplified one-layer transformer, and calculate the generalization error for in & out of distribution tasks.

**Strengths:**

The authors provide key steps towards a more complete understanding of ICL in transformers, through their analysis of a simplified transformer & problem setting.
 - The subject matter considered is relevant to the state of the field and in particular to AI safety (Understanding distribution shifts in transformers is important for safety concerns)
 - The work is, to my knowledge, original
 - The paper is presented clearly, with all mathematical objects given appropriate intuition
 - While I have not carefully checked the calculations, the results presented follow earlier work I am familiar with and the methods used seem plausible.

**Weaknesses:**

In my opinion, the main weakness of the paper as it stands is that not enough is done to explain the limits of the toy model considered. While the authors do show empirically that their model explains behaviors of non-toy, nonlinear, transformers trained on linear problems, it is not yet clear how far one can stretch this model.

In particular:
- The data is still quite simple: Does the model hold up for non-gaussian data (with a linear rule)?
- I imagine the model will break if the in-context regression problem is nonlinear. Is this correct?
- What can we learn about real data from the alignment measures proposed in the paper?
- The authors claim that mismatched task distributions are often optimal. Is there a real-world example of this phenomenon? (I.e. does the theory at least make a qualitative prediction about LLMs?)
- What further steps must be taken to bring theory & practice closer together?

**Questions:**

In addition to the questions above, I have a few more:
- From Figure 1, it seems that $e_\text{misalign}$ captures much of the behavior of $e_\text{ICL}$. In what regime is this true?
- In many figures, specific values for the thermodynamic parameters $\alpha, \tau, \kappa$ are chosen. Are the values chosen part of a typical regime? I might imagine that some of the conclusions drawn will change as these parameters are varied (as an example, the authors note that previous work identifies a phase transition at $\tau=1$)
- What are the main obstacles to generalizing the theory beyond linear problems & Gaussian data? (I.e. what goes wrong in the calculation)

---

> ### Author Response · Authors · 2025-11-21
> **Response (Part 1)**
>
> Thank you very much for your review. We agree with you strongly that there are many further extentions of this work, both on the theoretical side (data modelling, task modeling), and on the experimental side. We have begun some of these investigations and will share our preliminary thoughts here. Overall, we highlight that your questions are precisely questions that we have for future work and extentions of this model.
>
> ``The data is still quite simple: Does the model hold up for non-gaussian data (with a linear rule)?``
>
> This is something that should be further and more properly investigated, which for now we have not had time to do. We are optimistic however, that given the strong similiarity between the derivations in this work and existing high-dimensional analysis of basic ridge regression, that universality results from the ridge regression setting will transfer here (there is a substantial body of literature on universality in high-dimensional ridge regression, see for instance [Hu20, Misiakiewicz24]). In particular, the analysis in the appendix boils down to finding the deterministic equivalent for some complicated resolvent matrix. The cavity method here requires the concentration of terms like $\textbf{z}^\top G \textbf{z}$ where $G$ is the resolvent (independent of $\textbf{z}$) and $\textbf{z}$ is some vector dependent on $\textbf{x}$ and $y$. For non-Gaussian tokens and tasks, the concentration analysis of this term will be different but we don't think it will completely break. A final note is that all of our formulas depend on deterministic equivalents $F_\kappa()$ and $M_\kappa()$ for the task Wishart resolvent and Stieltjes transform. In the case of non-Gaussian tasks, these deterministic equivalences will still hold, by known results on universality for random covariance matrices (e.g. [Tao09, Pillai14]).
>
> ``I imagine the model will break if the in-context regression problem is nonlinear. Is this correct?``
>
> This is correct in the general sense, as  linear attention will not be expressive enough to both learn the task structure (during pretraining) and apply the task function at inference time. For example, considering the case of the single index model $y_i = f(\textbf{x}_i\cdot\textbf{w})$, we see very simply that linear attention is not going to be expressive enough to learn the $\textbf{x}_i\cdot\textbf{w}$ structure and also learn and apply the nonlinearity $f$, so linear attention will always implement some sub-optimal predictor for this setting. This indeed raises an interesting future question, which we currently do not have an answer to: what is the minimal architecture for ICL of simple nonlinear regression, such as single index tasks? Is softmax attention necessary? For example, in this single index case, perhaps an architecture of linear attention + MLP would be sufficient.
>
> That said, we are currently exploring whether or not the measure of alignment we derive for linear tasks here will still have some value in predicting ICL error for tasks of the form $y_i = f(\textbf{x}_i\cdot\textbf{w})$ in nonlinear transformers, with promising preliminary results (we hope to have these experiments finalised by the camera ready version).
>
> ``What can we learn about real data from the alignment measures proposed in the paper? The authors claim that mismatched task distributions are often optimal ... real-world example of this phenomenon ... qualitative prediction about LLMs?)``
>
> Due to the toy nature of the model considered here, we are hestiant in over-extrapolating our results into insights about LLMs. LLMs will consider data with far more sequential and complex structure, unlike the simplistic regression setting considered here. It's also not immediately clear how the concept of "task diversity" can be robustly defined for a realistic dataset: that is, given a dataset of ICL prompts, how should one compute how many "unique tasks" are within this dataset without knowing something further about the task structure? Generalising this work towards insights into realistic ICL setups is something that will require much more thought. The message we share here is simply that, in the case of low task diversity, it can be benificial to overfit the model to tasks that are well-represented in the test set. What this specifically looks like in a more general setting we leave to future investigation.
>
> ``What further steps must be taken to bring theory & practice closer together?``
>
> On a short-term scale, the most important thing we think is to extend the analysis around Figure 4 to realistic architectures to see if the same spectral power story still holds, and in general, what can be said about the optimal pretraining covariance choice for a realistic transformer.
>
> In general, to more fully understand ICL in transformers, an important next step is to allow for correlations between the tokens, as this is what sequence models were meant to handle; currently the tokens are completely independent.

---

> > ### Author Response · Authors · 2025-11-21
> > **Response (Part 2)**
> >
> > ``From Figure 1, it seems that e_misalign captures much of the behavior of e_ICL. In what regime is this true?``
> >
> > This depends on the value of $e_\text{scalar}(\lambda_\text{train})$, as again we have $$e_\text{ICL}(C_\text{train}, C_\text{test}) = e_\text{scalar}(\lambda_\text{train}) + e_\text{misalign}(C_\text{train}, C_\text{test}).$$ Indeed, we can see very clearly from this that $e_\text{ICL}$ will always be driven by $e_\text{misalign}$ but the specific offset between them, and in particular, the parameter ($\alpha, \kappa, \tau$) dependence of this offset is an interesting question. We haven't yet done an extensive investigation of the behaviour of this $e_\text{scalar}(\lambda_\text{train})$ term, particularly how it depends on properties of the eigenvalues $\lambda_\text{train}$ and the sample sizes $\alpha, \kappa$. One preliminary conjecture we have is that $e_\text{scalar}(\lambda_\text{train})$ is increasing in $\kappa$, but this is not proven and warrants further investigation.
> >
> > ``In many figures, specific values for the thermodynamic parameters alpha,kappa,tau are chosen. Are the values chosen part of a typical regime? I might imagine that some of the conclusions drawn will change as these parameters are varied (as an example, the authors note that previous work identifies a phase transition at tau=1 )``
> >
> > These figures focus on the effect of task diversity $\kappa$, and so to highlight that, we've taken fairly innocent values of the other parameters. Specifically, we usually take $\alpha = 1,2$ i.e. enough context length that the model isn't bottlenecked by estimating $\sum x_ix_i^\top$ but not too much that the problem is trivial. We also usually take $\tau = 4$ which puts the model in the overparameterised regime and is far enough away from the double descent at $\tau = 1$ to avoid numerical problems. Indeed a more thorough study of the effect of pretraining structure on ICL error and how this phenomenology changes with different parameter values.
> >
> > ``What are the main obstacles to generalizing the theory beyond linear problems & Gaussian data? (I.e. what goes wrong in the calculation)``
> >
> > As discussed above, the Gaussian data shouldn't be too much of a problem, so long as we're in a similiar universility class as ridge regression, which we conjecture we are. The question of linear to nonlinear tasks is more complicated. First, as also discussed earlier in this review, we would most certainly need more expressive architectures than linear attention. The problem with this is that we then may not be able to exactly write down the optimal parameters for a given batch of data, as we have been able to do here with $\Gamma^*$. This makes analysis harder, as one then probably has to resort to gradient based arguments instead of getting a sharper result. That said, this is something we are very interested in exploring.
> >
> >
> > [Hu20]
> > _Universality Laws for High-Dimensional Learning with Random Features_
> > Hong Hu, Yue M. Lu
> >
> > [Misiakiewicz24]
> > _A non-asymptotic theory of Kernel Ridge Regression: deterministic equivalents, test error, and GCV estimator_
> > Theodor Misiakiewicz, Basil Saeed
> >
> > [Pillai14]
> > _Universality of Covariance Matrices_
> > Natesh Pillai, Jun Yin
> >
> > [Tao09]
> > _Random covariance matrices: Universality of local statistics of eigenvalues_
> > Terence Tao, Van Vu

---

### Official Review · Reviewer_4Nn9 · 2025-10-27

**Soundness:** 3
**Presentation:** 3
**Contribution:** 2
**Rating:** 4
**Confidence:** 4

**Summary:**

This paper  investigates how aligning or misaligning train and test over task diversity contributes to or minimizes ICL error.  The authors use a simplified linear attention model for ICL of linear regression tasks that enables them to formally characterize ICL generalization error and that gives them a new way of measuring alignment between training and testing data sets with respect to tasks.

**Strengths:**

Understanding generalization error in ICL is clearly linked to pretraining and this paper provides a novel approach to thinking about generalization in terms of task diversity, whereas prior work has looked at other generalization problems.  This work can in principle give us a better understanding of how pretraining can affect ICL.  Intuitively task diversity is a dimension on which real ICL as opposed to memorization can emerge.

**Weaknesses:**

However if I understand correctly, task diversity here has apparently quite a limited definition.  as in equations 2, 3, 18 for example task diversity in the linear equation set up discussed here is primarily the variance in the matrix W in the linear relationship y_i = Wx_i +b for the inputs in the task.  You do not vary $d$ the dimensionality of the tokens in C train nor do you vary the distributions for x_i or b.

Thus, the authors propose an alignment metric for train and test that predicts ICL error on a limited testing suite when the main parameter of variation is the slope of the linear relationship between x and y in the input context.  As the linear relationship here is in d dimensions, in fact each dimension m can convey a "different" affine equation, $y^m = a^mx^m$.   Thus task diversity comes down to how many linear equations are encoded in the matrix W.

This is potentially very interesting.  If we knew that language generation tasks are in general expressible in terms of m linear relationships using d dimensions, then this gives us a general approach to ICL.  However, even for simple tasks and simple attention layer only models, the outputs of the attention layer formulated as a mathematical function become highly nonlinear

This is a partial step but only a partial step in learning how ICL really works and how it relies on pretraining.  The reason for this partiality is that that the divergence of test and train is very limited.  Only in the case in which we are varying C train with respect to task, in the very specific sense you have defined, do, for instance, the elements  F(z) and M(z) measure the degree to which we can capture the information in C train.

But generalization should also cover generalizing from the test data inputs for $x$ and  $b$, for the linear function learning task as in Garg et al 2022.   Naim and Asher (2024), for instance, take a detailed look at what happens when we shift distributions for inputs, slopes and intercepts (x, a, b) in a one dimensional linear function learning task.  The results show that transformer models with softmax and normalization don't really compute functions like regression and develop systematic errors as training and test distributions diverge.   Another paper, Naim and Asher 2025, shows formally that transformer architectures have inherent limitations in computing linear functions.  However, they also show that the same transformer behavior applies to polynomial functions and even some continuous functions.  There should be some discussion of how generalization on task diversity interacts with these other forms of generalization, espeically since the observation that adding task samples does not always improve performance matches Naim and Asher's observation of degradation of predictions when extending in context examples after a certain point.

More generally, omitting a discussion about how other dimensions of the problem affect distribution vitiates at least to some degree the authors' claim that "these results identify train-test task alignment as a key
determinant of generalization in ICL,"

There are also some specific worries.



In formula 4, you do add but not Norm, as is standard in transformer architectures.  With linear attention in addition there is no exponential. Thus it looks like formula 9 becomes the output of a 1 layer 1 attention head model, which Naim and Asher 2024 show doesn't really have much predictive power.

In equation 8 you drop certain terms that do contribute to the model output values for given tokens.  So I have worries about how these analytical results transfer actually to working transformer models with multiple layers, multiple attention heads and non linear scoring functions.

You are working with linear attention and with restricted assumptions on how the calculation of values in the attention layers is made (see remarks above).  So while it's really interesting that the measure works on a more standard transformer as in H2, we can't transfer the mathematical analysis very readily to the non linear transformer of H2.





The last section is interesting  but given that only task diversity seems to be changing, then it's not clear why this is so surprising

In the conclusion you speak of "fully general task covariates".  But the entire exercise is built around a linear relationship between inputs x_i and labels y_i.  So how does this provide a fully general task variation?  In many generation ICL tasks it is not at all obvious that there is any linear relationship between the input context and the generated string.  The linear relationship is a very specific ICL restriction.

Omar Naim and Nicholas Asher. Re-examining learning linear functions in context. arXiv:2411.11465
[cs.LG], 2024

Omar Naim and Nicholas Asher.  analyzing limits of icl
https://arxiv.org/pdf/2502.03503, 2025

**Questions:**

Are you predicting only for $y_n$ given a fixed input sequence $x_1, y_1, ... x_n$ ?

governing the input?

Is generalization or testing as in equation 12 is just limited to shifting the slope, as the distribution for test x input and scalar epsilon are identical to train?


Here are a few typos and grammatical suggestions:

l, 391: increases/ aggravates/exacerbates but not extremizes

l 431  all the signal

---

> ### Author Response · Authors · 2025-11-21
> **Response (Part 1)**
>
> Thank you for your detailed review, comments, and questions. We will go through your points and questions one by one, hopefully clarifying the specific setup and definitions of terms we’re using here, which we believe will help clarify the context and contributions of our work.
>
> As a general overall comment, we agree - as stated in our manuscript - with your points that this is not a perfectly general model of ICL. There are many extensions that you mention (token generality, architecture generality, task generality) that can and should be studied. In this work, we consider a minimal solvable model, and find that task alignment emerges as a quantity that affects ICL error. We certainly do not believe task alignment to be the _only_ quantity affecting ICL error, but indeed we are claiming that a more general theory of ICL will still include task alignment as a subset of its phenomenology.
>
> ``Task diversity here has apparently quite a limited definition. as in equations 2, 3, 18 for example task diversity in the linear equation set up discussed here is primarily the variance in the matrix W in the linear relationship y_i = Wx_i +b for the inputs in the task. You do not vary d the dimensionality of the tokens in C train nor do you vary the distributions for x_i or b.``
>
> In our work, the term “task diversity” specifically refers to the number $k$ of regression vectors or task vectors $\textbf{t}_j$ for $j = 1,...,k$ that are seen during pretraining. We borrow this term from the paper that introduced it [Raventos23], which uses this to again refer to the **number** of unique **tasks** seen in a pretraining batch. We are not using the term “task diversity” to claim that this is the most general data setup that can be studied; we understand that the setting analysed in our work is not the most general data model for ridge regression.

---

> ### Author Response · Authors · 2025-11-21
> **Response (Part 2)**
>
> ``But generalization should also cover generalizing from the test data inputs for x and b, for the linear function learning task as in Garg et al 2022. Naim and Asher (2024), for instance, take a detailed look at what happens when we shift distributions for inputs, slopes and intercepts (x, a, b) in a one dimensional linear function learning task.``
>
> We agree with your specific point that the tokens $x_i^\mu$ could be handled in greater generality. We sample these vectors i.i.d. with mean $0$ and isotropic covariance; this is a choice made for analytical tractability. Indeed in the early stages of the theoretical analysis for this work, we attempted to solve the theory in the general Gaussian case of tokens sampled with non-isotropic covariance matrices $\Sigma_\text{train}$ and $\Sigma_\text{test}$ at training and testing time respectively. This causes analytical problems that we could not yet resolve. Specifically, the form of the extended resolvent $[G_e(\pi)]$ as given in Appendix equation (B.43) now no longer factorises: we have
>
> $$\left(\frac{\tau}{1+\chi_{\pi}}\Sigma_{\text{train}} \otimes B_{\text{IDG}}+\pi\Sigma_{\text{test}}  \otimes B_{\mathrm{test}}+\tau\lambda I_d  \otimes I_{d+1}\right)^{-1}$$
>
> As you can see for $\Sigma_\text{train}, \Sigma_\text{test} \neq I_d$ the matrix products in this larger resolvent do not factorise, which impedes later analysis.
>
> A small point on the “bias” vector $\textbf{b}$ you are referring to in your review: this is equivalent to introducing a non-zero mean $\textbf{m}$ in the distribution of the token vectors $\textbf{x}$, which will appear in the calculation as the addition of $\textbf{m}\textbf{m}^\top$ terms to the token covariance, which leads us to the same factorisation problem described above unless $\textbf{m} = \textbf{0}$.
>
> As a philosophical point regarding the generality of token covariance, we agree that it is certainly worth studying and we have amended the discussion to include further acknowledgement of works that study this, including the works you mention in your review [Naim24, Naim25]. We further highlight a recent work that you may find interesting, that shows that the question of token covariance shift becomes very interesting as one considers ICL done by deeper architectures [Bordelon25].
>
> In the case that we study in our work here, however, since there is only one layer of linear attention, the $\Gamma$ matrix in the population limit (large context length $\ell$) should converge to something containing $\Sigma_\text{train}^{-1}$. This can be seen by expanding (9) and (10) (and ignoring label noise terms)
>
> $$\hat{y}\_{\ell+1}\approx \mathrm{tr}\left(v\_{22}M\_{11}\left(\frac{1}{\ell}\sum\_{i=1}^\ell \mathbf{x}\_i\mathbf{x}\_i^\top\right)\mathbf{w}\mathbf{x}\_{\ell+1}^\top\right)$$
>
> and noting that if the sample token covariance $\frac{1}{\ell}\sum_{i=1}^\ell \textbf{x}\_i\textbf{x}\_i^\top$ is close to the population token covariance $\Sigma_\text{train}$, then having $v_{22}M_{11} \to \Sigma_\text{train}^{-1}$ (note that this is independent of ${w}$) will give $\hat{y}\_{\ell+1} \approx \text{tr}(w x\_{\ell+1}^\top) = {w}\cdot {x}\_{\ell+1}$ independently of ${w}$, which is exactly what we want.
>
> In other words, for the specific choice of architecture and setup we chose to study here, it is irrelevant to consider distribution shift between $\Sigma_\text{train}$ and $\Sigma_\text{test}$ due to the fact that the model will always perform better on $\Sigma_\text{test} = \Sigma_\text{train}$. We highlight that this is an artifact of the particular model that we study here, and we completely agree with you that studying the effect of token distribution shift is an interesting question. Our focus for this work however is specifically **task** distribution shift, and for this reason, the modeling assumptions we make are sufficient for this purpose.
>
> ``There should be some discussion of how generalization on task diversity interacts with these other forms of generalization, espeically since the observation that adding task samples does not always improve performance matches Naim and Asher's observation of degradation of predictions when extending in context examples after a certain point.``
>
> Thank you for pointing out this work to us, we agree that it should be added to the discussion about the effect of task samples around Figure 1. We will do so for the next revision of our work.

---

> ### Author Response · Authors · 2025-11-21
> **Response (Part 3)**
>
> ``In the conclusion you speak of "fully general task covariates". But the entire exercise is built around a linear relationship between inputs x_i and labels y_i. So how does this provide a fully general task variation? In many generation ICL tasks it is not at all obvious that there is any linear relationship between the input context and the generated string. The linear relationship is a very specific ICL restriction.``
>
> We completely acknowledge that our work specifically and exclusively focuses on a **linear** task relationship between $\textbf{x}_i$ and $y_i$. The term "task covariates" specifically refers to the covariance matrices of the **task vectors** which are the $\textbf{w}$ vectors that define this linear relationship. Because we study general covariance matrices for the task vector distribution at both training and testing time, we feel justified in saying that we are studying fully general task covariates. Again, this by no means claims that we study fully general tasks (i.e. beyond linear tasks), which is a longer-term goal for our program of research in this direction.
>
> ``In formula 4, you do add but not Norm, as is standard in transformer architectures. With linear attention in addition there is no exponential. Thus it looks like formula 9 becomes the output of a 1 layer 1 attention head model, which Naim and Asher 2024 show doesn't really have much predictive power.``
>
> Indeed the linear attention model is not a very expressive architecture, and this is evidenced by the fact that it cannot always perform 0 loss on these tasks, as shown by [Naim24]. That said, we still find it to be a suitable model for initial theoretical study, as it gives us an analytically tractable setting as well as the fact that in the limit of lots of data (large context lengths and task diversity), the model can indeed actually converge to an optimal predictor as measured by the Ridge estimator, which is Bayes optimal for this task [Lu24]. So indeed the linear transformer is not the most powerful architecture for this task, but it does provide us with, in some sense, a minimally expressive model with which to begin investigations. Our focus here is using this model to derive phenomenology-based insights that extend beyond the specifics of the linear attention model, which we feel we have done.
>
> ``I have worries about how these analytical results transfer actually to working transformer models with multiple layers, multiple attention heads and non linear scoring functions._"
> We highlight that, while our alignment measure indeed is derived for the reduced-parameter linear attention model, Figure 2 indeed plots the loss of a trained transformer with **multiple layers** of softmax attention and MLPs. Beyond this, we agree that there are many further areas of future investigation to explore, such as multiple attention heads.``
>
> In general, we agree that the mathematics used here to analyse the reduced model of linear attention will not generalise to the analysis of general transformers [Naim24]. The goal of these theoretical computations is to use what we have derived in the simplified setting to gain phenomenological understanding of the problem. In this case, through our derivation of a formula for ICL error in the reduced model, we identify train-test task alignment as a quantity that affects ICL performance, and this phenomenological observation indeed transfers to nonlinear transformers despite the calculation itself not transfering. We again highlight that this is demonstrated by Figure 3.
>
> Towards your comment on nonlinear function regression, it is not clear *a priori* if the specific alignment measure that we derive for linear function ICL regression should generalise to nonlinear regression, but we expect that train-test task alignment will still be an important component of ICL error. We have preliminary experiments (which once finished will be included in the Camera Ready version), in the form of plots equivalent to Figure 3 i.e. nonlinear architectures with tasks now taken to be nonlinear (such as $y_i = \text{tanh}(\textbf{x}_i\cdot\textbf{w}) + \text{noise}$). So far in these results we see that all alignment measures (except the CKA one) are still highly correlated with ICL loss of the transformer. This suggests that covariance alignment of the task distributions will still play a role in general, but the specific measure of alignment should be further studied.

---

> ### Author Response · Authors · 2025-11-21
> **Response (Part 4)**
>
> ``Are you predicting only for y_n given a fixed input sequence x_1, y_1, …, x_n governing the input?``
>
> For a given context $(x_1,y_1,...,x_{\ell},y_{\ell},x_{\ell+1})$ as input, indeed the model outputs a single scalar logit, which we train to match $y_{\ell+1}$.
>
> ``Is generalization or testing as in equation 12 is just limited to shifting the slope, as the distribution for test x input and scalar epsilon are identical to train?``
>
> We stress there are different levels of "generality" here. The **distribution** of the tokens indeed does not change between training and testing time, but indeed the model is tested on new tokens (i.e. a fresh sample from the same distribution), and so this provides one level of in-distribution generalisation. The out-of-distribution generalisation indeed appears exclusively through the task distribution change; see the discussion above about this point as well.
>
> `` The last section is interesting but given that only task diversity seems to be changing, then it's not clear why this is so surprising. ``
>
> We'd appreciate if you could clarify your point here. This section provides a preliminary investigation regarding the tradeoff between the spectral power of the training distribution and the task diversity. Indeed, in the case of low task diversity (when we know that the model is memorising the pretraining distribution based on prior work by [Lu24]) having lower-dimensional signal (higher pretraining spectral power) means that the model can memorise highly-aligned task vectors more effectively, improving performance. Similar sentiments are echoed in [Naim25]. We highlight that this result and interpretation also applies when we consider task spectral power vs context length (measured by $\alpha$) instead of task diversity; we recover something that looks exactly like Figure 4 but with $\alpha$ on the $x$ axis.
>
> **Typos** Thanks for your attention to detail! We have decided to keep our wording the same, as we do indeed mean specifically "extremizes" and "test signal" here.
>
> [Raventos23]
> _Pretraining task diversity and the emergence of non-Bayesian in-context learning for regression_
> Allan Raventós, Mansheej Paul, Feng Chen, Surya Ganguli
>
> [Bordelon25]
> _Theory of Scaling Laws for In-Context Regression: Depth, Width, Context and Time_
> Blake Bordelon, Mary I. Letey, Cengiz Pehlevan
>
> [Naim24]
> _Re-examining learning linear functions in context_
> Omar Naim, Guilhem Fouilhé, Nicholas Asher
>
> [Naim25]
> _Analyzing limits for in-context learning_
> Omar Naim, Jerome Bolte, Nicholas Asher
>
> [Lu24]
> _Asymptotic theory of in-context learning by linear attention_
> Yue M. Lu, Mary I. Letey, Jacob A. Zavatone-Veth, Anindita Maiti, Cengiz Pehlevan

---

### Official Review · Reviewer_vZ3u · 2025-10-30

**Soundness:** 3
**Presentation:** 4
**Contribution:** 3
**Rating:** 6
**Confidence:** 4

**Summary:**

This paper studies a linear self-attention model trained on an in-context linear regression task, where the task vectors in training and at test time follow different distributions. By deriving the test error in terms of the data statistics in a high-dimensional limit, the authors quantify how the information in training is useful at test time. The analysis also reveals that the optimal pretraining distribution in the sense of minimizing the test error is not the test distribution in general.

**Strengths:**

- The paper is well-written and mostly clear. Frequently providing intuitive interpretations for the equations was very helpful for me to follow the theoretical results and their implications.
- The paper studies an interesting setting in which the in-context task distribution is different in training and at test time, providing analytical expressions for the test error in a high-dimensional limit.
- The finding that it is not always optimal to pretrain on the test distribution is interesting and somewhat surprising.
- The alignment measure introduced in this paper predicts the test error of softmax transformers more accurately than the conventional alignment measures, justifying the applicability of the theory developed in the simplified setup.

**Weaknesses:**

- I have several questions regarding the motivation to drop two terms in Equation (7).

  First, [1] showed that $v_{21}$ remains zero if initialized at zero with  $w_\mu\sim \mathcal N(0,I)$ and $n\to \infty$. Their proof relies on the mean of $w_\mu$ over $n$ sample sequences being zero. However, in this paper's setup with a finite (and possibly small) $k$, the mean of $w_\mu$ isn't close to zero. Hence, using [1] to motivate setting $v_{21}=0$ in line 165 may not be applicable.

  Second, I'm not sure why $x_ix_i^\top v_{21}$ is considered to contain no task information (line 161). One reason linear attention cannot achieve zero training/test loss, even when the noise $\epsilon=0$ and the sequence length is sufficient to recover $w$, is that the model relies on the in-context covariance estimated from the training set to approximate $\sum_{i=1}^l x_i x_i^\top$ for new test sequences, rather than using the true $x_i$ covariance of the current context. So $\sum_{i=1}^l x_i x_i^\top$ should contain essential information for robustly inferring the task vector. The limitation instead seems to come from expressivity -- single-layer linear attention lacks the expressivity to properly utilize the information in $\sum_{i=1}^l x_i x_i^\top$.

- In Equation (11), does ridge regularization in the reduced model match ridge regularization in the original model? Training the reduced model regularizing $\Gamma$ may yield different models from training the original linear attention regularizing the value, key and query weights. If they are not equivalent, it would be important to clarify this distinction.

[1] Zhang, R., Frei, S., & Bartlett, P. L. Trained transformers learn linear models in-context. 2024 JMLR.

**Questions:**

- Section 4 makes it clear that, given a test distribution, it is not always optimal to pretrain on the test distribution. But it's not yet clear to me how one should design the optimal pretraining distribution that minimizes the test error. Could the authors clarify whether this question is addressed in the paper, or if it remains an open problem?
- In line 369, what are MLP connections? Is the architecture softmax attention and MLP with skip connections?
- A very minor suggestion: it might be helpful to avoid using the word "attention" in its psychological sense (lines 38, 407) in a paper that studies the self-attention model.

---

> ### Author Response · Authors · 2025-11-21
> **Response**
>
> Thanks for your review! We really appreciate the questions you've asked and feel it's important to clarify these, which we hope we have done satisfactorily in the following.
>
> *First, [1] showed that  remains zero if initialized at zero with $w_\mu \sim N(0,I)$ and $n \to \infty$. Their proof relies on the mean of $w_\mu$ over $n$ sample sequences being zero. However, in this paper's setup with a finite (and possibly small) $k$, the mean of $w_\mu$ isn't close to zero. Hence, using [1] to motivate setting $v_{21}$ in line 165 may not be applicable*
>
> It's true that [1] (arguing $v_{21}=0$ remains stable under SGD) takes an infinite number of tasks $w_\mu \sim \mathcal N(0,I)$, so that $\mathbb{E}[w_\mu]=0$. You're right that in our finite sample setting this assumption is no longer satisfied. However, the gradient updates to $v_{21}$ depend only on the empirical mean of the mixed term $y_i x_i$, whose expectation remains zero and whose variance decays as $1/k$. Consequently, even for small $k$, $v_{21}$ receives very small updates even in the finite task sample case.
>
> *Second, I'm not sure why $x_ix_i^\top v_{21}$ is considered to contain no task information (line 161) ... $\sum x_ix_i^\top$ should contain essential information for robustly inferring the task vector. The limitation instead seems to come from expressivity -- single-layer linear attention lacks the expressivity to properly utilize the information in $\sum x_ix_i^\top$.*
>
> Indeed, the covariance term $\sum_i x_i x_i^\top$ generally contains geometric information relevant for estimating the task vector $w$. By saying "$x_i x_i^\top v_{21}$ contains no task information" we mean that, since $v_{21}$ is a fixed vector shared across tasks and independent of $y_i$ or $w$, the projection $x_i x_i^\top v_{21}$ cannot capture task-specific variation. It only contributes a task-independent offset that does not aid in estimating $w$; this echoes what you're saying about linear attention lacking the expressivity to use the information in $\sum_i x_i x_i^\top$ effectively.
>
> *In Equation (11), does ridge regularization in the reduced model match ridge regularization in the original model? Training the reduced model regularizing  may yield different models from training the original linear attention regularizing the value, key and query weights.*
>
> Perhaps we're missunderstanding your question, but we don't introduce ridge regularisation in the original model, equation (11) is the first time it's introduced. This is done for numerical stability as eventually the solution matrix for $\Gamma^\ast$ will depend on a resolvent (matrix inverse) that could be badly conditioned without the addition of the ridge. Ultimately after everything has been solved, we take $\lambda \to 0$. The simulations of the reduced $\Gamma^\ast$ model done for Figure 1 take a very small numerical value of $\lambda$ ($\lambda = 0.00001$) to mimic this limit.
>
> When we train an actual transformer (i.e. Figure 3) we do not use explicit regularisation in the loss function.
>
> *It's not yet clear to me how one should design the optimal pretraining distribution that minimizes the test error. Could the authors clarify whether this question is addressed in the paper, or if it remains an open problem?*
>
> This is a great question, which we indeed currently leave this as an open problem. Our goal in Section 4 is to illustrate this preliminary investigation with powerlaw covariances, showing the tradeoff between spectral power and task diversity. There are certainly many further steps that can and should be taken here: first, we should investigate whether this phenomena holds in realistic architectures and or if something different happens. Secondly, the main impediment to a more complete analysis of pretraining covariance optimality is that differentiating $e_\text{ICL}(C_\text{train}, C_\text{test})$ for fixed $C_\text{test}$ wrt the spectrum of $C_\text{train}$ (assume $C_\text{train}$ is diagonal) yields complicated, implicitly-defined, and uninterpretable equations for the optimal eigenvalues. The next step should be a numerical investigation, i.e. numerically optimising our $e_\text{ICL}(C_\text{train}, C_\text{test})$ over $C_\text{train}$ to see if principles emerge for optimal covariance selection.
>
> *In line 369, what are MLP connections? Is the architecture softmax attention and MLP with skip connections?*
>
> Thanks, yes this is weirdly phrased and we will ammend our language. We meant that the architecture consists of blocks of the form $$\text{out} = \text{layernorm}\left(\text{in} + \text{MLP}\left(\text{in} + \text{softmaxAttn}(\text{in})\right) \right)$$ The MLP is dense fully connected with GELU activation.

---

> > ### Comment · Reviewer_vZ3u · 2025-11-24
> >
> > Thank you for your response. Most of my questions have been effectively clarified, and I think this is a good paper overall.
> >
> > However, I retain one disagreement regarding the statement in line 167 "$v_{21} =0$ is stable under SGD". Stability is a stronger property than being a fixed point, as stability requires $v_{21}$ to converge back to zero when starting from a slightly perturbed, small nonzero value. Even in the $k\to\infty$ setting of Zhang et al. (2024), their result establishes $v_{21} =0$ only as a fixed point under gradient flow, not a stable fixed point under SGD. Both finite $k$ (instead of infinite $k$) and SGD (instead of GF) make stability harder to guarantee. In fact, for finite $k$, $v_{21} =0$ may not be a fixed point at all. Thus, the statement in line 167 appears to be inaccurate.
> >
> > That said, even if $v_{21}=0$ is not a fixed point under SGD, I consider this assumption acceptable, as the simplified model yields non-trivial insights. But the motivation for the assumption should avoid relying on statements that are not proved in this paper or existing literature.

---

> > > ### Author Response · Authors · 2025-11-24
> > > **Response regarding stability of v_{21} under SGD**
> > >
> > > Hi thanks for your response!
> > >
> > > This is a fair point. We will amend the wording in this section to soften the claim (particularly around line 167) and make things clearer that this parameter reduction of $v_{21}$ is indeed, at its core, an Ansatz that helps us simplify the model.
> > >
> > > We think that this Ansatz can be better justified through further experiments that we're currently running, specifically training a true linear transformer with full K, Q, and V matrices and comparing the final test losses to the reduced Gamma model. We're running this over a range of data parameters ($\alpha$, $\tau$, $\kappa$), task covariances, and most importantly, initialisation choice of V. We are currently seeing that $||v_{21}||$ does not deviate "too far" after training from its value at initialisation (relative to $d$), even for low $\kappa$; we stress this is preliminary and we will detail this investigation after it's finished in an additional appendix section.

---

> > > > ### Comment · Reviewer_vZ3u · 2025-11-25
> > > >
> > > > Thank you for your response and for addressing the issue with the stability statement. I have updated my rating to acceptance. I look forward to reading the final revision.

---

### Official Review · Reviewer_ZJRu · 2025-11-01

**Soundness:** 3
**Presentation:** 4
**Contribution:** 3
**Rating:** 8
**Confidence:** 4

**Summary:**

This paper studies how the structure of pretraining data affects in-context learning (ICL) using a simplified linear attention model for linear regression, where training and testing tasks come from Gaussian distributions with different covariances. The authors derive an exact analytical expression for ICL error and propose a new alignment measure between pretraining and test task distributions that accounts for finite-sample effects, task diversity, and noise. This measure correlates extremely well with actual ICL performance, even in trained nonlinear Transformers. Finally, they show that pretraining on the same task distribution is not always optimal—when task diversity is limited, models generalize better if pretrained on a lower-dimensional (power-law structured) set of tasks.

**Strengths:**

1. The paper is well written and clearly states its problem; overall, it is an engaging and enjoyable read.
2. The analysis in Section 4 is particularly insightful and well structured. The authors examine two complementary cases—fixing `C_train` to find the optimal `C_test`, and fixing `C_test` to find the optimal `C_train`—to understand when pretraining and test task structures align or conflict. The finding that pretraining on the true test distribution `C_test` can be suboptimal in low-data regimes, where a lower-dimensional or more anisotropic pretraining distribution yields better ICL performance, is especially thought-provoking.
3. The proposed alignment measure is well-motivated and technically novel. It quantifies how well the pretraining and test task structures align while explicitly accounting for finite-sample effects, task diversity, and noise—factors often ignored in simpler population-level alignment metrics.
4. The paper provides strong empirical validation: the proposed measure correlates almost perfectly with actual ICL generalization error in both the analytically solvable linear model, outperforming other common similarity metrics such as `C_test C_train^{-1}` and CKA.

**Weaknesses:**

I think the paper is beautiful from a theoretical perspective. However, from a more practical ICL standpoint:

1. The paper’s main results are construction-based. It would strengthen the work to show whether a linear attention model trained with stochastic gradient methods actually converges to the analytically derived (constructed) solution. This matters because, while constructions can represent any algorithm in principle, demonstrating convergence under realistic training dynamics would make the results more compelling for practical settings.

2. It remains unclear how additional architectural components—such as multiple layers or MLP blocks—would affect the proposed alignment and the observed “non-optimality of pretraining on the test distribution.” In more general Transformer settings, would the same phenomenon hold, or could deeper architectures mitigate or amplify this effect? Some discussion or intuition on this would be valuable.

**Questions:**

See weaknesses above and the following:

I’d appreciate the authors’ view on how Section 4 relates to the “task-scaling vs. context-scaling” perspective (e.g., Abedsoltan et al., 2024: https://arxiv.org/pdf/2410.12783). That work finds that Transformers (unlike MLPs) benefit from both (i) increasing the number of pretraining tasks and (ii) longer context at test time.

In your Fig. 4, my understanding is that context length is held fixed, while task diversity (κ) and spectral shape (p_train) are varied. Could you clarify:

1) If we allow context length to grow (while task diversity remains small), do your results predict that matching the pretraining distribution to the test distribution becomes optimal in the large-context limit? Or does some degree of structured misalignment remain beneficial?

2) Since Fig. 4 already shows that increasing task diversity removes the non-optimality, how would you expect context scaling to interact with this effect? Would larger contexts further reduce alignment sensitivity, or play an orthogonal role?

3) Finally, It would be interesting to understand whether your framework can jointly explain task and context scaling effects.

---

> ### Author Response · Authors · 2025-11-21
> **Response (part 1)**
>
> Thank you very much for your review. You raise many interesting questions and concerns that we are also interested in and have thought about throughout the course of working on this paper. We will answer your points one by one with our current preliminary results and investigations towards these questions.
>
> ``The paper’s main results are construction-based. It would strengthen the work to show whether a linear attention model trained with stochastic gradient methods actually converges to the analytically derived (constructed) solution. This matters because, while constructions can represent any algorithm in principle, demonstrating convergence under realistic training dynamics would make the results more compelling for practical settings.``
>
> We completely agree; we will provide figures showing a fully-parameterised (i.e. trainable and general $K,Q,V$ matrices) linear transformer trained on the linear regression ICL distribution with SGD and AdamW for the Camera ready version; we have not finished running these experiments at this time but we strongly agree with you that this is something that would greatly strengthen the work here.
>
> ``It remains unclear how additional architectural components—such as multiple layers or MLP blocks—would affect the proposed alignment and the observed “non-optimality of pretraining on the test distribution.” In more general Transformer settings, would the same phenomenon hold, or could deeper architectures mitigate or amplify this effect? Some discussion or intuition on this would be valuable.``
>
> Indeed we see that this non-optimality still holds in non-linear transformers given the results of Figure 3: lower error for a given $C_{\text{train}}$ can be achieved by changing $C_{\text{test}}$ away from $C_{\text{train}}$. The effect of this in more generality we agree is something that should be further investigated. In particular, we want to make the equivalent of Figure 4 for a range of nonlinear transformer architectures. This is a more computationally expensive sweep to do however, and so we cannot show any results yet.
>
> ``I’d appreciate the authors’ view on how Section 4 relates to the “task-scaling vs. context-scaling” perspective (e.g., Abedsoltan et al., 2024: https://arxiv.org/pdf/2410.12783). That work finds that Transformers (unlike MLPs) benefit from both (i) increasing the number of pretraining tasks and (ii) longer context at test time.``
>
> Indeed we discuss the effect of test-time context length in the _linear toy theory_ in Appendix G, showing that longer context lengths at test time always monotonically decrease ICL error. At the time of writing, we ran some initial investigations as to whether or not this still holds in nonlinear architectures; preliminarily we indeed found that the introduction of MLPs seems to cause the effectiveness of test-time scaling to break, completely echoing the findings of the paper you share here. We currently don't have any robust conjectures as to what is happening, although we suspect that the regime of data one is working in will matter a lot: notice that in the paper you show, when context length is being studied the task diversity is taken to be very large. We would like to study the effect of both training and test-time context length in realistic transformers far more robustly, but we currently do not have results at this time beyond the preliminary. Intuitively, a more complete description of this phenomena will require sweeps over various architectures and $\kappa$ values to disentangle all of these effects.

---

> > ### Author Response · Authors · 2025-11-21
> > **Response (part 2)**
> >
> > ``In your Fig. 4, my understanding is that context length is held fixed, while task diversity (κ) and spectral shape (p_train) are varied.``
> >
> > This is correct and you raise a great point; the interacting effect of task diversity and context length on pretrain optimality is something that we want to investigate further.
> >
> > ``If we allow context length to grow (while task diversity remains small), do your results predict that matching the pretraining distribution to the test distribution becomes optimal in the large-context limit? Or does some degree of structured misalignment remain beneficial? Since Fig. 4 already shows that increasing task diversity removes the non-optimality, how would you expect context scaling to interact with this effect? Would larger contexts further reduce alignment sensitivity, or play an orthogonal role?``
> >
> > This is a great question and we thank you a lot for raising it, because it's something we're very interested in answering properly. At the moment, we can provide you with a partial answer.
> >
> > As discussed in the prior work of [Lu24], who considered isotropic tasks, the large training context length limit is sometimes problematic. The reason for is this, schematically, is that $\alpha$ appears in these formulas through $\nu = (\rho + \text{tr}[C])/\alpha + \tilde{\lambda}$ and so the infinite context limit can correspond to a vanishing effective ridge. This causes limit interchange issues because we have already taken the explicit ridgeless limit $\lambda \to 0$. This is discussed more completely in Appendix E4 ("A note on ordering of $\lambda \to 0$ and $\alpha \to \infty$ limits") of [Lu24].
> >
> > The workaround to this problem is to take $\alpha \to \infty$ jointly with $\tau \to \infty$, which is precisely what we do in Appendix F of our work when we derive the phase transition in ICL error in $\kappa$. This provides a starting point towards answering your question, in the form of Corollary F.1. Here we see that there is distinct ICL behavior for values of low and high $\kappa$. Specifically, if $C_{\text{train}}$ is full rank, then the ICL error in this limit for $\kappa > 1$ is the Bayes error $\rho$, while for $\kappa < 1$ we still see the effect of train-test alignment through the term $$\text{trace}[C_{\text{test}}(xC_\text{train} + I)^{-1}]$$ A preliminary investigation of this term appears to show that for fixed $C_\text{test}$ then it is optimal to choose $C_\text{train}$ to be rank-1 aligned with the top eigenvector of $C_\text{test}$, echoing the result of Corollary 4.2. This requires further investigation, particularly in the case of real Transformer architectures: at large context length, is it actually optimal to pretrain on low-dimensional distributions? We have experiements running for this and hope to provide results for the camera ready version.
> >
> > ``Finally, It would be interesting to understand whether your framework can jointly explain task and context scaling effects.``
> >
> > This is indeed a larger goal we have. Indeed there are some preliminary things we can say which we've tried to highlight in this response: task diversity causes a phase transition at large contexts (Appendix F), longer test-time context length always improves error (Appendix G)
> >
> > [Lu24]
> > _Asymptotic theory of in-context learning by linear attention_
> > Yue M. Lu, Mary I. Letey, Jacob A. Zavatone-Veth, Anindita Maiti, Cengiz Pehlevan

---

### Meta-Review · Area_Chair_2vjQ · 2026-01-14

**Summary:**

This paper investigates structure of pretraining tasks effects the generalization in ICL using a simplified Transformer setup trained on linear regression tasks. This particular setup of Transformer model has been studied previously in ICL settings. The authors derive an analytical expression for the generalization error in the high-dimension and propose a new misalignment measure and show that it is a robust predictor of ICL performance.

The paper is, in general, interesting but the practical utility of the paper is somewhat arguable. Major concerns of the reviewers include: (1) the paper's definition of task diversity is incredibly narrow, (2) criticism about the theoretical setup with linear regression tasks (although prior papers also use this), (3) limited practical value , (4) strong assumptions.

The paper is interesting. However, it is unclear if the results have practical value due to the setup and assumptions used in the paper. Reviewer 4Nn9, ZJRu & vZ3u touch upon this topic. I was hoping that the empirical analysis provide more supporting evidence for the results in the paper. However, the empirical analysis of the paper is fairly limited. However, I think there is overall analysis of the paper is interesting and, I believe, it is useful for the wider community to be aware of these results and the misalignment measure.

**Reviewer Concerns:**

Reviewer 4Nn9 raised a concern regarding the definition of task diversity. The reviewer argued that this limited definition undermines the claim that the paper identifies key determinants of generalization in ICL. The authors acknowledge the limitations of the work. I believe the concerns of the reviewer are not fully addressed in my opinion.

Reviewer vZ3u raised concern about the assumption that the cross-term v_{21} remains zero under SGD. This concern was acknowledge by the authors and was addressed to some degree.

Reviewer ZJRu raised concern about the practical value of the work. I think this is a valid concern and hasn't been sufficiently addressed in my opinion. I think the paper can greatly benefit by addressing this concern carefully.

**Reviewer Scores:**

Reviewer 4Nn9 would probably keep their weak reject score. Reviewer vZ3u & Reviewer ZJRu would also probably give a weak accept score. Overall the paper is still truly on the borderline.

---

### Decision · Program_Chairs · 2026-01-26

Accept (Poster)